# Soft Robust MDPs and Risk-Sensitive MDPs: Equivalence, Policy Gradient, and Sample Complexity

**Runyu (Cathy) Zhang**
Harvard University
runyuzhang@fas.harvard.edu

**Yang Hu**
Harvard University
yanghu@g.harvard.edu

**Na Li**
Harvard University
nali@seas.harvard.edu

## Abstract

Robust Markov Decision Processes (MDPs) and risk-sensitive MDPs are both powerful tools for making decisions in the presence of uncertainties. Previous efforts have aimed to establish their connections, revealing equivalences in specific formulations. This paper introduces a new formulation for risk-sensitive MDPs, which assesses risk in a slightly different manner compared to the classical Markov risk measure [71], and establishes its equivalence with a class of soft robust MDP (RMDP) problems, including the standard RMDP as a special case. Leveraging this equivalence, we further derive the policy gradient theorem for both problems, proving gradient domination and global convergence of the exact policy gradient method under the tabular setting with direct parameterization. This forms a sharp contrast to the Markov risk measure, known to be potentially non-gradient-dominant [39]. We also propose a sample-based offline learning algorithm, namely the robust fitted-Z iteration (RFZI), for a specific soft RMDP problem with a KL-divergence regularization term (or equivalently the risk-sensitive MDP with an entropy risk measure). We showcase its streamlined design and less stringent assumptions due to the equivalence and analyze its sample complexity.

## 1 Introduction

Making decisions amidst uncertainty presents a fundamental challenge cutting across diverse domains, including finance [32, 80], engineering [45, 74], and robotics [88] etc. Within these realms, decisions carry consequences that depend not only on expected rewards but also on the level of uncertainty and associated risks. Addressing this challenge necessitates approaches such as robust, and risk-sensitive decision-making. These approaches explicitly incorporate uncertainty and aim to find policies that perform well across a spectrum of scenarios and adeptly strike a balance between expected gains and potential risks.

For robust decision-making in a dynamic environment, the robust Markov Decision Process (RMDP) is a popular framework. RMDPs model the environment as a Markov decision process, seeking policies that excel across various potential models. This involves solving a max-min problem, optimizing an objective function that considers the policy's worst-case performance across all models within a defined uncertainty set. The RMDP framework was introduced by [41, 57], spurring research into efficient planning algorithms when the model is given [34, 96, 93, 100, 56].There are also works focusing on the computational facets for these problems [37, 6, 35, 21] which leverage convex formulation and regularization techniques to tackle robustness. In cases of unknown models [1], recent efforts have designed reinforcement learning (RL) algorithms with guarantees, but most are model-based for tabular cases, i.e., requiring an empirical estimation of the probability transition model [51, 63, 105, 99, 98, 77], thereby impeding their applicability to large state spaces. Some works focus on the model-free setting and employ linear function approximation for handling large state spaces [85, 70, 4]. However, these approaches provide only asymptotic guarantees and rely on approximated robust dynamic programming, which inherently is computationally more expensive than standard dynamic programming. A recent contribution by [64] offers non-asymptotic sample

---

[1]By 'unknown model' we refer to the setting where the nominal probability transition model is unknown. Both model-based and model-free methods belong to this setting, where model-based methods keep an empirical estimate of the nominal model whereas model-free algorithms don't require this empirical estimation step.

complexity guarantees in the context of model-free robust RL. This achievement, however, introduces additional dual variables, thus adding additional computational complexity and imposing more stringent assumptions.

An alternative approach for handling uncertainty is risk-sensitive decision-making, which intriguingly shares an elegant equivalence with robust decision-making. The concept of coherent risk measures was initially introduced and explored in [2, 18, 69], where the uncertainty is represented by a static random variable. The connection to robustness was established by characterizing risk measures as the infimum of expected shortfall across a set of probability measures, known as the risk envelope. The risk notion is further extended to convex risk measures which capture a broader class of risk evaluation functions [30, 73, 31]. Subsequently, conditional and dynamic risk measures were introduced to generalize risk assessment from static random variables to stochastic processes [3, 14, 29, 22, 68, 72, 65]. In particular, [71] introduces the *Markov risk measure* in the context of Markov Decision Processes (MDPs). However, the equivalence between the Markov risk measure and robust MDPs is not as straightforward as in static settings. Notably, [71, 75, 11, 5, 62] established the equivalence between optimizing the Markov risk measure and solving a modified RMDP problem, where the uncertainty set dynamically changes with the implemented policy. This differs from the standard RMDPs, where the uncertainty sets are typically unrelated to the policy. Though [62] attains stronger equivalence results with RMDPs, it is only applicable to specific risk measures, such as Conditional Value at Risk (CVaR). Similar to RMDPs, optimizing Markov risk measures also faces many challenges. Firstly, building upon the equivalence with the modified RMDP with policy-dependent uncertainty set, Huang et al. [39] highlights that, even in a tabular setting with direct parameterization, Markov risk measures may lack gradient-dominance – a stark contrast to the gradient domination observed in standard MDPs [1]. This implies that policy gradient algorithms may not ensure global optima, even in a straightforward, full-information environment. Further, the sample complexity is also harder to obtain. While there is a series of efforts dedicated to optimizing the Markov risk measure within the realm of RL [12, 76, 46], these works primarily provide asymptotic convergence results.

The challenges outlined above motivate us to investigate the potential of introducing an alternative risk formulation. This new formulation seeks to capture risk in a way similar to Markov risk measures while achieving a stronger and broader equivalence with RMDPs. Moreover, we aim to enhance convergence properties, including the crucial aspect of gradient domination. These improvements are poised to support the development of learning algorithms for both RMDPs and risk-sensitive MDPs while maintaining provable guarantees.

**Our Contributions:** In this paper, we propose a new formulation for risk-sensitive MDP, whose definition incorporates the general concepts of convex risk measures. We first establish the equivalence of risk-sensitive MDP with a class of soft RMDP problems, which includes the standard RMDP as a special case. Leveraging this equivalence, we proceed to derive the policy gradient theorem for both the aforementioned class of soft RMDPs and risk-sensitive MDPs (Theorem 3) and prove the global convergence of the exact policy gradient method under the tabular setting with direct parameterization. Our result, to the best of our knowledge, presents the first global convergence analysis with iteration complexity for a general class of risk-sensitive MDPs.

Based on the policy gradient theorem, we also highlight the difficulty of gradient estimation using samples compared with the standard MDP setting, motivating us to seek other types of sample-based learning methods. In the last part of this paper, we mainly focus on the setting of offline learning with nonlinear function approximation which is a relatively less-studied scenario, and propose a sample-based offline learning algorithm, namely the robust fitted-Z iteration (RFZI), that resembles policy iteration rather than policy gradient.Specifically, we focus on a setting where the regularization term for the RMDP is a KL-divergence term, which is equivalent to the risk-sensitive MDP with the entropy risk measure. The algorithm utilizes the equivalence between the two problems, which enables simpler algorithm design. Notably, our algorithm is model-free and does not rely on an empirically estimated probability transition model. The sample complexity for RFZI is also provided. Compared with [64] which considers offline robust RL with sample-complexity guarantees, our work considers a different uncertainty set, requires less computational and implementation complexity, and less stringent assumptions.

Due to space limit, we defer a detailed literature review and numerical simulations to the appendix.

## 2 PROBLEM SETTINGS AND PRELIMINARIES

**Markov Decision Processes (MDPs).** A finite Markov decision process (MDP) is defined by a tuple $\mathcal{M} = (\mathcal{S}, \mathcal{A}, P, r, \gamma, \rho)$, where $\mathcal{S}$ is a finite set of states, $\mathcal{A}$ is a finite set of actions available to the agent, and $P$ is the transition probability function such that $P(s'|s, a)$ describes the probability of transitioning from one state $s$ to another $s'$ given a particular action $a$. For the sake of notation simplicity, we use $P_{s,a}$ to denote the probability distribution $P(\cdot|s, a)$ over the state space $\mathcal{S}$. $r : \mathcal{S} \times \mathcal{A} \to [0, 1]$ is a reward function, $\gamma \in [0, 1)$ is a discounting factor, and $\rho$ specifies the initial probability distribution over the state space $\mathcal{S}$.

A stochastic policy $\pi : \mathcal{S} \to \Delta^{|\mathcal{A}|}$ specifies a strategy where the agent chooses its action based on the current state in a stochastic fashion; more specifically, the probability of choosing action $a$ at state $s$ is given by $\Pr(a|s) = \pi(a|s)$. A deterministic policy is a special case of the stochastic policy where for every state $s$ there is an action $a_s$ such that $\pi(a_s|s) = 1$. For notation simplicity, we slightly overload the notation and use $\pi(s)$ to denote the action $a_s$ for deterministic policies. For a given stationary policy $\pi$ and a set of transition probability distributions $\{P_{s,a}\}_{s\in\mathcal{S}, a\in\mathcal{A}}$, we denote the discounted state visitation distribution by

$$d^{\pi,P}(s) := (1-\gamma) \sum_{t=0}^{+\infty} \gamma^t \Pr^{\pi,P}(s_t = s \mid s_0 \sim \rho).$$

**Robust MDPs (RMDPs) and Soft Robust MDPs.** Unlike the standard MDP which considers a fixed transition model $\{P_{s,a}\}$, the robust MDP considers a set $\mathcal{P}$ of transition probability distributions and aims to solve the sup-inf problem [41]

$$\sup_\pi \inf_{\{\widehat{P}_t \in \mathcal{P}\}_{t\geq 0}} \mathbb{E}_{s_t, a_t \sim \pi, \widehat{P}, s_0 \sim \rho} \sum_{t=0}^{+\infty} \gamma^t \left(r(s_t, a_t)\right)^2 \tag{1}$$

where the objective is to find the best action sequence that maximizes a worst-case objective over all possible models in the uncertainty set $\mathcal{P}$. Many papers [41, 57, 99, 63, 4] consider the uncertainty set under the $(s, a)$-rectangularity condition $\mathcal{P} = \otimes_{s\in\mathcal{S}, a\in\mathcal{A}} \mathcal{P}_{s,a}$, where $\mathcal{P}_{s,a} = \{\widehat{P}_{s,a} : \ell(\widehat{P}_{s,a}, P_{s,a}) \leq \epsilon\}$, and $\ell$ is a penalty function that captures the deviation of $\widehat{P}_{s,a}$ from a nominal model $P_{s,a}$. Some popular penalty functions are KL divergence, total variation distance, etc.

In this paper, we generalize the above robust MDP problem to a wider range of problems which we call the *soft robust MDP*[3]. The objective of the soft robust MDP solves the following sup-inf problem:

$$\sup_\pi \inf_{\{\widehat{P}_t\}_{t\geq 0}} \mathbb{E}_{s_t, a_t \sim \pi, \widehat{P}, s_0 \sim \rho} \sum_{t=0}^{+\infty} \gamma^t \left(r(s_t, a_t) + \gamma D(\widehat{P}_{t;s_t,a_t}, P_{s_t,a_t})\right). \tag{2}$$

Note that here $\inf_{\{\widehat{P}_t\}_{t\geq 0}}$ is with respect to all the possible state-transition probability distributions. When the penalty function $D$ is chosen as the indicator function

$$D(\widehat{P}_{s,a}, P_{s,a}) = \begin{cases} 0 & \ell(\widehat{P}_{s,a}, P_{s,a}) \leq \epsilon \\ +\infty & \text{otherwise} \end{cases},$$

it recovers the robust MDP problem (1). When $D$ is set as non-indicator functions, for example, $D(\widehat{P}_{s,a}, P_{s,a}) = \text{KL}(\widehat{P}_{s,a}||P_{s,a})$, Problem (2) is a robust MDP with a soft penalty term $D$ on the deviation of $\widehat{P}_{s,a}$ from $P_{s,a}$ rather than a hard constraint on $\widehat{P}_{s,a}$.

Similar to the robust MDP problem, we can define the optimal value function as

$$\overline{V}^\star(s) := \sup_\pi \inf_{\{\widehat{P}_t\}_{t\geq 0}} \mathbb{E}_{s_t \sim \widehat{P}} \left[ \sum_{t=0}^{+\infty} \gamma^t \left(r(s_t, a_t) + \gamma D(\widehat{P}_{t;s_t,a_t}, P_{s_t,a_t})\right) \Big| s_0 = s \right]. \tag{3}$$

Additionally, given a stationary policy $\pi$, the value function $\overline{V}^\pi$ under policy $\pi$ is defined as follows:

$$\overline{V}^\pi(s) := \inf_{\{\widehat{P}_t\}_{t\geq 0}} \mathbb{E}_{s_t, a_t \sim \pi, \widehat{P}} \left[ \sum_{t=0}^{+\infty} \gamma^t \left(r(s_t, a_t) + \gamma D(\widehat{P}_{t;s_t,a_t}, P_{s_t,a_t})\right) \Big| s_0 = s \right]. \tag{4}$$

We also define the corresponding Q-functions as

$$\overline{Q}^\star(s, a) := \sup_{\{a_t\}_{t\geq 1}} \inf_{\{\widehat{P}_t\}_{t\geq 0}} \mathbb{E}_{s_t \sim \widehat{P}} \left[ \sum_{t=0}^{+\infty} \gamma^t \left(r(s_t, a_t) + \gamma D(\widehat{P}_{t;s_t,a_t}, P_{s_t,a_t})\right) \Big| s_0 = s, a_0 = a \right]$$

$$\overline{Q}^\pi(s, a) := \inf_{\{\widehat{P}_t\}_{t\geq 0}} \mathbb{E}_{s_t, a_t \sim \pi, t\geq 1, \widehat{P}} \left[ \sum_{t=0}^{+\infty} \gamma^t \left(r(s_t, a_t) + \gamma D(\widehat{P}_{t;s_t,a_t}, P_{s_t,a_t})\right) \Big| s_0 = s, a_0 = a \right].$$

[2]For the sake of generality, we allow the transition probability to be non-stationary and the policy to be non-Markovian and stochastic. However, in later sections we will show that the sup-inf solution can be obtained by a stationary deterministic Markov policy and a stationary transition probability (Theorem 2).

[3]We adopt the term from robust optimization literature, the concept of regularizing the adversaries actions is referred as soft-robustness [9] (or comprehensive robustness [8] and globalized robustness [10]).

**Remark 1** (Soft Robust MDP.). *The soft robust MDP problem is useful, especially when the uncertainty set is not explicitly given. In this case, it is more desirable to consider all possible probability transition models $\{\widehat{P}_t\}_{t \geq 0}$ while treating the deviation from the nominal model as a soft penalty term $D$ rather than constraining it to be within a specified uncertainty set.*

In this paper, we establish a connection between the soft robust MDP and another class of MDPs, namely risk-sensitive MDPs. To define the risk-sensitive MDP, we will first introduce the notation of convex risk measures.

**Convex Risk Measures [30].** Consider a finite set $\mathcal{S}$, let $\mathbb{R}^{|\mathcal{S}|}$ denote the set of real-valued functions over $\mathcal{S}$. A convex risk measure $\sigma : \mathbb{R}^{|\mathcal{S}|} \to \mathbb{R}$ is a function that satisfies the following properties:

1. Monotonicity: for any $V', V \in \mathbb{R}^{|\mathcal{S}|}$, if $V' \leq V$, then $\sigma(V) \leq \sigma(V')$.

2. Translation invariance: for any $V \in \mathbb{R}^{|\mathcal{S}|}, m \in \mathbb{R}, \sigma(V + m) = \sigma(V) - m$.

3. Convexity: for any $V', V \in \mathbb{R}^{|\mathcal{S}|}, \lambda \in [0, 1], \sigma(\lambda V + (1 - \lambda)V') \leq \lambda \sigma(V) + (1 - \lambda)\sigma(V')$.

Using standard duality theory, it is shown in classical results [30] that convex risk measures satisfy the following dual representation theorem:

**Theorem 1** (Dual Representation Theorem [30]). *The function $\sigma : \mathbb{R}^{|\mathcal{S}|} \to \mathbb{R}$ is a convex risk measure if and only if there exists a "penalty function" $D(\cdot) : \Delta^{|\mathcal{S}|} \to \mathbb{R}$ such that*

$$\sigma(V) = \sup_{\widehat{\mu} \in \Delta^{|\mathcal{S}|}} \left( -\mathbb{E}_{\widehat{\mu}} V - D(\widehat{\mu}) \right). \tag{5}$$

*Further, the penalty function $D$ can be chosen to satisfy the condition $D(\widehat{\mu}) \geq -\sigma(0)$ for any $\widehat{\mu} \in \Delta^{|\mathcal{S}|}$ and it can be taken to be convex and lower-semicontinuous. In specific, it can be written in the following form:*

$$D(\widehat{\mu}) = \sup_V \left( -\sigma(V) - \mathbb{E}_{s \sim \widehat{\mu}} V(s) \right) \tag{6}$$

Note that $\sigma$ and $D$ serve as the Fenchel conjugate of each other. In most cases, the convex risk measure $\sigma(V)$ can be interpreted as the risk associated with a random variable that takes on values $V(s)$ where $s$ is drawn from some distribution $s \sim \mu$. Consequently, most commonly used risk measures are typically associated with an underlying probability distribution $\mu \in \Delta^{|\mathcal{S}|}$ (e.g., Examples 1). This paper focuses on this type of risk measures and thus we use $\sigma(\mu, \cdot)$ to denote the risk measure, where the additional variable $\mu$ indicates the associated probability distribution. Correspondingly, we denote the penalty term $D(\hat{\mu})$ of $\sigma(\mu, \cdot)$ in the dual representation theorem as $D(\hat{\mu}, \mu)$.[4]

Here we provide an example of convex risk measure and its dual form.

**Example 1** (Entropy risk measure [30]). *For a given $\beta > 0$, the entropy risk measure takes the form:*

$$\sigma(\mu, V) = \beta^{-1} \log \mathbb{E}_{s \sim \mu} e^{-\beta V(s)}.$$

*Its corresponding penalty function $D$ in the dual representation theorem is the KL divergence*

$$D(\widehat{\mu}, \mu) = \beta^{-1} \text{KL}(\widehat{\mu}||\mu) = \beta^{-1} \sum_{s \in \mathcal{S}} \widehat{\mu}(s) \log (\widehat{\mu}(s)/\mu(s)).$$

**Risk-Sensitive MDPs.** Convex risk measures capture the risk associated with random variables. It would be desirable if the notion could be adapted to the MDP to capture the risk of a given policy under the Markov process. Given an MDP $\mathcal{M}$, a class of convex risk measures $\{\sigma(P_{s,a}, \cdot)\}_{s \in \mathcal{S}, a \in \mathcal{A}}$, and a policy $\pi(\cdot|s)$, the risk-sensitive value function $\widetilde{V}^\pi$ for the infinite discounted MDP is given as

$$\widetilde{V}^\pi(s) = \sum_a \pi(a|s) \left( r(s, a) - \gamma \sigma(P_{s,a}, \widetilde{V}^\pi) \right), \forall s \in \mathcal{S}. \tag{7}$$

With the definition of risk-sensitive $\widetilde{V}^\pi$, the risk-sensitive MDP problem is to find the policy that maximizes $\max_\pi \widetilde{V}^\pi$. We denote the optimal value by $\widetilde{V}^\star$, which is the fix-point solution of the following equation,

---

[4]Please note that the symbol $D$ serves a dual purpose, representing both the regularization term in (2) and the penalty function for a risk measure in (5) and (6). This intentional notation overlap will become clear in the following sections, which reveal the connection between these two terms.

$$\widetilde{V}^\star(s) := \max_a \left( r(s,a) - \gamma\sigma(P_{s,a}, \widetilde{V}^\star) \right), \forall s \in \mathcal{S}. \tag{8}$$

It is worth noting that the fixed point operators for (7),(8) are contractive (proof deferred to Appendix D), which immediately implies the following lemma which verifies that the fixed point equations for $\widetilde{V}^\pi$ (7) and $\widetilde{V}^\star$ (8) are well-defined .

**Lemma 1.** *The solution to* (7) *exists and is unique. Same argument holds for* (8).

**Remark 2.** *We would like to emphasize that when the policy $\pi$ is stochastic, our definition of the value functions $\widetilde{V}^\pi$ are different from the Markov risk measures defined in [71, 39, 86, 87]* [5]. *However, the two quantities are equivalent when $\pi$ is deterministic. Additionally, when further assuming that the risk measure $\sigma$ is mixture quasiconcave (c.f. [17]), the optimal policy for the Markov risk measure is also deterministic and thus the risk-sensitive MDP and the Markov risk measure obtain the same optimal value $\widetilde{V}^\star$* [6] *(see Appendix C for more details).*

We also define the Q-function of the risk sensitive MDP as:

$$\widetilde{Q}^\star(s,a) := r(s,a) - \gamma\sigma(P_{s,a}, \widetilde{V}^\star), \quad \widetilde{Q}^\pi(s,a) := r(s,a) - \gamma\sigma(P_{s,a}, \widetilde{V}^\pi).$$

**Other notations:** For any function $f : \mathcal{S} \times \mathcal{A} \to \mathbb{R}$, state-action distribution $\mu \in \Delta(\mathcal{S} \times \mathcal{A})$ the $\mu$-weighted 2-norm of $f$ is defined as $\|f\|_{2,\mu} = \left( \mathbb{E}_{s,a\sim\mu} f(s,a)^2 \right)^{1/2}$.

## 3 EQUIVALENCE OF SOFT RMDPs AND RISK-SENSITIVE MDPs

**Theorem 2** (Equivalence of Soft RMDPs and Risk-Sensitive MDPs). *For a given MDP $\mathcal{M}$, a penalty function $D$, a class of convex risk measures $\{\sigma(P_{s,a}, \cdot)\}$, and a stationary policy $\pi$, if the penalty function $D$ satisfies*

$$D(\widehat{P}_{s,a}, P_{s,a}) = \sup_V \left( -\sigma(P_{s,a}, V) - \mathbb{E}_{s'\sim\widehat{P}_{s,a}} V(s') \right), \tag{9}$$

*then the value functions and Q-functions of the soft RMDP and the risk-sensitive MDP are always the same. That is, $\overline{V}^\star = \widetilde{V}^\star =: V^\star$, $\overline{V}^\pi = \widetilde{V}^\pi =: V^\pi$, $\overline{Q}^\star = \widetilde{Q}^\star =: Q^\star$, $\overline{Q}^\pi = \widetilde{Q}^\pi =: Q^\pi$.* [7]

*Further, for every initial state $s_0$, the sup-inf solution of the policy and transition probabilities for $V^\star(s_0)$ defined in (3) is given by:*

$$\pi^\star(s) = \operatorname{argmax}_a \left( r(s,a) - \gamma\sigma(P_{s,a}, V^\star) \right), \tag{10}$$
$$\widehat{P}^\star_{t;s,a} = \widehat{P}^\star_{s,a} = \operatorname{argmin}_{\widehat{P}} D(\widehat{P}, P_{s,a}) + \mathbb{E}_{s'\sim\widehat{P}} V^\star(s').$$

*where* (10) *means that the optimal action sequence $\{a_t\}_{t\geq 1}$ can be achieved by implementing the deterministic policy $a_t = \pi^\star(s_t)$.*

*Similarly, for any initial state $s_0$, the minimum solution of the transition probabilities for $V^\pi(s_0)$ defined in (4) is given by*

$$\widehat{P}^\pi_{t;s,a} = \widehat{P}^\pi_{s,a} = \operatorname{argmin}_{\widehat{P}} D(\widehat{P}, P_{s,a}) + \mathbb{E}_{s'\sim\widehat{P}} V^\pi(s'). \tag{11}$$

Since Theorem 2 has established the equivalence of risk-sensitive MDPs and soft RMDPs, *from now on we use $V^\star, V^\pi, Q^\star, Q^\pi$ to denote the value functions and Q-functions for both settings and assume by default that the penalty function $D$ and the risk measure $\sigma$ satisfy relationship* (9).

**Remark 3.** *As a comparison to the equivalence result for the Markov risk measures [72, 71, 86], their uncertainty set for the robust problem generally depends on the policy $\pi$ (see e.g. Assumption 2.2 in [86]), while in our setting, the penalization function $D$ is independent of the policy and matches with the most standard formulation of RMDPs.*

---

[5]Due to this difference, the value function $V^\pi$ can no longer be written as $\rho(\sum_{t=0}^{+\infty} \gamma^t r(s_t, a_t))$ where $\rho$ is a time-consistent dynamic risk measure. This makes our definition different from the usual interpretation of the dynamic risk measures.

[6]We would like to note that the equivalence of optimal value might fail if $\sigma$ is not mixture semiconcave (e.g. mean (semi)-deviation, mean (semi)-moment measures [17]) or if policy regularization is added into the value function because the optimal policy might no longer be deterministic.

[7]The equivalence $\overline{V}^\pi = \widetilde{V}^\pi, \overline{Q}^\pi = \widetilde{Q}^\pi$ easily extends to the setting with policy regularization, since adding regularization only requires changing the reward function $r(s,a)$ to be $r^\pi(s,a) = r(s,a) + \mathcal{R}(\pi(\cdot|s))$, where $\mathcal{R}$ is the policy regularizer, in which case the proof of Theorem 2 can still carry through naturally.

# 4 POLICY GRADIENT FOR SOFT RMDPs

In this section, we present the policy gradient theorem for a differentiable policy $\pi_\theta$ parameterized by $\theta$, which provides an analytical method for computing the gradient in soft RMDPs. Additionally, we prove the global convergence of the exact policy gradient ascent algorithm for the direct parameterization case. For simplicity, in this section we use the abbreviations $V^\theta, Q^\theta, \widehat{P}^\theta, V^{(t)}, Q^{(t)}, \widehat{P}^{(t)}$ to denote $V^{\pi_\theta}, Q^{\pi_\theta}, \widehat{P}^{\pi_\theta}, V^{\pi_{\theta^{(t)}}}, Q^{\pi_{\theta^{(t)}}}, \widehat{P}^{\pi_{\theta^{(t)}}}$, respectively.

**Theorem 3** (Policy gradient theorem). *Suppose that $\pi_\theta$ is differentiable with respect to $\theta$ and that $\sigma(P_{s,a}, \cdot) : \mathbb{R}^{|\mathcal{S}|} \to \mathbb{R}$ is a differentiable function, then $V^\theta(s)$ is also a differentiable function with respect to $\theta$ and the gradient is given by*

$$\nabla_\theta V^\theta(s) = \mathbb{E}_{a_t \sim \pi_\theta(\cdot|s_t), s_{t+1} \sim \widehat{P}^\theta_{s_t,a_t}} \left[ \sum_{t=0}^{+\infty} \gamma^t Q^\theta(s_t, a_t) \nabla_\theta \log \pi_\theta(a_t|s_t) \Big| s_0 = s \right],$$

*where $\widehat{P}^\theta$ is defined in* (11).

We leave the discussion of this result to the end of this section in Remark 4. Theorem 3 immediately implies the following corollary on the policy gradient under direct parameterization (c.f. [1, 94]), where the parameter $\theta_{s,a}$ directly represents the probability of choosing action $a$ at state $s$, i.e., $\theta_{s,a} = \pi_\theta(a|s)$.

**Corollary 1** (Policy gradient for direct parameterization). *Under direct parameterization,*

$$\frac{\partial \mathbb{E}_{s_0 \sim \rho} V^\theta(s_0)}{\partial \theta_{s,a}} = \frac{1}{1-\gamma} d^{\pi_\theta, \widehat{P}^\theta}(s) Q^\theta(s, a). \tag{12}$$

Note that the policy gradient theorem only holds for the case where $\sigma(P_{s,a}, \cdot)$ is differentiable; nevertheless, we can generalize (12) to the non-differentiable case by defining the variable $G(\theta) \in \mathbb{R}^{|\mathcal{S}| \times |\mathcal{A}|}$ as follows:

$$[G(\theta)]_{s,a} := \frac{1}{1-\gamma} d^{\pi_\theta, \widehat{P}^\theta}(s) Q^\theta(s, a).$$

For both differentiable and non-differentiable cases, we could perform the following ('quasi'-)gradient ascent algorithm:

$$\theta^{(t+1)} = \text{Proj}_\mathcal{X}(\theta^{(t)} + \eta G(\theta^{(t)})), \tag{13}$$

where $\mathcal{X} = \otimes_{s \in \mathcal{S}} \Delta^{|\mathcal{A}|}$ denotes the feasible region of $\theta$. For the standard MDP case, it is known that the value function satisfies the gradient domination property under direct parameterization [1], which enables global convergence of the policy gradient algorithm. Similar properties also hold for the soft RMDP/risk-sensitive MDP setting which is shown in the following lemma:

**Lemma 2** (Gradient domination under direct parameterization).

$$\mathbb{E}_{s_0 \sim \rho} V^\star(s_0) - V^\theta(s) \le \left\| \frac{d^{\pi^\star, \widehat{P}^\theta}}{d^{\pi_\theta, \widehat{P}^\theta}} \right\|_\infty \max_{\overline{\pi}} \langle \overline{\pi} - \pi_\theta, G(\theta) \rangle,$$

*where* $\left\| \frac{d^{\pi^\star, \widehat{P}^\theta}}{d^{\pi_\theta, \widehat{P}^\theta}} \right\|_\infty := \max_s \frac{d^{\pi^\star, \widehat{P}^\theta}(s)}{d^{\pi_\theta, \widehat{P}^\theta}(s)}$.

The gradient domination property suggests that as long as the term $\left\| \frac{d^{\pi^\star, \widehat{P}^\theta}}{d^{\pi_\theta, \widehat{P}^\theta}} \right\|_\infty$ is not infinite, all the first order stationary points are global optimal solutions. Based on this observation, we further derive the convergence rate for the policy gradient algorithm. Before that, we introduce the following sufficient exploration assumption:

**Assumption 1** (Sufficient Exploration). *For any policy $\pi$, it holds that $d^{\pi, \widehat{P}^\pi}(s) > 0$, where $\widehat{P}^\pi$ is defined as in* (11). *We define the distributional shift factor $M$ to be a constant that satisfies $M \ge \frac{1}{d^{\pi, \widehat{P}^\pi}(s)}$ for all state $s$ and policy $\pi$.*

Note that when we start with a initial distribution where $\rho(s) > 0$ for every state $s$, the term $M$ can be upper bounded by $\frac{1}{(1-\gamma) \min_s \rho(s)}$. If Assumption 1 is satisfied, it can be concluded that $\left\| \frac{d^{\pi^\star, \widehat{P}^\theta}}{d^{\pi_\theta, \widehat{P}^\theta}} \right\|_\infty \le M$. Thus we could use gradient domination to derive the global convergence rate.

**Theorem 4** (Convergence rate for exact policy gradient under direct parameterization). *Under Assumption 1, by setting $\eta = \frac{(1-\gamma)^3}{2|\mathcal{A}|M}$, running (13) guarantees that*

$$\sum_{k=1}^{K} \left( \mathbb{E}_{s_0 \sim \rho} V^\star(s_0) - V^{(k)}(s_0) \right)^2 \leq \frac{16|\mathcal{A}|M^4}{(1-\gamma)^4}.$$

*Therefore, by setting $K \geq \frac{16|\mathcal{A}|M^4}{(1-\gamma)^4 \epsilon^2}$, it is guaranteed that* $\min_{1 \leq k \leq K} \mathbb{E}_{s_0 \sim \rho}(V^\star(s_0) - V^{(k)}(s_0)) \leq \epsilon$.

If we apply the same proof technique to standard MDPs, the convergence rate is $O\left( \frac{|\mathcal{A}|M^2}{(1-\gamma)^4} \right)$. The dependency on the distributional shift factor $M$ is worse for soft RMDPs, which is caused by the choice of a smaller stepsize $\eta$ (see Remark 7 in the Appendix for more details). It is an interesting open question whether this worse dependency is fundamental or just a proof artifact.

**Remark 4** (Difficulties of Sample-based Gradient Estimation). *Though Theorem 4 establishes the global convergence of exact policy gradient, it is hard to generalize the result to sample-based settings. Note that the policy gradient in Theorem 3 takes a similar form as compared to standard MDPs [83], however, there's a primary distinction that the expectation is taken over trajectories sampled from the probability transition model $\widehat{P}^\theta$ instead of the nominal model $P$. Consequently, when confined to samples exclusively from the nominal model, estimating this expectation becomes exceptionally challenging, particularly in the context of non-generative models.*

## 5 OFFLINE REINFORCEMENT LEARNING OF THE KL-SOFT RMDP

Since the previous section considers learning with full information and studies iteration complexity, the major motivation for this section is to examine sample-based learning for risk sensitivity MDPs and soft robust MDPs. As discussed in Remark 4, developing sample-based policy gradient learning methods might be difficult, therefore, we seek an alternative sample-based method that resembles policy iteration rather than policy gradient. Specifically, we mainly focus on the setting of offline learning with nonlinear function approximation which is a relatively less-studied scenario. Moreover, due to the challenge in developing a method for soft MDPs with general $D$ functions (or equivalently for risk sensitive MDPs with general risk functions $\sigma$), in this section, we look into a particular and important case of soft RMDP where the regularization term is the KL-divergence, i.e.,

$$\max_\pi \min_{\widehat{P}_t} \mathbb{E}_{s_t, a_t \sim \pi, \widehat{P}_t, s_0 \sim \rho} \sum_{t=0}^{+\infty} \gamma^t \left( r(s_t, a_t) + \gamma\beta^{-1} \mathrm{KL}(\widehat{P}_{t;s_t,a_t} || P_{s_t,a_t}) \right). \quad (14)$$

The hyperparameter $\beta$ represents the penalty strength of the deviation of $\widehat{P}$ from $P$, the smaller $\beta$ is, the larger the penalty strength. From Example 1 and Theorem 2, the KL-soft RMDP is equivalent to the risk-sensitive MDP problem with the risk measures $\sigma(P_{s,a}, \cdot)$ chosen as the entropy risk measure

$$\sigma(P_{s,a}, V) = \beta^{-1} \log \mathbb{E}_{s' \sim P_{s,a}} e^{-\beta V(s')}.$$

In this case, the Bellman equations for the value functions $V^\pi, V^\star, Q^\pi, Q^\star$ are given by:

$$V^\pi(s) = \sum_a \pi(a|s)Q(s,a), \quad Q^\pi(s,a) = r(s,a) - \gamma\beta^{-1} \log \mathbb{E}_{s' \sim P_{s,a}} e^{-\beta V^\pi(s')},$$
$$V^\star(s) = \max_a Q(s,a), \quad Q^\star(s,a) = r(s,a) - \gamma\beta^{-1} \log \mathbb{E}_{s' \sim P_{s,a}} e^{-\beta V^\star(s')}.$$

For notational simplicity, we define the Bellman operator on the Q-functions $\mathcal{T}_Q : \mathbb{R}^{|\mathcal{S}| \times |\mathcal{A}|} \to \mathbb{R}^{|\mathcal{S}| \times |\mathcal{A}|}$ as:

$$[\mathcal{T}_Q Q](s,a) := r(s,a) - \gamma\beta^{-1} \log \mathbb{E}_{s' \sim P(\cdot|s,a)} e^{-\beta \max_{a'} Q(s',a')}. \quad (15)$$

It is not hard to verify from the above arguments that the optimal Q function $Q^\star$ satisfies

$$Q^\star = \mathcal{T}_Q Q^\star.$$

**Offline robust reinforcement learning.** The remainder of the paper focuses on finding the optimal robust policy $\pi^\star$ for the soft robust MDP problem (14). Specifically, we explore offline robust reinforcement learning algorithms which use a pre-collected dataset $\mathcal{D}$ to learn $\pi^\star$. The dataset is typically generated under the nominal model $\{P_{s,a}\}_{s \in \mathcal{S}, a \in \mathcal{A}}$, such that $\mathcal{D} = \{s_i, a_i, r_i, s'_i\}_{i=1}^N$, where the state-action pairs $(s_i, a_i) \sim \mu$ are drawn from a specific data-generating distribution $\mu$.

**Definition 1** (Robustly Admissible Distributions). *A distribution $\nu \in \Delta^{|\mathcal{S}| \times |\mathcal{A}|}$ is robustly admissible if there exists $h \geq 0$ and a policy $\pi$ and transition probability $\widehat{P} \in \{P' : \mathrm{KL}(P'_{t;s,a} || P_{s,a}) \leq \beta\}$ (both can be non-stationary) such that $\nu(s,a) = \mathrm{Pr}(s_h, a_h | s_0 \sim \rho, \pi, \widehat{P})$.*

**Assumption 2** (Concentrability). *The data-generating distribution $\mu$ satisfies concentrability if there exists a constant $C$ such that for any $\nu$ that is robustly admissible, $\max_{s,a} \frac{\nu(s,a)}{\mu(s,a)} \leq C$.*

**Remark 5.** *The notion of robustly admissible distribution and concentrability are adapted from the the corresponding notions defined for the standard MDP setting [13], where they also demonstrate the necessity of this assumption for standard RL with function approximation. It would be an interesting open question whether Assumption 2 is also necessary for robust RL settings. Recent works for standard offline RL also show that by considering variations of the RL algorithms (e.g. exploring pessimism [95] or the primal-dual formulation [102]), the concentrability assumption can be weakened to single-policy concentrability. Another interesting future direction is to study whether applying similar approaches for the soft RMDP would result in the same improvement.*

### 5.1 ROBUST FITTED-Z ITERATION (RFZI)

The offline robust MDP learning method we propose is Robust fitted-Z iteration (RFZI). The main idea is to utilize the fix point equation $Q^\star = \mathcal{T}_Q Q^\star$ with the Bellman operator (15) from the corresponding equivalent risk-sensitive MDP. However, $\mathcal{T}_Q$ involves a term $\log \mathbb{E}_{s' \sim P_{s,a}}$ which is hard to approximate with empirical estimation. Thus, instead of directly solving $Q^*$ using $Q^\star = \mathcal{T}_Q Q^\star$, we introduce an auxiliary variable, Z-function and solve a fix point equation for $Z$, which play an important role in our algorithm design and theoretical analysis.

**The Z-functions.** For a given Q-function $Q : \mathcal{S} \times \mathcal{A} \to \mathbb{R}$, we define its corresponding Z-function as below:

$$Z(s,a) := \mathbb{E}_{s' \sim P_{s,a}} e^{-\beta \max_{a'} Q(s',a')}.$$

One can establish the relationship between the Z-function and the Q-function by

$$[\mathcal{T}_Q Q](s,a) = r(s,a) - \gamma \beta^{-1} \log Z(s,a).$$

Further, we also define the Z-Bellman operator on Z-functions as:

$$[\mathcal{T}_Z Z](s,a) := \mathbb{E}_{s' \sim P_{s,a}} e^{-\beta \max_{a'} (r(s',a') - \gamma \beta^{-1} \log Z(s',a'))}.$$

Then $\mathcal{T}_Q[\mathcal{T}_Q Q]](s,a) = r(s,a) - \gamma \beta^{-1} \log[\mathcal{T}_Z Z](s,a)$. Thus, instead of solving $Q^\star = \widetilde{\mathcal{T}}_Q Q^\star$, an alternative approach is to solve $Z^\star = \widetilde{\mathcal{T}}_Z Z^\star$ and recover $Q^\star$ by $Q^\star = r - \gamma \beta^{-1} \log Z^\star$. This is the key intuition of our RFZI algorithm. Note that compare with $\mathcal{T}_Q$, $\mathcal{T}_Z$ eliminates the log dependency on the expectation term $\mathbb{E}_{s' \sim P_{s,a}}$, which makes it easier for empirical estimation.

**Function approximation and projected Z-Bellman operator.** Given that $\widetilde{\mathcal{T}}_Z$ is a contraction mapping, the solution $Z^\star$ can be obtained by running $Z_{k+1} = \widetilde{\mathcal{T}}_Z Z_k$, $\lim_{k \to +\infty} Z_k = Z^\star$. However, when the problem considered is of large state space, it is computationally very expensive to compute the Bellman operator $\widetilde{\mathcal{T}}_Z$ exactly. Thus function approximation might be needed to solve the problem approximately. Given a function class $\mathcal{F}$, we define the projected Z-Bellman operator as:

$$[\mathcal{T}_{Z,\mathcal{F}} Z](s,a) := \operatorname{argmin}_{Z' \in \mathcal{F}} \|Z' - \mathcal{T}_Z Z\|_{2,\mu}^2.$$

One can verify that $\mathcal{T}_{Z,\mathcal{F}} Z$ is also the minimizer of the following loss function $\mathcal{L}$,

$$\mathcal{L}(Z', Z) := \mathbb{E}_{s,a \sim \mu} \mathbb{E}_{s' \sim P_{s,a}} \left( Z'(s,a) - \exp\left(-\beta \max_{a'}(r(s',a') - \gamma \beta^{-1} \log Z(s',a'))\right) \right)^2,$$

i.e., $\mathcal{T}_{Z,\mathcal{F}} Z = \operatorname{argmin}_{Z' \in \mathcal{F}} \mathcal{L}(Z', Z)$.

We make the following assumptions on the expressive power of the function class $\mathcal{F}$

**Assumption 3** (Approximate Completeness). $\sup_{Z \in \mathcal{F}} \inf_{Z' \in \mathcal{F}} \|Z' - \mathcal{T}_Z Z\|_{2,\mu} \leq \epsilon_c$.

**Assumption 4** (Positivity). $e^{-\frac{\beta}{1-\gamma}} \leq Z \leq 1, \ \forall Z \in \mathcal{F}$.

**Approximate the projected Z-Bellman operator with empirical loss minimization.** The computation of the loss function $\mathcal{L}$ requires knowledge of the empirical model $P_{s,a}$ which the algorithm doesn't have access to. Thus, we introduce the following empirical loss to further approximate the loss function $\mathcal{L}$. Given an offline data set $\{(s_i, a_i, s_i')\}_{i=1}^N$ generated from the distribution $(s_i, a_i) \sim \mu$, $s_i' \sim P_{s_i, a_i}$, we can define the empirical loss $\widehat{\mathcal{L}}$ as:

$$\widehat{\mathcal{L}}(Z', Z) := \frac{1}{N} \sum_{i=1}^N \left( Z'(s_i, a_i) - \exp\left( -\beta \max_{a'} (r(s_i', a') - \gamma\beta^{-1} \log Z(s_i', a')) \right) \right)^2$$

Given the empirical loss $\widehat{\mathcal{L}}$ the empirical projected Bellman operator is defined as $\widehat{\mathcal{T}}_{Z,\mathcal{F}} Z := \operatorname{argmin}_{Z' \in \mathcal{F}} \widehat{\mathcal{L}}(Z', Z)$. Our robust fitted Z iteration (RFZI) is essentially updating the Z-functions iteratively by $Z_{k+1} = \widehat{\mathcal{T}}_{Z,\mathcal{F}} Z_k$. The detailed algorithm is displayed in Algorithm 1.

---

**Algorithm 1** Robust Fitted Z Iteration (RFZI)

---

1: **Input:** Offline dataset $\mathcal{D} = (s_i, a_i, r_i, s_i')_{i=1}^N$, function class $\mathcal{F}$.
2: **Initialize:** $Z_0 = 1 \in \mathcal{F}$
3: **for** $k = 0, \ldots, K-1$ **do**
4:     Update $Z_{k+1} = \operatorname{argmin}_{Z \in \mathcal{F}} \widehat{\mathcal{L}}(Z, Z_k)$.
5: **end for**
6: **Output:** $\pi_K = \operatorname{argmax}_a r(s, a) - \gamma\beta^{-1} \log Z_K(s, a)$

---

## 5.2 SAMPLE COMPLEXITY

This section provides the theoretical guarantee for the convergence of the RFZI algorithm. Due to space limit, we defer the proof sketches as well as detailed proofs to Appendix H.

**Theorem 5** (Sample complexity for RFZI). *Suppose Assumption 2,3 and 4 hold, then for any $\delta \in (0, 1)$, with probability at least $1 - \delta$, the policy $\pi_K$ obtained from RFQI algorithm (Algorithm 1) satisfies:*

$$\mathbb{E}_{s_0 \sim \rho} V^\star(s_0) - V^{\pi_K}(s_0) \leq \frac{2\gamma^K}{(1-\gamma)^2} + \gamma\beta^{-1} e^{\frac{\beta}{1-\gamma}} \frac{2C}{(1-\gamma)^2} \left( 4\sqrt{\frac{2\log(|\mathcal{F}|)}{N}} + 5\sqrt{\frac{2\log(8/\delta)}{N}} + \epsilon_c \right).$$

The performance gap in Theorem 5 consists of three parts. The first part $\frac{2\gamma^K}{(1-\gamma)^2}$ captures the effect of $\gamma$-contraction of the Bellman operators. The second term, which is the term with $\gamma\beta^{-1} e^{\frac{\beta}{1-\gamma}} \frac{2C}{(1-\gamma)^2} \epsilon_c$, is related to the approximation error caused by using function approximation. The third term $\gamma\beta^{-1} e^{\frac{\beta}{1-\gamma}} \frac{2C}{(1-\gamma)^2} \left( 4\sqrt{\frac{2\log(|\mathcal{F}|)}{N}} + 5\sqrt{\frac{2\log(8/\delta)}{N}} \right)$ is caused by the error of replacing the projected Z-Bellman operator with its empirical version.

**Remark 6** (Comparison and Discussions). *Under similar Bellman completeness and concentrability assumptions, the sample complexity for risk-neutral offline RL [13] is $\widetilde{O}\left( C \frac{\log |\mathcal{F}|}{(1-\gamma)^4 \epsilon^2} \right)$, while our result gives $O\left( C^2 \left( \beta^{-1} e^{\frac{\beta}{1-\gamma}} \right)^2 \frac{\log |\mathcal{F}|}{(1-\gamma)^4 \epsilon^2} \right)$ (assuming $\epsilon_c = 0$). As a consequence of robustness, our bound has a worse dependency on the concentrability factor $C$ and an additional factor $\left( \beta^{-1} e^{\frac{\beta}{1-\gamma}} \right)^2$. Note that the term $\beta^{-1} e^{\frac{\beta}{1-\gamma}}$ first decreases and then increases with $\beta$ as it goes from 0 to $+\infty$, suggesting that the hyperparameter $\beta$ also affects the learning difficulty of the problem. The choice of $\beta$ should not be either too large or too small, ideally on the same scale with $1 - \gamma$. It is still unclear to us whether the exponential dependency $e^{\frac{\beta}{1-\gamma}}$ is a proof artifact or intrinsic in our setting, however, there are results under similar settings that suggest this exponential dependency on parameter $\beta$ and the effective length $\frac{1}{1-\gamma}$ is fundamental (e.g. Theorem 3 in [26]).*

*We also compare our performance bound with the RFQI algorithm [64] which considers a similar offline learning setting and obtains sample complexity $O\left( \frac{\log(|\mathcal{F}||\mathcal{G}|)}{(\beta\epsilon)^2 (1-\gamma)^6} \right)$, where $\beta$ in their setting is the radius of the uncertainty set. Note that both results share the same dependency on $\epsilon$ and the concentrability constant $C$. However, the bound in [64] includes an additional term on the size of the dual variable space $\log |\mathcal{G}|$, whereas we have the exponential dependence term $e^{\frac{2\beta}{1-\gamma}}$.*

## 6 CONCLUSIONS AND DISCUSSIONS

This paper proposes a new formulation of risk-sensitive MDP and establishes its equivalence with the soft robust MDP. This equivalence enables us to develop the policy gradient theorem and prove the global convergence of the exact policy gradient method under direct parameterization. Additionally, for the KL-soft robust MDP (or equivalently the risk-sensitive MDP with entropy risk measure) scenario, we propose a sample-based offline learning algorithm, namely the robust fitted-Z iteration (RFZI), and analyze its sample complexity.

Our work admittedly has its limitations. Currently, our policy gradient result is limited to the exact gradient case, and further research is needed to extend it to approximate gradients. The RFZI algorithm is specifically designed for KL-soft problems and may be more suitable for small action spaces. Our future work will focus on developing practical algorithms that can handle large or even continuous state and action spaces, as well as generalizing the approach to accommodate different penalty functions.

ACKNOWLEDGEMENT

This work was funded by NSF AI institute: 2112085, NSF ECCS: 2328241.

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
