# A    MORE RELATED WORKS

**Relationship between risk and robustness:**    Apart from the works mentioned in the Introduction, there are also other works that discuss the relationship between risk and robustness under different settings or utility functions [59, 60, 25, 19]. In particular, Eysenbach and Levine [25] do not establish exact equivalence, but rather an equivalence of a lower-bound of certain robust objective; Noorani and Baras [59, 60] considers exponential utility which is a different risk measure with the risk-sensitive MDP considered in the paper as well as the Markov risk measure.

**Policy Gradient for Risk-Sensitive/Robust Learning:**    There are many previous works focusing on applying policy gradient-based algorithm for risk-sensitive/robust learning [84, 86, 87, 39, 101, 42, 16, 89, 91, 47, 50]. However, most of the works lack theoretical guarantees on the global convergence of these gradient-based algorithms [84, 86, 87, 42, 16]. There are some recent studies focusing on the global convergence guarantees for the policy gradient theorem. In particular, the works by Huang et al. [39] and Yu and Ying [101] are most related to our work. Huang et al. [39] give negative results showing that Markov risk measures are not gradient-dominant. Yu and Ying [101] prove global convergence for one specific type of risk measure - expected conditional risk measures (ECRMs). There are also works that considers policy gradient for RMDP with different types of uncertainty sets, for example, Wang and Zou [91] considers RMDPs with a particular R-contamination ambiguity set Kumar et al. [47], Li et al. [50], Wang et al. [89] considers policy gradient for RMDP with different uncertainty sets (e.g. s-rectangularity and non-rectangularity uncertainty set), which is out of the scope of discussion of this paper.

**Offline learning for robust RL:**    More and more attention is drawn to the offline learning of robust MDP. In the tabular setting, Zhou et al. [105] examined the uncertainty set defined by the KL divergence for offline data with uniformly lower bounded data visitation distribution. Shi and Chi [77] and Li et al. [49] provide near-optimal sample complexity bound for offline RL with weaker data coverage assumptions. There are also works focusing on offline learning with large state space, for example, Ma et al. [54] considers offline RL with linear approximation to deal with large state space. Our offline algorithm is most related to the work by Panaganti et al. [64], where they propose the fitted-Q iteration where the Q function as well as certain dual variables are approximated by possibly nonlinear functions such as neural networks.

**Sample complexity for robust RL:**    Apart from the offline setting, there are a number of works focusing on finite-sample performance guarantees of robust RL algorithms under different data-generating mechanisms. For example, Panaganti and Kalathil [63], Yang et al. [99], Shi et al. [78] developed sample complexities for a model-based robust RL algorithm with a variety of uncertainty sets where the data are collected using a generative model. In the online learning setting, Wang and Zou [90] proposed a robust Q-learning algorithm with an R-contamination uncertain set which achieves a similar bound as its non-robust counterpart. Badrinath and Kalathil [4] proposed a model-free algorithm with linear function approximation to cope with large state spaces.

**Other related works**    In addition to advances in the learning for RMDPs, there are also works focusing on the planning and computational facets for these problems [37, 6, 35]. Furthermore, research efforts have extended beyond theoretical considerations (e.g. [48, 66, 20, 55, 97, 103, 23]) where robust RL algorithms are proposed to handle more complicated practical problems.

A variety of different approaches have been proposed to model and address risk and robustness. Notably, in addition to Markov risk measures and robust MDPs, researchers have explored alternative methodologies, such as the use of exponential utility [38, 26, 27, 28, 58, 61], constraint MDPs [92, 15, 36, 33], distributional RL[7, 79, 53, 81, 82], and robust control [44, 104] etc.

We would also like to note that the risk-sensitive MDP with entropy risk measure (which we have proved to be equivalent to the KL-soft RMDP) is related but not identical to the exponential utility [38, 26, 27, 28, 58, 61]. There are works that discuss the equivalence of optimizing the exponential utility and solving the KL-soft RMDP in the finite-horizon undiscounted-sum setting [62]. However, the result cannot be generalized to the infinite-horizon-discounted-sum setting which is considered in this paper. Under this setting, it is unclear whether exponential utility still obtains a similar interpretation in terms of robustness.

## B    NUMERICAL SIMULATIONS

In this section, we present simulation results that evaluate the exact policy gradient algorithm (see Section 3) and the RFZI algorithm (see Section 5.1) in the following environment.

**Environment setups: traveling on a cycle graph.**    This is a environment with finite state and action space consisting of $n$ states $\mathcal{S} = \mathbb{Z}_n$, which can be conceptually regarded as arranged on a cycle. The initial distribution is a uniformly random distribution over $\mathcal{S}$. At each state the agent is allowed to select one action from $\mathcal{A} = \{-1, 0, 1\}$ (corresponding to `left`, `stay` and `right`), and receives a reward $r(s)$ that only depends on $s$ ($r(\cdot)$ is called the hitting reward function). As the aliases suggest, an action $a$ at state $s$ is supposed to move the agent to $(s + a) \mod n$. The uncertainty in the environment appears in the form of stochastic transitions, and is characterized by the probability $\alpha \in [0, \frac{1}{2}]$ of missing the expected destination by one step; i.e., transition probability is

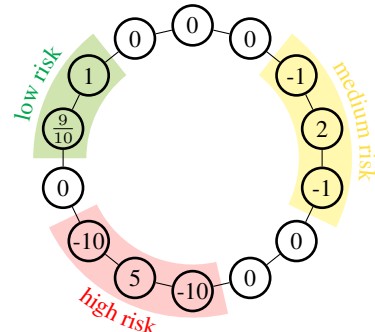

Figure 1: An exemplary 14-state environment with high-, medium-, and low-risk zones.

$$P(s' \mid s, a) = \begin{cases} \alpha & s' = (s + a \pm 1) \mod n \\ 1 - 2\alpha & s' = (s + a) \mod n \\ 0 & \text{otherwise} \end{cases}.$$

In this section, we focus on solving the KL-soft robust MDP problem (14) in this environment with varying penalty magnitude $\beta$.

**Metrics.**    The performance of the algorithms may be evaluated by the following metrics:

- *Optimality gap* $\mathbb{E}_{s_0 \sim \rho}[V^\star(s_0) - V^\pi(s_0)]$, where $\pi$ is the policy generated by the algorithm.

- *Average test reward* over a few test episodes (20 by default).

- *Robustness value* $\widehat{V}_\pi(\delta) := \inf_{P \in \mathcal{P}_\delta} \mathbb{E}^{\pi, P} \left[ \sum_{t=0}^H \gamma^t r(s_t, a_t) \,\middle|\, s_0 \sim \rho \right]$, where the model uncertainty set is selected as $\mathcal{P}_\delta := \{\widetilde{P} \mid \mathrm{KL}(\widetilde{P}_{s,a} || P_{s,a}) \leq \delta, \forall s, a\}$.

Note that the optimality gap and average test reward are usually plotted along the training trajectory, while the robustness value is usually plotted against the perturbation magnitude $\delta$.

**Exact policy gradient.**    We first examine the performance of the exact policy gradient algorithm under direct parameterization. Here we consider a 14-state environment as illustrated in Figure 1, where the rewards are marked in the nodes representing states. For this specific example, we can roughly classify the states into high-, medium-, and low-risk zones, as suggested in the figure.

For clarity of exposition, for now we focus on three specific settings, i.e. $(\alpha, \beta) = (0.01, 0.1)$, $(0.01, 1.0)$, $(0.15, 1.0)$ (more results can be found in Section B.1). The optimality gap curves under these settings are shown in Figure 2 below. It can be observed that the optimality gap decays to exactly 0 in all these settings, which justifies Theorem 4 that guarantees convergence of the exact policy gradient algorithm under direct parameterization. We also point out that the loss curve is a little crooked because we use projected gradient ascent, so that the optimization dynamics is not smooth when the policy at each state is pushed to the boundary.

Further, we take a closer look at the learned policies in these settings. The policies are illustrated in Figure 3. To understand the role of $\beta$, we compare Figure 3a with 3b — when the uncertainty in the model (represented by $\alpha$) is fixed and mild, the optimal policy for $\beta = 0.1$ is to greedily pursue the largest possible reward (i.e., staying at the state in high-risk zone with hitting reward 5); however, an agent with higher risk-sensitivity level $\beta = 1$ cares more about the potential losses caused by model uncertainty, and thus its optimal policy shifts to seeking safer options (i.e., moving to the medium-risk zone and staying there).

Similarly, to understand the role of $\alpha$, we compare Figure 3b with 3c — for an agent with fixed moderate penalty magnitude $\beta$, when the noise in the model is small ($\alpha = 0.01$), it is still optimal

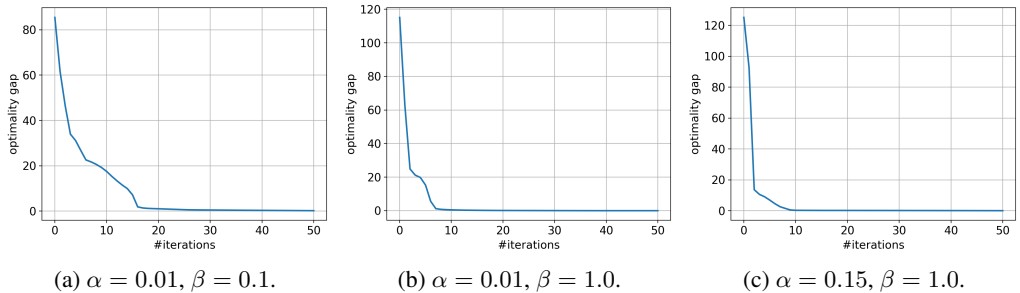

(a) $\alpha = 0.01, \beta = 0.1$.      (b) $\alpha = 0.01, \beta = 1.0$.      (c) $\alpha = 0.15, \beta = 1.0$.

Figure 2: Optimality gap curves for the exact policy gradient algorithm in different settings.

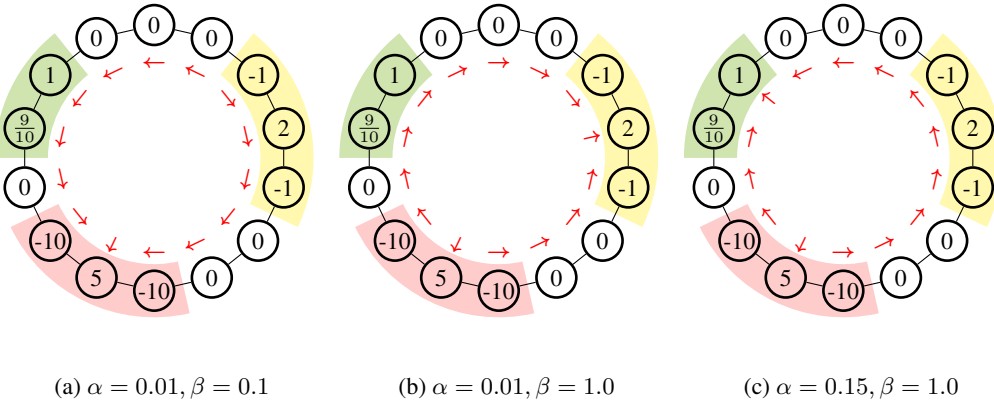

(a) $\alpha = 0.01, \beta = 0.1$      (b) $\alpha = 0.01, \beta = 1.0$      (c) $\alpha = 0.15, \beta = 1.0$

Figure 3: Illustrated policies learned by the exact policy gradient algorithm in different settings.

for the agent to stay in the medium-risk zone for a balance of risks and rewards; nevertheless, if the agent is put in a noisier environment ($\alpha = 0.15$), then its optimal policy would be directly moving to the low-risk zone to avoid potential risks.

To further examine the robustness of the risk-sensitive policies generated by the exact policy gradient algorithm, we calculate their robustness values with respect to different perturbation magnitudes $\delta$, and plot them for comparison in Figure 4. Here the *risk-neutral* policy refers to the optimal policy of the standard risk-neutral MDP, while the *robust baseline* policy refers to the optimal policy of the RMDP with KL-rectangular ambiguity set $\mathcal{P}_\delta$ (as defined above). It can be observed that, when $\delta$ is large, the risk-sensitive policies outperform the risk-neutral policy in both settings. Meanwhile, our algorithm generally exhibits a comparable level of robustness as compared to the robust baseline that directly optimizes over RMDPs; sometimes it is even more robust than the baseline, especially when the actual ambiguity set is significantly larger than the one assumed in training (see Figure 4a for the curve where $\delta \gg 0.3$). Moreover, policies generated from higher penalty magnitude $\beta$ tend to have lower robustness values when $\delta$ is small, but gradually become more robust as $\delta$ increases.

The above discussion reveals that risk-sensitive agents in face of the transition uncertainty do learn to avoid those states that could bring small instant rewards at the risk of potential future losses, and their risk-averse tendency increases when $\beta$ is set larger. These numerical evidence, in turn, further motivates and justifies our focus on KL-soft RMDPs. On the one hand, learning in the context of KL-soft RMDPs are generally more computationally tractable by relatively straightforward algorithm designs, without introducing complicated optimization techniques or additional assumptions. On the other hand, the optimal policy learned in the KL-soft RMDP context does exhibit robustness in the presence of model uncertainty. These two observations, in combination, show that the research into KL-soft RMDPs may offer an alternative, analytically tractable, and potentially more accessible approach to robust reinforcement learning while maintaining comparable robust behavior.

**RFZI.** Now we proceed to examine the performance of the RFZI algorithm. The algorithm is tested in an 100-state environment. The hitting reward design of this environment is conceptually similar to

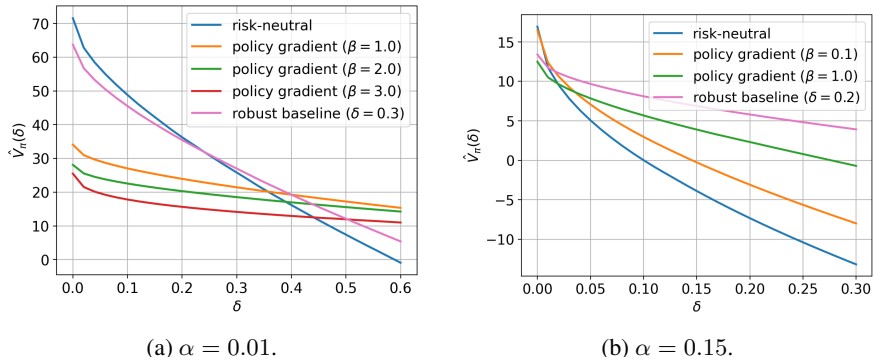

(a) $\alpha = 0.01$.  (b) $\alpha = 0.15$.

Figure 4: Robustness values of the generated policies with respect to different $\delta$.
[†]*Risk-neutral* policy refers to the optimal policy of the risk-neutral MDP.
*Robust baseline* refers to the optimal policy of the RMDP with KL-rectangular ambiguity set $\mathcal{P}_\delta$.

the exemplary 14-state environment (see our code for more details). For practical implementation, the function family is selected as a 3-layer neural network, which may take any proper state-action representation as input (details deferred to Section B.1). Note that it is crucial to select good representations for efficient reinforcement learning, as the state itself might not provide sufficient information for learning (see e.g. [24] for more discussions). Further, since the minimizer in each iteration cannot be exactly calculated, instead we simply perform a batch of stochastic gradient descent updates for approximation, where for each update we sample a subset from the offline dataset. The practical algorithm is shown in Algorithm 2 below.

---

**Algorithm 2** The practical RFZI algorithm

---

1: **Input:** Offline dataset $\mathcal{D} = (s_i, a_i, r_i, s'_i)_{i=1}^N$, function family $\mathcal{F} = \{Z(\cdot; \theta) \mid \theta\}$, learning rate $\eta$, update rate $\tau$, number of batches $T_{\text{batch}}$, batch size $N_{\text{batch}}$.
2: **Initialize:** $\theta_{\text{current}}, \theta_{\text{target}}$.
3: **for** $k = 0, \ldots, K-1$ **do**
4:    **for** $t = 1, 2, \ldots, T_{\text{batch}}$ **do**
5:        Sample a batch of transitions $\{(s_i, a_i, r_i, s'_i) \mid i \in [N_{\text{batch}}]\}$ from the dataset $\mathcal{D}$.
6:        Perform gradient descent $\theta_{\text{current}} \leftarrow \theta_{\text{current}} - \eta \nabla \widehat{\mathcal{L}}(\theta_{\text{current}})$, where

$$\widehat{\mathcal{L}}(\theta_{\text{current}}) := \widehat{\mathcal{L}}(Z(\cdot; \theta_{\text{current}}), Z(\cdot; \theta_{\text{target}})).$$

7:    **end for**
8:    Update: $\theta_{\text{target}} \leftarrow (1 - \tau)\theta_{\text{target}} + \tau\theta_{\text{current}}$, $\theta_{\text{current}} \leftarrow \theta_{\text{target}}$.
9: **end for**
10: **Output:** $\pi_K = \text{argmax}_a[r(s, a) - \gamma\beta^{-1}\log Z(s, a; \theta_{\text{target}})]$.

---

Simulation results for different penalty magnitudes are shown below in Figure 5. It can be observed that in both cases the optimality gap decays to close to 0 over time, and the average test reward also converges to oscillating around a stable value. To further examine the robustness of the policies, we compare their robustness values against the risk-neutral policy (i.e., the optimal policy of the standard risk-neutral MDP). The robustness value curve suggests that the performance of our RFZI policy is more robust than the risk-neutral policy in face of model uncertainty, and the advantage increases with larger penalty magnitude $\beta$.

However, we would also like to point out some limitations of the practical RFZI algorithm. Firstly, the training dynamics becomes unstable with larger $\beta$, reflected by slower convergence of the gradient descent updates in each iteration. Additionally, the training of the network is sensitive to other hyperparameters including learning rate, number of batches and batch size, which have to be carefully tuned for satisfactory performance. It remains future work to design algorithms that are more robust and more stable with regard to the choice of the hyperparameters.

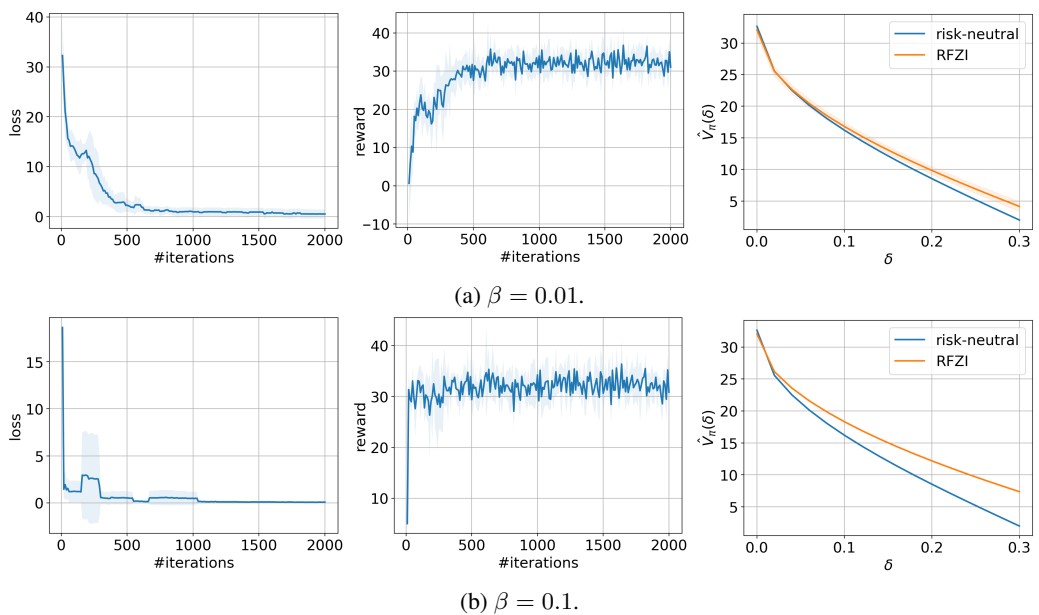

(a) $\beta = 0.01$.

(b) $\beta = 0.1$.

Figure 5: Simulation results for practical RFZI in the 100-state environment. (**Left:** optimality gap; **Middle:** average test reward; **Right:** robustness value.)

### B.1 MORE NUMERICAL DETAILS

Code for reproducing the simulation results can be found at `https://github.com/huyangsh/risk-sensitive-RL_ICRL-2024`.

**Exact policy gradient.** Here we show the optimality gap (Figure 6) and the policies (Table 1) of a full range of experiments as specified therein. It can be verified that all converged policies are exactly the optimal policies obtained by solving the Bellman optimality equation for $Q^\star$. A closer look at the policies reveals a pattern that is similar to what we have observed before — agents learn to move to lower-risk zones when the uncertainty in the environment (i.e., $\alpha$) is higher or when its penalty magnitude (i.e., $\beta$) is higher.

Table 1: Policies found by the exact policy gradient algorithm.

| $\alpha$ | $\beta$ | #steps ($\eta = 0.1$) | policy[†] |
|---|---|---|---|
| | 0.1 | 226 | [-1, -1, 1, 1, 1, 1, 1, 1, 0, -1, -1, -1, -1, -1] |
| 0.01 | 1.0 | 67 | [1, 1, 1, 0, -1, -1, -1, -1, 0, 1, 1, 1, 1, 1] |
| | 2.0 | 76 | [-1, -1, 1, 0, -1, -1, -1, -1, 0, 1, 1, 1, 0, -1] |
| | 3.0 | 192 | [-1, -1, -1, 0, -1, -1, -1, -1, 0, 1, 1, 1, 0, -1] |
| | 0.01 | 114 | [1, 1, 1, 0, -1, -1, -1, -1, 0, 1, 1, 1, 0, 1] |
| 0.15 | 0.1 | 109 | [1, 1, 1, 0, -1, -1, -1, -1, 0, 1, 1, 1, 0, -1] |
| | 1.0 | 234 | [-1, -1, -1, -1, -1, -1, -1, -1, 0, 1, 1, 1, 0, -1] |

[†] Deterministic policies are represented by a vector in $\mathcal{A}^n$, where an entry of the vector represents the action taken at the corresponding state.

**RFZI.** In the implementation, we use sinusoidal embedding of states, i.e.

$$\phi(s) = \left[\sin\frac{2\pi}{N}, \sin\frac{4\pi}{N}, \ldots, \sin\frac{2N\pi}{N}, \cos\frac{2\pi}{N}, \cos\frac{4\pi}{N}, \ldots, \cos\frac{2N\pi}{N}\right],$$

which is similar to the embedding used in [67]. The Z-functions are approximated by a 3-layer network with a 256-dimensional first hidden layer and a 32-dimensional second hidden layer (both fully-connected and activated by ReLU). The output of the network is normalized by a sigmoid function to clamp the output in $(0, 1)$ (in accordance with Assumption 4).

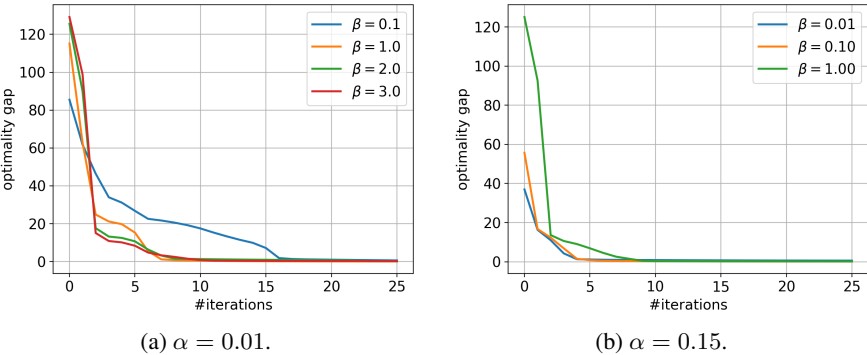

Figure 6: Optimality gap curves for the exact policy gradient algorithm in different settings.

**Training details.** Training is performed on a workstation equipped with a 32-core CPU (Intel® Xeon Platinum 8358, 2.60 GHz) and a NVIDIA® A100 GPU. The average training time for a typical RFZI training scheme of 2000 iterations and 500 batches is about 5 hours.

## C RELATIONSHIP AND DIFFERENCE WITH MARKOV RISK MEASURES

In this section, we compare our definition of risk-sensitive MDP with the Markov risk measure. Intuitively speaking, the Markov risk measure also takes the risk generated by the randomness of the policy into account whereas our definition treats the randomness of the policy in a risk-neutral manner and only considers risk from the uncertainty of the transition probability. This intuitively explains why the two notions are equivalent for deterministic policies but not for stochastic policies. For clearness, we compare with the definition considered in [39, 86, 87], where the reward $r(s)$ is only dependent on the state $s$ but not on action $a$. The Markov risk measure for policy $\pi$ is defined as :

$$V_{\mathrm{MRM}}^\pi(s_0) = r(s_0) - \gamma\sigma\left(P_{s_0}^\pi, r(s_1) - \gamma\sigma\left(P_{s_1}^\pi, r(s_2) - \gamma\sigma\left(P_{s_2}^\pi, r(s_3) - \cdots\right)\right)\right),$$

where $P_s^\pi$ is the transition probability defined by $P_s^\pi(s') = \sum_a \pi(a|s)P_{s,a}(s')$. Thus, if we define the Markov-risk-measure-Bellman operator $\widetilde{\mathcal{T}}_{\mathrm{MRM}}^\pi : \mathbb{R}^{|\mathcal{S}|\to\mathbb{R}^{|\mathcal{S}|}}$ as:

$$[\widetilde{\mathcal{T}}_{\mathrm{MRM}}^\pi V](s) := r(s) - \gamma\sigma(P_s^\pi, V) = r(s) - \gamma\sigma(\sum_a \pi(a|s)P_{s,a}, V)$$

then the Markov risk measure is the fixed point of the Bellman operator, i.e. $V_{\mathrm{MRM}}^\pi = \widetilde{\mathcal{T}}_{\mathrm{MRM}}^\pi V_{\mathrm{MRM}}^\pi$.

In contrast, the value function $\widetilde{V}^\pi$ of the risk-sensitive MDP (7) is the fixed point of the following risk-sensitive Bellman operator:

$$[\widetilde{\mathcal{T}}^\pi V](s) := r(s) - \gamma\sum_a \pi(a|s)\sigma(P_{s,a}, V)$$

Note that the risk-sensitive Bellman operator $\widetilde{\mathcal{T}}^\pi$ is linear with respect to the policy $\pi$, whereas $\widetilde{\mathcal{T}}_{\mathrm{MRM}}^\pi$ can be potentially nonlinear w.r.t. $\pi$ as $\sigma$ is generally a nonlinear function. Thus for stochastic policies, $V_{\mathrm{MRM}}^\pi$ and $\widetilde{V}^\pi$ are not equivalent. However, it is not hard to verify that when $\pi$ is a deterministic policy, $\widetilde{\mathcal{T}}^\pi$ and $\widetilde{\mathcal{T}}_{\mathrm{MRM}}^\pi$ are the same

$$[\widetilde{\mathcal{T}}_{\mathrm{MRM}}^\pi V](s) = r(s) - \gamma\sigma(P_{s,\pi(s)}, V) = [\widetilde{\mathcal{T}}^\pi V](s).$$

Thus, for deterministic policies, the value function $\widetilde{V}^\pi$ and the Markov risk measure $V_{\mathrm{MRM}}^\pi$ are the same. Additionally, when the risk-measure $\sigma$ is mixture quasiconcave (c.f. [17]), it can be shown that the optimal policy $\pi$ for the Markov risk measure can also be chosen as a deterministic policy; thus under this case the Markov risk measure and the risk-sensitive MDP obtain the same optimal value, i.e. $V_{\mathrm{MRM}}^\star = \widetilde{V}^\star$. However, we would also like to emphasize that when adding policy regularization or that the risk measure $\sigma$ is not mixture quasiconcave (e.g. mean semi-deviation), the optimal policy might no longer be a deterministic policy. In this setting, the optimal policy and optimal value of the Markov risk measure and the risk-sensitive MDP might not be the same.

## D    PROOF OF LEMMA 1

Given a Markovian policy $\pi$ and a function on the state space $\mathcal{S}$, define the Bellman operator as:

$$[\widetilde{\mathcal{T}}^\pi V](s) := \sum_a \pi(a|s)\left(r(s,a) - \gamma\sigma(P_{s,a}, V)\right), \tag{16}$$

Further, define the optimal Bellman operator $\widetilde{\mathcal{T}}^\star$ as:

$$[\widetilde{\mathcal{T}}^\star V](s) := \max_a \left(r(s,a) - \gamma\sigma(P_{s,a}, V)\right). \tag{17}$$

**Lemma 3.**

$$\|\widetilde{\mathcal{T}}^\pi(V' - V)\|_\infty \le \gamma\|V' - V\|_\infty, \;\; \|\widetilde{\mathcal{T}}^\star(V' - V)\|_\infty \le \gamma\|V' - V\|_\infty.$$

*Proof.*

$$
\begin{aligned}
[\widetilde{\mathcal{T}}^\pi(V' - V)](s) &= \sum_a \pi(a|s)\left(r(s,a) - \gamma\sigma(P_{s,a}, V')\right) - \sum_a \pi(a|s)\left(r(s,a) - \gamma\sigma(P_{s,a}, V)\right) \\
&= \gamma\left(\sigma(P_{s,a}, V) - \sigma(P_{s,a}, V')\right) \\
&\le \gamma\left(\sigma_{P_{s,a}}(V' - \|V' - V\|_\infty) - \sigma(P_{s,a}, V')\right) \quad \text{(monotonicity)} \\
&= \gamma\|V' - V\|_\infty \quad \text{(translation invariance)}.
\end{aligned}
$$

Using the same analysis we can also get,

$$
\begin{aligned}
[\widetilde{\mathcal{T}}^\pi(V - V')](s) &\le \gamma\|V' - V\|_\infty \\
\implies \|\widetilde{\mathcal{T}}^\pi(V' - V)\|_\infty &\le \gamma\|V' - V\|_\infty.
\end{aligned}
$$

Similarly, for $\widetilde{\mathcal{T}}^\star$,

$$
\begin{aligned}
[\widetilde{\mathcal{T}}^\star(V' - V)](s) &= \max_a\left(r(s,a) - \gamma\sigma(P_{s,a}, V')\right) - \max_a\left(r(s,a) - \gamma\sigma(P_{s,a}, V)\right) \\
&= \gamma\max_a\left(\sigma(P_{s,a}, V) - \sigma(P_{s,a}, V')\right) \quad (\max_x f(x) - \max_x g(x) \le \max_x(f(x) - g(x))) \\
&\le \gamma\max_a\left(\sigma_{P_{s,a}}(V' - \|V' - V\|_\infty) - \sigma(P_{s,a}, V')\right) \quad \text{(monotonicity)} \\
&= \gamma\|V' - V\|_\infty \quad \text{(translation invariance)}.
\end{aligned}
$$

$$\implies [\widetilde{\mathcal{T}}^\star(V - V')](s) \le \gamma\|V' - V\|_\infty.$$
$$\implies \|\widetilde{\mathcal{T}}^\star(V' - V)\|_\infty \le \gamma\|V' - V\|_\infty.$$

$\square$

*Proof of Lemma 1.* Lemma 1 is an immediate corollary of Lemma 3. Note that $\widetilde{V}^\pi$ and $\widetilde{V}^\star$ in (7) and (8) is the fixed point solution of

$$\widetilde{V}^\pi = \widetilde{\mathcal{T}}^\pi\widetilde{V}^\pi, \quad \widetilde{V}^\star = \widetilde{\mathcal{T}}^\star\widetilde{V}^\star.$$

From Lemma 9 and the contraction mapping theorem [40], the fixed point solution exists and is unique, which completes the proof.    $\square$

## E    PROOF OF THEOREM 2

### E.1    FINITE HORIZON DISCOUNTING CASE

We first define the value functions and Bellman operators for the finite horizon case. For any policy $\pi$ (doesn't necessarily need to be stationary or Markovian), define the value function as:

$$\overline{V}^\pi_{0:h}(s) := \min_{\{\widehat{P}_t\}_{t=0}^h} \mathbb{E}_{s_t,a_t\sim\pi,\widehat{P}}\sum_{t=0}^h \gamma^t\left(r(s_t,a_t) + \gamma D(\widehat{P}_{t;s_t,a_t}, P_{s_t,a_t})\right). \tag{18}$$

Define $\overline{V}^\star_{0:h}(s)$ as

$$\overline{V}^\star_{0:h}(s) := \max_\pi \min_{\{\widehat{P}_t\}_{t=0}^h} \mathbb{E}_{s_t,a_t\sim\pi,\widehat{P}}\sum_{t=0}^h \gamma^t\left(r(s_t,a_t) + \gamma D(\widehat{P}_{t;s_t,a_t}, P_{s_t,a_t})\right). \tag{19}$$

**Lemma 4.** $\overline{V}^{\star}_{0:h}$ *is given by:*

$$\overline{V}^{\star}_{0:h+1} = \widetilde{\mathcal{T}}^{\star}\overline{V}^{\star}_{0:h}, \quad (\overline{V}^{\star}_{0:-1} := 0)$$

*where* $\widetilde{\mathcal{T}}^{\star}$ *is defined as in* (17). *Further, for all state* $s$, *the max-min solution of* (19) *is given by a same set of policies and probability transitions:*

$$\pi^{\star}_{t|h}(s) = \operatorname*{argmax}_{a} \left( r(s,a) - \gamma\sigma(P_{s,a}, \overline{V}^{\star}_{0:h-t-1}) \right),$$

$$\widehat{P}^{\star}_{t|h;s,a} = \operatorname*{argmin}_{\widehat{P}} D(\widehat{P}, P_{s,a}) + \mathbb{E}_{s'\sim\widehat{P}}\overline{V}^{\star}_{0:h-t-1}(s').$$

*Proof.* We prove by induction. The statements are trivial for $h = 0$. Assume that the statements are true for $0 \le t \le h$, then for $h + 1$, from the definition of $\overline{V}^{\star}_h$'s we have that

$$\overline{V}^{\star}_{0:h+1}(s_0) = \max_{\pi_0} \max_{\{\pi_t\}_{t=1}^{h+1}} \min_{\widehat{P}_0} \min_{\{\widehat{P}_t\}} \mathbb{E}_{\pi,\widehat{P}} \sum_{t=0}^{h+1} \gamma^t \left( r(s_t, a_t) + \gamma D(\widehat{P}_{t;s_t,a_t}, P_{s_t,a_t}) \right)$$

$$\le \max_{\pi_0} \min_{\widehat{P}_0} \max_{\{\pi_t\}_{t=1}^{h+1}} \min_{\{\widehat{P}_t\}} \mathbb{E}_{\pi,\widehat{P}} \sum_{t=0}^{h+1} \gamma^t \left( r(s_t, a_t) + \gamma D(\widehat{P}_{t;s_t,a_t}, P_{s_t,a_t}) \right) \quad (\max_x \min_y f(x,y) \le \min_y \max_x f(x,y))$$

$$= \max_{\pi_0} \min_{\widehat{P}_0} \mathbb{E}_{a_0,s_1\sim\pi_0,\widehat{P}_0} \left[ r(s_0,a_0) + \gamma D(\widehat{P}_{0;s_0,a_0}, P_{s_0,a_0}) + \left[ \max_{\{\pi_t\}_{t=1}^{h+1}} \min_{\{\widehat{P}_t\}} \mathbb{E}_{\pi,\widehat{P}} \sum_{t=1}^{h+1} \gamma^t \left( r(s_t, a_t) + \gamma D(\widehat{P}_{t;s_t,a_t}, P_{s_t,a_t}) \right) \right] \right]$$

$$= \max_{\pi_0} \min_{\widehat{P}_0} \mathbb{E}_{a_0,s_1\sim\pi_0,\widehat{P}_0} \left[ r(s_0,a_0) + \gamma D(\widehat{P}_{0;s_0,a_0}, P_{s_0,a_0}) + \gamma\overline{V}^{\star}_{0:h}(s_1) \right].$$

Further, from the statement that $\pi^{\star}_{t|h}, \widehat{P}^{\star}_{t|h;s,a}$ solves the min-max problem (19), we have

$$\overline{V}^{\star}_{0:h+1}(s_0) = \max_{\pi_0} \max_{\{\pi_t\}_{t=1}^{h+1}} \min_{\widehat{P}_0} \min_{\{\widehat{P}_t\}} \mathbb{E}_{\pi,\widehat{P}} \sum_{t=0}^{h+1} \gamma^t \left( r(s_t, a_t) + \gamma D(\widehat{P}_{t;s_t,a_t}, P_{s_t,a_t}) \right)$$

$$\ge \max_{\pi_0} \min_{\widehat{P}_0} \min_{\{\widehat{P}_t\}} \mathbb{E}_{\pi_0,\{\pi^{\star}_{t|h+1}\}_{t=1}^{h+1},\widehat{P}} \sum_{t=0}^{h+1} \gamma^t \left( r(s_t, a_t) + \gamma D(\widehat{P}_{t;s_t,a_t}, P_{s_t,a_t}) \right)$$

$$= \max_{\pi_0} \min_{\widehat{P}_0} \mathbb{E}_{a_0,s_1\sim\pi_0,\widehat{P}_0} \left[ r(s_0,a_0) + \gamma D(\widehat{P}_{0;s_0,a_0}, P_{s_0,a_0}) \right.$$

$$\left. + \left[ \min_{\{\widehat{P}_t\}_{t=1}^{h+1}} \mathbb{E}_{\{\pi^{\star}_{t|h+1}\}_{t=1}^{h+1},\widehat{P}} \sum_{t=1}^{h+1} \gamma^t \left( r(s_t, a_t) + \gamma D(\widehat{P}_{t;s_t,a_t}, P_{s_t,a_t}) \right) \right] \right]$$

$$= \max_{\pi_0} \min_{\widehat{P}_0} \mathbb{E}_{a_0,s_1\sim\pi_0,\widehat{P}_0} \left[ r(s_0,a_0) + \gamma D(\widehat{P}_{0;s_0,a_0}, P_{s_0,a_0}) + \mathbb{E}_{\{\pi^{\star}_{t|h+1}\}_{t=1}^{h+1},\widehat{P}^{\star}_{t|h+1}} \sum_{t=1}^{h+1} \gamma^t \left( r(s_t, a_t) + \gamma D(\widehat{P}_{t;s_t,a_t}, P_{s_t,a_t}) \right) \right]$$

$$= \max_{\pi_0} \min_{\widehat{P}_0} \mathbb{E}_{a_0,s_1\sim\pi_0,\widehat{P}_0} \left[ r(s_0,a_0) + \gamma D(\widehat{P}_{0;s_0,a_0}, P_{s_0,a_0}) + \gamma\overline{V}^{\star}_{0:h}(s_1) \right] \quad (\pi^{\star}_{t+1|h+1} = \pi^{\star}_{t|h}, \widehat{P}^{\star}_{t+1|h+1} = \widehat{P}^{\star}_{t|h}).$$

Thus, we may conclude that

$$\overline{V}^{\star}_{0:h+1}(s_0) = \max_{a_0} \min_{\widehat{P}_0} r(s_0, a_0) + \gamma \left( D(\widehat{P}_{0;s_0,a_0}, P_{s_0,a_0}) + \mathbb{E}_{s_1\sim\widehat{P}_0}\overline{V}^{\star}_{0:h}(s_1) \right)$$

$$= \max_{a_0} r(s_0, a_0) - \gamma\sigma(P_{s_0,a_0}, \overline{V}^{\star}_{0:h}) \quad \text{(dual representation theorem)}$$

$$= \widetilde{\mathcal{T}}^{\star}\overline{V}^{\star}_{0:h},$$

and that the max-min policies and probability transitions can be taken as:

$$\pi^{\star}_{0|h+1}(s) = \operatorname*{argmax}_{a} r(s,a) - \gamma\sigma(P_{s,a}, \overline{V}^{\star}_{0:h})$$

$$\pi^{\star}_{t|h+1}(s) = \pi^{\star}_{t-1|h}(s) = \operatorname*{argmax}_{a} r(s,a) - \gamma\sigma(P_{s,a}, \overline{V}^{\star}_{0:h-t}), \quad t \ge 1$$

$$\widehat{P}^{\star}_{0|h+1;s,a} = \underset{\widehat{P}}{\operatorname{argmin}}\, D(\widehat{P}, P_{s,a}) + \mathbb{E}_{s'\sim\widehat{P}}\overline{V}^{\star}_{0:h}(s')$$

$$\widehat{P}^{\star}_{t|h+1;s,a} = \widehat{P}^{\star}_{t-1|h;s,a} = \underset{\widehat{P}}{\operatorname{argmin}}\, D(\widehat{P}, P_{s,a}) + \mathbb{E}_{s'\sim\widehat{P}}\overline{V}^{\star}_{0:h-t}(s'), \quad t \geq 1.$$

The above arguments complete the proof. $\qquad\square$

**Lemma 5.** *For any stationary and Markovian policy $\pi$, we have that*

$$\overline{V}^{\pi}_{0:h+1} = \widetilde{\mathcal{T}}^{\pi}\overline{V}^{\pi}_{0:h},$$

*where $\widetilde{\mathcal{T}}^{\pi}$ is defined as in* (16)*. Further, for all state $s$, minimal solution of* (18) *is given by the same set of probability transitions:*

$$\widehat{P}^{\pi}_{t|h;s,a} = \underset{\widehat{P}}{\operatorname{argmin}}\, D(\widehat{P}, P_{s,a}) + \mathbb{E}_{s'\sim\widehat{P}}\overline{V}^{\pi}_{0:h-t-1}(s').$$

*Proof.* Our proof is largely similar to Lemma 4 and is again by induction. The statements are trivial for $h = 0$. Assume that the statements are true for $0 \leq t \leq h$, then for $h + 1$, from the definition of $V_h^{\pi}$'s we have that

$$\overline{V}^{\pi}_{h+1}(s_0) = \min_{\widehat{P}_0}\min_{\{\widehat{P}_t\}} \mathbb{E}_{\pi,\widehat{P}} \sum_{t=0}^{h+1} \gamma^t \left( r(s_t, a_t) + \gamma D(\widehat{P}_{t;s_t,a_t}, P_{s_t,a_t}) \right)$$

$$= \min_{\widehat{P}_0}\min_{\{\widehat{P}_t\}} \mathbb{E}_{\pi_0,\widehat{P}_0} \left[ r(s_0, a_0) + \gamma D(\widehat{P}_{0;s_0,a_0}, P_{s_0,a_0}) + \mathbb{E}_{\pi_{1:h+1},\widehat{P}_{1:h+1}} \sum_{t=1}^{h+1} \gamma^t \left( r(s_t, a_t) + \gamma D(\widehat{P}_{t;s_t,a_t}, P_{s_t,a_t}) \right) \right]$$

$$= \min_{\widehat{P}_0} \mathbb{E}_{\pi_0,\widehat{P}_0} \left[ r(s_0, a_0) + \gamma D(\widehat{P}_{0;s_0,a_0}, P_{s_0,a_0}) + \min_{\{\widehat{P}_t\}} \mathbb{E}_{\pi_{1:h+1},\widehat{P}_{1:h+1}} \sum_{t=1}^{h+1} \gamma^t \left( r(s_t, a_t) + \gamma D(\widehat{P}_{t;s_t,a_t}, P_{s_t,a_t}) \right) \right]$$

$$= \min_{\widehat{P}_0} \mathbb{E}_{a_0,s_1\sim\pi_0,\widehat{P}_0} \left( r(s_0, a_0) + \gamma D(\widehat{P}_{0;s_0,a_0}, P_{s_0,a_0}) + \gamma\overline{V}^{\pi}_{0:h}(s_1) \right)$$

$$= \mathbb{E}_{a_0\sim\pi_0} \left( r(s_0, a_0) + \gamma \min_{\widehat{P}_0} \left( D(\widehat{P}_{0;s_0,a_0}, P_{s_0,a_0}) + \mathbb{E}_{s_1\sim\widehat{P}_{0;s_0,a_0}} \overline{V}^{\pi}_{0:h}(s_1) \right) \right)$$

$$= \mathbb{E}_{a_0\sim\pi_0} \left( r(s_0, a_0) - \gamma\sigma(P_{s_0,a_0}, \overline{V}^{\pi}_{0:h}) \right).$$

Here the last step follows from dual representation theorem. Further, the minimal probability transitions can be taken as:

$$\widehat{P}^{\pi}_{0|h+1;s,a} = \underset{\widehat{P}}{\operatorname{argmin}}\, D(\widehat{P}, P_{s,a}) + \mathbb{E}_{s'\sim\widehat{P}}\overline{V}^{\pi}_{0:h}(s')$$

$$\widehat{P}^{\pi}_{t|h+1;s,a} = \widehat{P}^{\pi}_{t-1|h;s,a} = \underset{\widehat{P}}{\operatorname{argmin}}\, D(\widehat{P}, P_{s,a}) + \mathbb{E}_{s'\sim\widehat{P}}\overline{V}^{\pi}_{0:h-t}(s'), \quad t \geq 1.$$

The above arguments complete the proof. $\qquad\square$

### E.2 PROOF OF THEOREM 2 (INFINITE HORIZON CASE)

*Proof.* We first verify that for $\overline{V}^{\star}, \overline{V}^{\pi}$ defined in (3), (4),

$$\lim_{h\to+\infty} \overline{V}^{\star}_{0:h} = \overline{V}^{\star}, \quad \lim_{h\to+\infty} \overline{V}^{\pi}_{0:h} = \overline{V}^{\pi}.$$

From the definition of $\overline{V}^{\pi}$, we have that

$$\overline{V}^{\pi}(s) = \inf_{\{\widehat{P}_t\}_{t\geq 0}} \mathbb{E}_{s_t,a_t\sim\pi,\widehat{P}} \left[ \sum_{t=0}^{+\infty} \gamma^t \left( r(s_t, a_t) + \gamma D(\widehat{P}_{t;s_t,a_t}, P_{s_t,a_t}) \right) \Big| s_0 = s \right]$$

$$\leq \inf_{\{\widehat{P}_t\}_{t=0}^{h}} \mathbb{E}_{s_t,a_t\sim\pi,\widehat{P}} \left[ \sum_{t=0}^{h} \gamma^t \left( r(s_t, a_t) + \gamma D(\widehat{P}_{t;s_t,a_t}, P_{s_t,a_t}) \right) \Big| s_0 = s \right]$$

$$+ \mathbb{E}_{s_t, a_t \sim \pi, P} \left[ \sum_{t=h}^{+\infty} \gamma^t \left( r(s_t, a_t) + \gamma D(P_{s_t, a_t}, P_{s_t, a_t}) \right) \Big| s_0 = s \right]$$

$$= \overline{V}_{0:h}^{\pi}(s) + \mathbb{E}_{s_t, a_t \sim \pi, P} \left[ \sum_{t=h}^{+\infty} \gamma^t \left( r(s_t, a_t) \right) \Big| s_0 = s \right]$$

$$\leq \overline{V}_{0:h}^{\pi}(s) + \frac{\gamma^h}{1 - \gamma},$$

$$\overline{V}^{\pi}(s) = \inf_{\{\widehat{P}_t\}_{t \geq 0}} \mathbb{E}_{s_t, a_t \sim \pi, \widehat{P}} \left[ \sum_{t=0}^{+\infty} \gamma^t \left( r(s_t, a_t) + \gamma D(\widehat{P}_{t; s_t, a_t}, P_{s_t, a_t}) \right) \Big| s_0 = s \right]$$

$$\geq \inf_{\{\widehat{P}_t\}_{t \geq 0}} \mathbb{E}_{s_t, a_t \sim \pi, \widehat{P}} \left[ \sum_{t=0}^{h} \gamma^t \left( r(s_t, a_t) + \gamma D(\widehat{P}_{t; s_t, a_t}, P_{s_t, a_t}) \right) \Big| s_0 = s \right]$$

$$= \inf_{\{\widehat{P}_t\}_{t=0}^{h}} \mathbb{E}_{s_t, a_t \sim \pi, \widehat{P}} \left[ \sum_{t=0}^{h} \gamma^t \left( r(s_t, a_t) + \gamma D(\widehat{P}_{t; s_t, a_t}, P_{s_t, a_t}) \right) \Big| s_0 = s \right]$$

$$= \overline{V}_{0:h}^{\pi}(s)$$

$$\implies |\overline{V}^{\pi}(s) - \overline{V}_{0:h}^{\pi}(s)| \leq \frac{\gamma^h}{1 - \gamma} \implies \lim_{h \to +\infty} \overline{V}_{0:h}^{\pi} = \overline{V}^{\pi}.$$

And similarly, for $\overline{V}^{\star}$, we have

$$\overline{V}^{\star}(s) = \sup_{\pi} \inf_{\{\widehat{P}_t\}_{t \geq 0}} \mathbb{E}_{s_t, a_t \sim \pi, \widehat{P}} \left[ \sum_{t=0}^{+\infty} \gamma^t \left( r(s_t, a_t) + \gamma D(\widehat{P}_{t; s_t, a_t}, P_{s_t, a_t}) \right) \Big| s_0 = s \right].$$

$$\leq \sup_{\pi} \left[ \inf_{\{\widehat{P}_t\}_{t=0}^{h}} \mathbb{E}_{s_t, a_t \sim \pi, \widehat{P}} \left[ \sum_{t=0}^{h} \gamma^t \left( r(s_t, a_t) + \gamma D(\widehat{P}_{t; s_t, a_t}, P_{s_t, a_t}) \right) \Big| s_0 = s \right] \right.$$

$$\left. + \mathbb{E}_{s_t, a_t \sim \pi, P} \left[ \sum_{t=h}^{+\infty} \gamma^t \left( r(s_t, a_t) + \gamma D(P_{s_t, a_t}, P_{s_t, a_t}) \right) \Big| s_0 = s \right] \right]$$

$$\leq \sup_{\pi} \inf_{\{\widehat{P}_t\}_{t=0}^{h}} \mathbb{E}_{s_t, a_t \sim \pi, \widehat{P}} \left[ \sum_{t=0}^{h} \gamma^t \left( r(s_t, a_t) + \gamma D(\widehat{P}_{t; s_t, a_t}, P_{s_t, a_t}) \right) \Big| s_0 = s \right]$$

$$+ \sup_{\pi} \mathbb{E}_{s_t, a_t \sim \pi, P} \left[ \sum_{t=h}^{+\infty} \gamma^t \left( r(s_t, a_t) + \gamma D(P_{s_t, a_t}, P_{s_t, a_t}) \right) \Big| s_0 = s \right]$$

$$= \overline{V}_{0:h}^{\pi}(s) + \sup_{\pi} \mathbb{E}_{s_t, a_t \sim \pi, P} \left[ \sum_{t=h}^{+\infty} \gamma^t \left( r(s_t, a_t) + \gamma D(P_{s_t, a_t}, P_{s_t, a_t}) \right) \Big| s_0 = s \right]$$

$$\leq \overline{V}_{0:h}^{\pi}(s) + \frac{\gamma^h}{1 - \gamma},$$

$$\overline{V}^{\star}(s) = \sup_{\pi} \inf_{\{\widehat{P}_t\}_{t \geq 0}} \mathbb{E}_{s_t, a_t \sim \pi, \widehat{P}} \left[ \sum_{t=0}^{+\infty} \gamma^t \left( r(s_t, a_t) + \gamma D(\widehat{P}_{t; s_t, a_t}, P_{s_t, a_t}) \right) \Big| s_0 = s \right].$$

$$\geq \sup_{\pi} \left[ \inf_{\{\widehat{P}_t\}_{t=0}^{h}} \mathbb{E}_{s_t, a_t \sim \pi, \widehat{P}} \left[ \sum_{t=0}^{h} \gamma^t \left( r(s_t, a_t) + \gamma D(\widehat{P}_{t; s_t, a_t}, P_{s_t, a_t}) \right) \Big| s_0 = s \right] \right]$$

$$= V_{0:h}^{\star}(s)$$

$$\implies |\overline{V}^{\star}(s) - \overline{V}_{0:h}^{\star}(s)| \leq \frac{\gamma^h}{1 - \gamma} \implies \lim_{h \to +\infty} \overline{V}_{0:h}^{\star} = \overline{V}^{\star}.$$

Then from Lemma 4 and Lemma 5 we have that

$$\overline{V}_{0:h+1}^{\star} = \widetilde{\mathcal{T}}^{\star} \overline{V}_{0:h}^{\star}, \quad \overline{V}_{0:h+1}^{\pi} = \widetilde{\mathcal{T}}^{\pi} \overline{V}_{0:h}^{\pi}.$$

Since $\widetilde{\mathcal{T}}^\star, \widetilde{\mathcal{T}}^\pi$ is a continuous mapping, taking the limit on both sides of the equations we get

$$\overline{V}^\star = \widetilde{\mathcal{T}}^\star \overline{V}^\star, \overline{V}^\pi = \widetilde{\mathcal{T}}^\pi \overline{V}^\pi,$$

i.e., (7) and (4) obtains the same solution $\overline{V}^\pi$ and (8) and (3) obtains the same solution $\overline{V}^\star$.

Next, we will show that the claim that the minimal solution of (4) is given by (11). Since $\overline{V}^\pi$ satisfies $\overline{V}^\pi = \widetilde{\mathcal{T}}^\pi \overline{V}^\pi$, we have

$$\overline{V}^\pi(s_0) = \mathbb{E}_{a_0 \sim \pi(\cdot|s_0)} \left( r(s_0, a_0) + \gamma \min_{\widehat{P}_0} D(\widehat{P}_{0;s_0,a_0}, P_{s_0,a_0}) + \mathbb{E}_{s_1 \sim \widehat{P}_{0;s_0,a_0}} \overline{V}^\pi(s_1) \right)$$

$$= \mathbb{E}_{a_0 \sim \pi(\cdot|s_0), s_1 \sim \widehat{P}^\pi_{s_0,a_0}} \left( r(s_0, a_0) + \gamma D(\widehat{P}^\pi_{s_0,a_0}, P_{s_0,a_0}) + \gamma \overline{V}^\pi(s_1) \right).$$

Apply this equation iteratively, we get

$$\overline{V}^\pi(s_0) = \mathbb{E}_{a_t \sim \pi(\cdot|s_t), s_{t+1} \sim \widehat{P}^\pi_{s_t,a_t}} \left( r(s_0, a_0) + \gamma D(\widehat{P}^\pi_{s_0,a_0}, P_{s_0,a_0}) + \gamma r(s_1, a_1) + \gamma^2 D(\widehat{P}^\pi_{s_1,a_1}, P_{s_1,a_1}) + \gamma^2 \overline{V}^\pi(s_2) \right)$$

$$= ...$$

$$= \mathbb{E}_{a_t \sim \pi(\cdot|s_t), s_{t+1} \sim \widehat{P}^\pi_{s_t,a_t}} \sum_{t=0}^{+\infty} \gamma^t \left( r(s_t, a_t) + \gamma D(\widehat{P}^\pi_{s_t,a_t}, P_{s_t,a_t}) \right),$$

which concludes that the minimal solution is given by $\widehat{P}^\pi$ defined in (11).

For $\overline{V}^\star$. We aim to show that $\overline{V}^\star = \overline{V}^{\pi^\star}$. From the definition of $\pi^\star$ and the fact that $\overline{V}^\star = \widetilde{\mathcal{T}}^\star \overline{V}^\star$,

$$\overline{V}^\star(s_0) = \max_{a_0} r(s_0, a_0) - \gamma \sigma(P_{s_0,a_0}, \overline{V}^\star)$$

$$\overset{(10)}{=} \mathbb{E}_{a_0 \sim \pi^\star(\cdot|s_0)} \left( r(s_0, a_0) - \gamma \sigma(P_{s_0,a_0}, \overline{V}^\star) \right)$$

$$= \mathbb{E}_{a_0 \sim \pi^\star(\cdot|s_0)} \left( r(s_0, a_0) + \gamma \min_{\widehat{P}_0} \left( D(\widehat{P}_{0;s_0,a_0}, P_{s_0,a_0}) + \mathbb{E}_{s_1 \sim \widehat{P}_{0;s_0,a_0}} \overline{V}^\star(s_1) \right) \right)$$

$$= \min_{\widehat{P}_0} \mathbb{E}_{a_0 \sim \pi^\star(\cdot|s_0)} \left( r(s_0, a_0) + \gamma \left( D(\widehat{P}_{0;s_0,a_0}, P_{s_0,a_0}) + \mathbb{E}_{s_1 \sim \widehat{P}_{0;s_0,a_0}} \overline{V}^\star(s_1) \right) \right).$$

Apply the above equation iteratively we get

$$\overline{V}^\star(s_0) = \min_{\{\widehat{P}_t\}_{t=1}^{+\infty}} \mathbb{E}_{a_t \sim \pi^\star(\cdot|s_t), s_{t+1} \sim \widehat{P}_{t;s_t,a_t}} \sum_{t=0}^{+\infty} \gamma^t \left( r(s_t, a_t) + \gamma D(\widehat{P}^\pi_{s_t,a_t}, P_{s_t,a_t}) \right)$$

$$= \overline{V}^{\pi^\star}(s_0),$$

which implies that the optimal value function can be obtained by the stationary policy $\pi^\star$. Thus, the minimal transition probability is given by

$$\widehat{P}^\star_{s,a} = \widehat{P}^{\pi^\star}_{s,a} = \operatorname*{argmin}_{\widehat{P}} D(\widehat{P}, P_{s,a}) + \mathbb{E}_{s' \sim \widehat{P}} \overline{V}^\star(s'),$$

which completes the proof. $\qquad \square$

## F    PROOF OF THEOREM 3

*Proof of Theorem 3.* We first prove the differentiability of $V^\theta$ with respect to $\theta$ by the implicit function theorem [52]. We define the $|\mathcal{S}|$-dimensional multivariate function $F(\theta, V)$ as follows:

$$[F(\theta, V)](s) = V(s) - \sum_a \pi_\theta(a|s)(r(s, a) - \gamma \sigma(P_{s,a}, V)).$$

From the definition of the value function for Markov risk measures, $V^\theta$ is given by the following implicit function:

$$F(\theta, V^\theta) = 0.$$

Thus, from the implicit function theorem, to prove the differentiability of $V^\theta$ with respect to $\theta$, it suffices to prove that the Jacobian matrix

$$J_{F,V}(\theta, V^\theta) = \left[ \frac{\partial F_s}{\partial V_{s'}} \Big|_{V=V^\theta} \right]_{s,s' \in \mathcal{S}}$$

is invertible. Here $F_s$ denotes the $s$-th entry of $F$ and $V_{s'}$ the $s'$-th entry of $V$. From Lemma 6,

$$J_{F,V}(\theta, V^\theta) = I - \gamma \widehat{P}_\mathcal{S}^\theta,$$

where $\widehat{P}_\mathcal{S}^\theta$ is a stochastic matrix. Thus, $\|\gamma \widehat{P}_\mathcal{S}^\theta\|_\infty \leq \gamma < 1$, which implies that $I - \gamma \widehat{P}_\mathcal{S}^\theta$ is invertible, thus from the implicit function theorem $V^\theta$ is differentiable w.r.t. $\theta$.

Given the differentiability, what is left is to calculate the gradient. We can further use implicit function theorem to compute the gradient, yet another easier way is through the following algebraic manipulation:

$$V^\theta(s_0) = \sum_{a_0} \pi_\theta(a_0|s_0) \left( r(s_0, a_0) - \gamma \sigma(P_{s_0,a_0}, V^\theta) \right)$$

$$\implies \nabla_\theta V^\theta(s_0) = \sum_{a_0} \nabla_\theta \pi_\theta(a_0|s_0) \left( r(s_0, a_0) - \gamma \sigma(P_{s_0,a_0}, V^\theta) \right)$$

$$- \gamma \sum_{a_0} \pi_\theta(a_0|s_0) \sum_{s_1} \frac{\partial \sigma(P_{s_0,a_0}, \cdot)}{\partial V_{s_1}} \Big|_{V=V^\theta} \nabla_\theta V^\theta(s_1)$$

$$= \sum_{a_0} \pi_\theta(a_0|s_0) Q^\theta(s_0, a_0) \nabla_\theta \log \pi_\theta(a_0|s_0)$$

$$+ \gamma \sum_{a_0} \pi_\theta(a_0|s_0) \sum_{s_1} \widehat{P}_{s_0,a_0}^\theta(s_1) \nabla_\theta V^\theta(s_1) \quad \text{(Lemma 6)}$$

$$= \mathbb{E}_{a_0 \sim \pi_\theta(\cdot|s_0)} Q^\theta(s_0, a_0) \nabla_\theta \log \pi_\theta(a_0|s_0) + \gamma \mathbb{E}_{s_1 \sim \widehat{P}_{s_0,a_0}^\theta} \nabla_\theta V^\theta(s_1)$$

Applying the above equation iteratively we get:

$$V^\theta(s_0) = \mathbb{E}_{a_0 \sim \pi_\theta(\cdot|s_0)} Q^\theta(s_0, a_0) \nabla_\theta \log \pi_\theta(a_0|s_0) + \gamma \mathbb{E}_{s_1 \sim \widehat{P}_{s_0,a_0}^\theta} \nabla_\theta V^\theta(s_1)$$

$$= \mathbb{E}_{a_t \sim \pi_\theta(\cdot|s_t), s_{t+1} \sim \widehat{P}_{s_t,a_t}^\theta, t=0,1} Q^\theta(s_0, a_0) \nabla_\theta \log \pi_\theta(a_0|s_0) + \gamma Q^\theta(s_1, a_1) \nabla_\theta \log \pi_\theta(a_1|s_1) + \gamma^2 \nabla_\theta V^\theta(s_2)$$

$$= \cdots$$

$$= \mathbb{E}_{a_t \sim \pi_\theta(\cdot|s_t), s_{t+1} \sim \widehat{P}_{s_t,a_t}^\theta} \sum_{t=1}^{+\infty} \gamma^t Q^\theta(s_t, a_t) \nabla_\theta \log \pi_\theta(a_t|s_t),$$

which completes the proof. $\qquad\square$

**Lemma 6.**

$$\frac{\partial \sigma(P_{s,a}, \cdot)}{\partial V_{s'}} \Big|_{V=V^\theta} = -\widehat{P}_{s,a}^\theta(s'),$$

*which implies that*

$$J_{F,V}(\theta, V^\theta) = I - \gamma \widehat{P}_\mathcal{S}^\theta,$$

*where $\widehat{P}_\mathcal{S}^\theta$ is a stochastic matrix given by*

$$[\widehat{P}_\mathcal{S}^\theta]_{s,s'} = \widehat{P}_\mathcal{S}^\theta(s'|s) = \sum_a \pi_\theta(a|s) \widehat{P}_{s,a}^\theta(s').$$

*Proof.* From the definition of $\widehat{P}_{s,a}^\theta(s')$ (11) and the dual representation theorem (Theorem 1) we have

$$\sigma(P_{s,a}, V^\theta) = - \left( \min_{\widehat{P}} \mathbb{E}_{s' \sim \widehat{P}} V^\theta(s') + D(\widehat{P}, P_{s,a}) \right) = - \mathbb{E}_{s' \sim \widehat{P}_{s,a}^\theta} V^\theta(s') - D(\widehat{P}_{s,a}^\theta, P_{s,a})$$

$$\implies D(\widehat{P}_{s,a}^{\theta}, P_{s,a}) = -\sigma(P_{s,a}, V^{\theta}) - \mathbb{E}_{s' \sim \widehat{P}_{s,a}^{\theta}} V^{\theta}(s').$$

From the definition of $D$:

$$D(\widehat{P}_{s,a}^{\theta}, P_{s,a}) = \sup_{V} -\sigma(P_{s,a}, V) - \mathbb{E}_{s' \sim \widehat{P}_{s,a}^{\theta}} V(s')$$

$$\implies V^{\theta} = \underset{V}{\operatorname{argmax}} -\sigma(P_{s,a}, V) - \mathbb{E}_{s' \sim \widehat{P}_{s,a}^{\theta}} V(s'),$$

thus

$$\nabla_V \left( \sigma(P_{s,a}, V) + \mathbb{E}_{s' \sim \widehat{P}_{s,a}^{\theta}} V(s') \right) \Big|_{V=V^{\theta}} = 0$$

$$\implies \frac{\partial \sigma(P_{s,a}, \cdot)}{\partial V_{s'}} \Big|_{V=V^{\theta}} = -\frac{\partial \mathbb{E}_{s'' \sim \widehat{P}_{s,a}^{\theta}} V(s'')}{\partial V_{s'}} \Big|_{V=V^{\theta}} = -\widehat{P}_{s,a}^{\theta}(s').$$

Then we have

$$\frac{\partial F_s}{\partial V_{s'}} \Big|_{V=V^{\theta}} = \frac{\partial \left( V(s) - \sum_a \pi_\theta(a|s)(r(s,a) - \gamma\sigma(P_{s,a}, V)) \right)}{\partial V_{s'}}$$

$$= \mathbf{1}\{s' = s\} + \gamma \sum_a \pi_\theta(a|s) \frac{\partial \sigma(P_{s,a}, \cdot)}{\partial V_{s'}} \Big|_{V=V^{\theta}}$$

$$= \mathbf{1}\{s' = s\} - \gamma \sum_a \pi_\theta(a|s) \widehat{P}_{s,a}^{\theta}(s')$$

$$= \mathbf{1}\{s' = s\} - \gamma [\widehat{P}_{\mathcal{S}}^{\theta}]_{s,s'},$$

which completes the proof. $\qquad\square$

## G    PROOF OF LEMMA 2 AND THEOREM 4

Before proving Lemma 2 and Theorem 4, we first introduce the performance difference lemma for soft RMDPs, which will play an important role in the following proofs. The lemma adopts from the performance difference lemma for risk-neutral MDPs (c.f. [43])

**Lemma 7** (Performance Difference Lemma for soft RMDPs). *Given stationary policies $\pi', \pi$, we have that*

$$\mathbb{E}_{s_t, a_t \sim \pi', \widehat{P}^\pi} \sum_{t=0}^{+\infty} \gamma^t \left( r(s_t, a_t) + \gamma D(\widehat{P}_{t;s_t,a_t}^\pi, P_{s_t,a_t}) \right) - \mathbb{E}_{s_t, a_t \sim \pi, \widehat{P}^\pi} \sum_{t=0}^{+\infty} \gamma^t \left( r(s_t, a_t) + \gamma D(\widehat{P}_{t;s_t,a_t}^\pi, P_{s_t,a_t}) \right)$$

$$= \frac{1}{1-\gamma} \sum_{s,a} d^{\pi^\star, \widehat{P}^\pi}(s)(\pi'(a|s) - \pi(a|s)) Q^\pi(s,a)$$

*Proof.* For notational simplicity in this proof we also define the value function $V^{\pi, \widehat{P}}$ and Q-function $Q^{\pi, \widehat{P}}$ for a given policy $\pi$ under a given probability transition $\widehat{P}$ as follows:

$$V^{\pi, \widehat{P}}(s) := \mathbb{E}_{s_t, a_t \sim \pi, \widehat{P}} \sum_{t=0}^{+\infty} \left[ \gamma^t \left( r(s_t, a_t) + \gamma D(\widehat{P}_{t;s_t,a_t}, P_{s_t,a_t}) \right) | s_0 = s \right],$$

$$Q^{\pi, \widehat{P}}(s) := \mathbb{E}_{s_t, a_t \sim \pi, \widehat{P}} \sum_{t=0}^{+\infty} \left[ \gamma^t \left( r(s_t, a_t) + \gamma D(\widehat{P}_{t;s_t,a_t}, P_{s_t,a_t}) \right) | s_0 = s, a_0 = a \right]$$

Also from Theorem 2 we know that $V^{\pi, \widehat{P}^\pi} = V^\pi, Q^{\pi, \widehat{P}^\pi} = Q^\pi$. Then we only need to show that

$$V^{\pi', \widehat{P}^\pi}(s) - V^{\pi, \widehat{P}^\pi}(s) = \frac{1}{1-\gamma} \sum_{s,a} d^{\pi^\star, \widehat{P}^\pi}(s)(\pi'(a|s) - \pi(a|s)) Q^\pi(s,a).$$

The left hand side of the equation can be decomposed as

$$\mathbb{E}_{s_t, a_t \sim \pi', \widehat{P}^\pi} \sum_{t=0}^{+\infty} \gamma^t \left( r(s_t, a_t) + \gamma D(\widehat{P}_{t;s_t,a_t}^\pi, P_{s_t,a_t}) \right) - \mathbb{E}_{s_t, a_t \sim \pi, \widehat{P}^\pi} \sum_{t=0}^{+\infty} \gamma^t \left( r(s_t, a_t) + \gamma D(\widehat{P}_{t;s_t,a_t}^\pi, P_{s_t,a_t}) \right)$$

$$= \underbrace{\mathbb{E}_{s_t, a_t \sim \pi', \widehat{P}^\pi} \sum_{t=0}^{+\infty} \gamma^t \Big( r(s_t, a_t) + \gamma D(\widehat{P}^\pi_{t; s_t, a_t}, P_{s_t, a_t}) \Big) - \mathbb{E}_{a_0 \sim \pi', s_t, a_t \sim \pi, \widehat{P}^\pi, t \geq 1} \sum_{t=0}^{+\infty} \gamma^t \Big( r(s_t, a_t) + \gamma D(\widehat{P}^\pi_{t; s_t, a_t}, P_{s_t, a_t}) \Big)}_{\text{Part A}}$$

$$+ \underbrace{\mathbb{E}_{a_0 \sim \pi', s_t, a_t \sim \pi, \widehat{P}^\pi, t \geq 1} \sum_{t=0}^{+\infty} \gamma^t \Big( r(s_t, a_t) + \gamma D(\widehat{P}^\pi_{t; s_t, a_t}, P_{s_t, a_t}) \Big) - \mathbb{E}_{s_t, a_t \sim \pi, \widehat{P}^\pi} \sum_{t=0}^{+\infty} \gamma^t \Big( r(s_t, a_t) + \gamma D(\widehat{P}^\pi_{t; s_t, a_t}, P_{s_t, a_t}) \Big)}_{\text{Part B}}.$$

Note that

Part A
$$= \mathbb{E}_{a_0 \sim \pi'} \Bigg( \mathbb{E}_{s_t, a_t \sim \pi', \widehat{P}^\pi, t \geq 1} \sum_{t=0}^{+\infty} \gamma^t \Big( r(s_t, a_t) + \gamma D(\widehat{P}^\pi_{t; s_t, a_t}, P_{s_t, a_t}) \Big)$$
$$- \mathbb{E}_{s_t, a_t \sim \pi, \widehat{P}^\pi, t \geq 1} \sum_{t=0}^{+\infty} \gamma^t \Big( r(s_t, a_t) + \gamma D(\widehat{P}^\pi_{t; s_t, a_t}, P_{s_t, a_t}) \Big) \Bigg)$$
$$= \gamma \mathbb{E}_{a_0 \sim \pi', s_1 \sim \widehat{P}^\pi} (V^{\pi', \widehat{P}^\pi}(s_1) - V^{\pi', \widehat{P}^\pi}(s_1))$$

and

$$\text{Part B} = \mathbb{E}_{a_0 \sim \pi'} Q^{\pi, \widehat{P}^\pi}(s_0, a_0) - \mathbb{E}_{a_0 \sim \pi} Q^{\pi, \widehat{P}^\pi}(s_0, a_0)$$
$$= \sum_{a_0} (\pi'(a_0|s_0) - \pi(a_0|s_0)) Q^{\pi, \widehat{P}^\pi}(s_0, a_0).$$

Thus we get

$$V^{\pi', \widehat{P}^\pi}(s_0) - V^{\pi, \widehat{P}^\pi}(s_0)$$
$$= \gamma \mathbb{E}_{a_0 \sim \pi', s_1 \sim \widehat{P}^\pi} (V^{\pi', \widehat{P}^\pi}(s_1) - V^{\pi', \widehat{P}^\pi}(s_1)) + \sum_{a_0} (\pi'(a_0|s_0) - \pi(a_0|s_0)) Q^{\pi, \widehat{P}^\pi}(s_0, a_0).$$

Applying this equality iteratively we get

$$V^{\pi', \widehat{P}^\pi}(s_0) - V^{\pi, \widehat{P}^\pi}(s_0) = \sum_{t=0}^{\infty} \gamma^t \mathbb{E}_{a_\tau, s_\tau \sim \pi', \widehat{P}^\pi} \sum_{a_t} (\pi'(a_t|s_t) - \pi(a_t|s_t)) Q^{\pi, \widehat{P}^\pi}(s_t, a_t)$$
$$= \frac{1}{1 - \gamma} \sum_{s, a} d^{\pi^\star, \widehat{P}^\pi}(s) (\pi'(a|s) - \pi(a|s)) Q^\pi(s, a),$$

which completes the proof. $\qquad\square$

*Proof of Lemma 2.* From Theorem 2, we have

$$\mathbb{E}_{s_0 \sim \rho} V^\star(s_0) = \min_{\{\widehat{P}_t\}_{t=0}^h} \mathbb{E}_{s_t, a_t \sim \pi^\star, \widehat{P}^\theta} \sum_{t=0}^{+\infty} \gamma^t \Big( r(s_t, a_t) + \gamma D(\widehat{P}^\theta_{t; s_t, a_t}, P_{s_t, a_t}) \Big)$$
$$\leq \mathbb{E}_{s_t, a_t \sim \pi^\star, \widehat{P}^\theta} \sum_{t=0}^{+\infty} \gamma^t \Big( r(s_t, a_t) + \gamma D(\widehat{P}^\theta_{t; s_t, a_t}, P_{s_t, a_t}) \Big)$$
$$\mathbb{E}_{s_0 \sim \rho} V^\theta(s_0) = \mathbb{E}_{s_t, a_t \sim \pi_\theta, \widehat{P}^\theta} \sum_{t=0}^{+\infty} \gamma^t \Big( r(s_t, a_t) + \gamma D(\widehat{P}^\theta_{t; s_t, a_t}, P_{s_t, a_t}) \Big)$$
.

Thus

$$\mathbb{E}_{s_0 \sim \rho} V^\star(s_0) - V^\theta(s_0)$$
$$\leq \mathbb{E}_{s_t, a_t \sim \pi^\star, \widehat{P}^\theta} \sum_{t=0}^{+\infty} \gamma^t \Big( r(s_t, a_t) + \gamma D(\widehat{P}^\theta_{t; s_t, a_t}, P_{s_t, a_t}) \Big)$$

$$- \mathbb{E}_{s_t,a_t \sim \pi_\theta, \widehat{P}^\theta} \sum_{t=0}^{+\infty} \gamma^t \left( r(s_t, a_t) + \gamma D(\widehat{P}^\theta_{t;s_t,a_t}, P_{s_t,a_t}) \right)$$

$$= \frac{1}{1-\gamma} \sum_{s,a} d^{\pi^\star, \widehat{P}^\theta}(s)(\pi^\star(a|s) - \pi_\theta(a|s))Q^\theta(s,a) \quad \text{(by Lemma 7)}$$

$$\leq \frac{1}{1-\gamma} \sum_{s,a} d^{\pi^\star, \widehat{P}^\theta}(s) \max_{\overline{\pi}}(\overline{\pi}(a|s) - \pi_\theta(a|s))Q^\theta(s,a) \quad (20)$$

$$\leq \frac{1}{1-\gamma} \left\| \frac{d^{\pi^\star, \widehat{P}^\theta}}{d^{\pi_\theta, \widehat{P}^\theta}} \right\|_\infty \sum_{s,a} d^{\pi_\theta, \widehat{P}^\theta}(s) \max_{\overline{\pi}}(\overline{\pi}(a|s) - \pi_\theta(a|s))Q^\theta(s,a)$$

$$= \left\| \frac{d^{\pi^\star, \widehat{P}^\theta}}{d^{\pi_\theta, \widehat{P}^\theta}} \right\|_\infty \max_{\overline{\pi}} \left\langle \overline{\pi} - \pi_\theta, \mathbb{E}_{s_0 \sim \rho} \nabla_\theta V^\theta(s_0) \right\rangle$$

$$\square$$

*Proof of Theorem 4.* For notational simplicity, we define use $\pi_s := \pi(\cdot|s)$ to denote the $|\mathcal{A}|$-dimensional probability distribution. We also use the abbreviation $Q^{(k)}, \widehat{P}^{(k)}$ to denote $Q^{\pi^{(k)}}, \widehat{P}^{\pi^{(k)}}$. We also define the following variable that will be useful throughout the proof.

$$G^{(k)}_\eta := \frac{1}{\eta} \left( \pi_s^{(k+1)} - \pi_s^{(k)} \right)$$

Similar to the proof of Lemma 2, we have

$$\mathbb{E}_{s_0 \sim \rho} V^{(k)}(s_0) = \min_{\{\widehat{P}_t\}_{t=0}^h} \mathbb{E}_{s_t,a_t \sim \pi^{(k)}, \widehat{P}} \sum_{t=0}^{+\infty} \gamma^t \left( r(s_t, a_t) + \gamma D(\widehat{P}_{t;s_t,a_t}, P_{s_t,a_t}) \right)$$

$$\leq \mathbb{E}_{s_t,a_t \sim \pi^{(k)}, \widehat{P}^{(k+1)}} \sum_{t=0}^{+\infty} \gamma^t \left( r(s_t, a_t) + \gamma D(\widehat{P}^{(k+1)}_{t;s_t,a_t}, P_{s_t,a_t}) \right)$$

$$\mathbb{E}_{s_0 \sim \rho} V^{(k+1)}(s_0) = \mathbb{E}_{s_t,a_t \sim \pi^{(k+1)}, \widehat{P}^{(k+1)}} \sum_{t=0}^{+\infty} \gamma^t \left( r(s_t, a_t) + \gamma D(\widehat{P}^{(k+1)}_{t;s_t,a_t}, P_{s_t,a_t}) \right).$$

Thus

$$\mathbb{E}_{s_0 \sim \rho} V^{(k+1)}(s_0) - V^{(k)}(s_0) \geq \mathbb{E}_{s_t,a_t \sim \pi^{(k+1)}, \widehat{P}^{(k+1)}} \sum_{t=0}^{+\infty} \gamma^t \left( r(s_t, a_t) + \gamma D(\widehat{P}^{(k+1)}_{t;s_t,a_t}, P_{s_t,a_t}) \right)$$

$$- \mathbb{E}_{s_t,a_t \sim \pi^{(k)}, \widehat{P}^{(k+1)}} \sum_{t=0}^{+\infty} \gamma^t \left( r(s_t, a_t) + \gamma D(\widehat{P}^{(k+1)}_{t;s_t,a_t}, P_{s_t,a_t}) \right)$$

$$= \frac{1}{1-\gamma} \sum_{s,a} d^{\pi^{(k)}, \widehat{P}^{(k+1)}}(s)(\pi^{(k+1)}(a|s) - \pi^{(k)}(a|s))Q^{(k+1)}(s,a) \quad \text{(by Lemma 7)}$$

$$= \underbrace{\frac{1}{1-\gamma} \sum_{s,a} d^{\pi^{(k)}, \widehat{P}^{(k+1)}}(s)(\pi^{(k+1)}(a|s) - \pi^{(k)}(a|s))Q^{(k)}(s,a)}_{\text{Part I}}$$

$$+ \underbrace{\frac{1}{1-\gamma} \sum_{s,a} d^{\pi^{(k)}, \widehat{P}^{(k+1)}}(s)(\pi^{(k+1)}(a|s) - \pi^{(k)}(a|s))(Q^{(k+1)}(s,a) - Q^{(k)}(s,a))}_{\text{Part II}}.$$

From Lemma 13, we have

$$\sum_a (\pi^{(k+1)}(a|s) - \pi^{(k)}(a|s))Q^{(k)}(s,a) = \langle \pi_s^{(k+1)} - \pi_s^{(k)}, Q^{(k)}(s,\cdot) \rangle$$

$$= \left\langle \text{Proj}_{\Delta^{|\mathcal{A}|}} \left( \pi_s^{(k)} + \eta \frac{1}{1-\gamma} d^{\pi^{(k)}, \widehat{P}^{(k)}}(s) Q^{(k)}(s, \cdot) \right), Q^{(k)}(s, \cdot) \right\rangle$$

$$\geq \frac{1-\gamma}{\eta d^{\pi^{(k)}, \widehat{P}^{(k)}}(s)} \|\pi_s^{(k+1)} - \pi_s^{(k)}\|_2^2.$$

Thus

$$\text{Part I} \geq \frac{1}{\eta} \sum_s \frac{d^{\pi^{(k)}, \widehat{P}^{(k+1)}}(s)}{d^{\pi^{(k)}, \widehat{P}^{(k)}}(s)} \|\pi_s^{(k+1)} - \pi_s^{(k)}\|_2^2$$

$$\geq \frac{1}{\eta M} \|\pi^{(k+1)} - \pi^{(k)}\|_2^2$$

$$= \frac{\eta}{M} \|G_\eta^{(k)}\|_2^2.$$

**Remark 7.** *Note that for standard MDP, the corresponding Part I can be bounded by $\eta \|G_\eta^{(k)}\|_2^2$ instead of $\frac{\eta}{M} \|G_\eta^{(k)}\|_2^2$. This is the key reason why the dependency on $M$ is worse for our setting.*

For Part II, we can bound it using Lemma 12

$$|\text{Part II}| \leq \frac{1}{1-\gamma} \sum_{s,a} d^{\pi^{(k)}, \widehat{P}^{(k+1)}}(s) \left| \pi^{(k+1)}(a|s) - \pi^{(k)}(a|s) \right| \left| Q^{(k+1)}(s,a) - Q^{(k)}(s,a) \right|$$

$$\leq \frac{1}{(1-\gamma)^3} \sum_{s,a} d^{\pi^{(k)}, \widehat{P}^{(k+1)}}(s) \left| \pi^{(k+1)}(a|s) - \pi^{(k)}(a|s) \right| \max_s \|\pi_s^{(k+1)} - \pi_s^{(k)}\|_1$$

$$\leq \frac{1}{(1-\gamma)^3} \left( \max_s \|\pi_s^{(k+1)} - \pi_s^{(k)}\|_1 \right)^2$$

$$\leq \frac{|\mathcal{A}|}{(1-\gamma)^3} \|\pi^{(k+1)} - \pi^{(k)}\|_2^2$$

$$= \frac{\eta^2 |\mathcal{A}|}{(1-\gamma)^3} \|G_\eta^{(k)}\|_2^2.$$

Combining the bounds on Part I and II we get, for $\eta = \frac{(1-\gamma)^3}{2|\mathcal{A}|M}$

$$\mathbb{E}_{s_0 \sim \rho} V^{(k+1)}(s_0) - V^{(k)}(s_0) \geq \left( \frac{\eta}{M} - \frac{\eta^2 |\mathcal{A}|}{(1-\gamma)^3} \right) \|G_\eta^{(k)}\|_2^2 = \frac{(1-\gamma)^3}{4|\mathcal{A}|M} \|G_\eta^{(k)}\|_2^2$$

Using the telescoping technique we get

$$\sum_{k=0}^{K-1} \|G_\eta^{(k)}\|_2^2 \leq \frac{4|\mathcal{A}|M^2}{(1-\gamma)^3} \sum_{k=0}^{K-1} \mathbb{E}_{s_0 \sim \rho}(V^{(k+1)}(s_0) - V^{(k)}(s_0))$$

$$\leq \frac{4|\mathcal{A}|M^2}{(1-\gamma)^3} \left( V^{(K)}(s_0) - V^{(0)}(s_0) \right) \leq \frac{4|\mathcal{A}|M^2}{(1-\gamma)^4} \tag{21}$$

where the last inequality uses the fact that $0 \leq V^\pi(s) \leq \mathbb{E}_{s_t, a_t \sim \pi, P}[\sum_{t=0}^{+\infty} \gamma^t r(s_t, a_t) | s_0 = s] \leq \frac{1}{1-\gamma}$.
*Claim:*

$$\mathbb{E}_{s_0 \sim \rho} V^\star(s_0) - V^{(k+1)}(s) \leq \left( M + \frac{\eta |\mathcal{A}|}{(1-\gamma)^3} \right) \|G_\eta^{(k)}\|_2$$

*Proof of Claim.* From the definition of projection, for any $\pi_s' \in \Delta^{|\mathcal{A}|}$,

$$\langle \pi_s^{(k)} + \eta \frac{1}{1-\gamma} d^{\pi^{(k)}, \widehat{P}^{(k)}}(s) Q^{(k)}(s, \cdot) - \pi_s^{(k+1)}, \pi_s' - \pi_s^{(k+1)} \rangle \leq 0$$

$$\implies \langle Q^{(k)}(s, \cdot), \pi_s' - \pi_s^{(k+1)} \rangle \leq \frac{1-\gamma}{\eta d^{\pi^{(k)}, \widehat{P}^{(k)}}(s)} \langle \pi_s^{(k+1)} - \pi_s^{(k)}, \pi_s' - \pi^{(k+1)} \rangle$$

$$\leq \frac{(1-\gamma)M}{\eta}\|\pi_s^{(k+1)} - \pi_s^{(k)}\|_2\|\pi_s' - \pi^{(k+1)}\|_2$$

$$\leq (1-\gamma)M\|G_\eta^{(k)}\|_2\|\pi_s' - \pi^{(k+1)}\|_2.$$

Thus

$$\langle Q^{(k+1)}(s,\cdot), \pi_s' - \pi_s^{(k+1)}\rangle \leq \langle Q^{(k)}(s,\cdot), \pi_s' - \pi_s^{(k+1)}\rangle + \langle Q^{(k+1)}(s,\cdot) - Q^{(k)}(s,\cdot), \pi_s' - \pi_s^{(k+1)}\rangle$$

$$\leq (1-\gamma)M\|G_\eta^{(k)}\|_2\|\pi_s' - \pi^{(k+1)}\|_2 + \|Q^{(k+1)}(s,\cdot) - Q^{(k)}(s,\cdot)\|_\infty\|\pi_s' - \pi_s^{(k+1)}\|_1$$

$$\leq (1-\gamma)M\|G_\eta^{(k)}\|_2\|\pi_s' - \pi^{(k+1)}\|_2 + \sqrt{|\mathcal{A}|}\|Q^{(k+1)}(s,\cdot) - Q^{(k)}(s,\cdot)\|_\infty\|\pi_s' - \pi_s^{(k+1)}\|_2.$$

Further, from Lemma 12,

$$\|Q^{(k+1)}(s,\cdot) - Q^{(k)}(s,\cdot)\|_\infty \leq \frac{1}{(1-\gamma)^2}\max_{\bar{s}\in\mathcal{S}}\|\pi_{\bar{s}}^{(k+1)} - \pi_{\bar{s}}^{(k)}\|_1 \leq \frac{\sqrt{|\mathcal{A}|}}{(1-\gamma)^2}\|\pi^{(k+1)} - \pi^{(k)}\|_2$$

$$= \frac{\eta\sqrt{|\mathcal{A}|}}{(1-\gamma)^2}\|G_\eta^{(k)}\|_2.$$

Thus

$$\langle Q^{(k+1)}(s,\cdot), \pi_s' - \pi_s^{(k+1)}\rangle \leq (1-\gamma)M\|G_\eta^{(k)}\|_2\|\pi_s' - \pi^{(k+1)}\|_2 + \frac{\eta|\mathcal{A}|}{(1-\gamma)^2}\|G_\eta^{(k)}\|_2\|\pi_s' - \pi^{(k+1)}\|_2$$

$$= \left((1-\gamma)M + \frac{\eta|\mathcal{A}|}{(1-\gamma)^2}\right)\|G_\eta^{(k)}\|_2\|\pi_s' - \pi^{(k+1)}\|_2$$

From the proof of Lemma 2 (inequality (20)), we have

$$\mathbb{E}_{s_0\sim\rho}V^\star(s_0) - V^{(k+1)}(s) \leq \frac{1}{1-\gamma}\sum_{s.a}d^{\pi^\star,\widehat{P}^{(k+1)}}(s)\max_{\pi'}(\pi'(a|s) - \pi^{k+1}(a|)s)Q^{(k+1)}(s,a)$$

$$= \frac{1}{1-\gamma}\sum_{s.a}d^{\pi^\star,\widehat{P}^{(k+1)}}(s)\max_{\pi'}\langle Q^{(k+1)}(s,\cdot), \pi_s' - \pi_s^{(k+1)}\rangle$$

$$\leq \frac{1}{1-\gamma}\max_s\max_{\pi'}\langle Q^{(k+1)}(s,\cdot), \pi_s' - \pi_s^{(k+1)}\rangle$$

$$\leq \frac{1}{1-\gamma}\left((1-\gamma)M + \frac{\eta|\mathcal{A}|}{(1-\gamma)^2}\right)\|G_\eta^{(k)}\|_2\|\pi_s' - \pi^{(k+1)}\|_2$$

$$= \left(M + \frac{\eta|A|}{(1-\gamma)^3}\right)\|G_\eta^{(k)}\|_2,$$

which completes the proof of the claim. $\qquad\square$

Substitute the claim into (21) we get

$$\sum_{k=1}^K\left(\mathbb{E}_{s_0\sim\rho}V^\star(s_0) - V^{(k)}(s_0)\right)^2 \leq \frac{16|\mathcal{A}|M^4}{(1-\gamma)^4},$$

and thus

$$\min_{1\leq k\leq K}\mathbb{E}_{s_0\sim\rho}V^\star(s_0) - V^{(k)}(s_0) \leq \sqrt{\frac{16|\mathcal{A}|M^4}{(1-\gamma)^4 K}}.$$

By setting

$$K \geq \frac{16|\mathcal{A}|M^4}{(1-\gamma)^4\epsilon^2},$$

we get

$$\min_{1\leq k\leq K}\mathbb{E}_{s_0\sim\rho}V^\star(s_0) - V^{(k)}(s_0) \leq \epsilon.$$

$\qquad\square$

# H  PROOF OF THEOREM 5

## H.1  PROOF SKETCHES

Before providing the full proof, in this section we first give a brief proof sketch of Theorem 5. The lemmas in the proof sketch is proved in the following sections. We define the following auxiliary variables:

$$Q_k(s,a) := r(s,a) - \gamma\beta^{-1}\log Z_k(s,a), \quad \pi_k(s) := \operatorname*{argmax}_a Q_k(s,a).$$

The proof of Theorem 5 can be decoupled into the following four steps:

**Step 1: Decomposition of the performance difference.**  In this step, we first decompose the performance difference $\mathbb{E}_{s_0\sim\rho}(V^\star - V^{\pi_k})(s_0)$ in terms of $[Q^\star - Q_k]$ and $[Q_k - Q^\star]$.

**Lemma 8.**

$$\mathbb{E}_{s_0\sim\rho}(V^\star - V^{\pi_k})(s_0) \leq$$

$$\mathbb{E}_{s_0\sim\rho,s_{\tau+1}\sim\widehat{P}^{\pi_k}_{s_\tau,a_\tau},a_\tau\sim\pi_k(s_\tau)}\sum_{t=0}^{+\infty}\gamma^t([Q^\star - Q_k](s_t,\pi^\star(s_t))) + ([Q_k - Q^\star](s_t,\pi_k(s_t))).$$

*where $\widehat{P}^{\pi_k}$ is defined as $\widehat{P}^{\pi_k}(s'|s,a) \propto P(s'|s,a)\exp(-\gamma V^{\pi_k}(s'))$.*

**Step 2: Bound $[Q^\star - Q_k]$ and $[Q_k - Q^\star]$.**  Given Lemma 8, the next step is to further upper-bound $[Q^\star - Q_k]$ and $[Q_k - Q^\star]$. We have the following lemma:

**Lemma 9.** *For any distribution $\nu$ that is admissible,*

$$\mathbb{E}_{s,a\sim\nu}(Q_k - Q^\star)(s,a), \ \mathbb{E}_{s,a\sim\nu}(Q^\star - Q_k)(s,a) \leq \frac{\gamma^k}{1-\gamma} + C\sum_{m=1}^k \gamma^{k-m}\|Q_m - \widetilde{\mathcal{T}}_Q Q_{m-1}\|_{1,\mu},$$

*where $C$ is defined in Assumption 2.*

The upper bound consists of two parts. The first part is caused by the contraction mapping of Bellman operator, and the second part captures the error of replacing the Bellman operation $\widetilde{\mathcal{T}}_Q Q_{m-1}$ with its approximation $Q_m$.

**Step 3: Bound $\|Q_k - \widetilde{\mathcal{T}}_Q Q_{k-1}\|_{1,\mu}$.**  According to Lemma 9, to bound $[Q^\star - Q_k]$ and $[Q_k - Q^\star]$, we need to have a better understanding of $\|Q_k - \widetilde{\mathcal{T}}_Q Q_{k-1}\|_{1,\mu}$.

**Lemma 10.**

$$\|Q_k - \widetilde{\mathcal{T}}_Q Q_{k-1}\|_{1,\mu} \leq \gamma\beta^{-1}e^{\frac{\beta}{1-\gamma}}\left(\|(\widehat{\mathcal{T}}_{Z,\mathcal{F}} - \mathcal{T}_{Z,\mathcal{F}})Z_{k-1}\|_{1,\mu} + \|(\mathcal{T}_{Z,\mathcal{F}} - \mathcal{T}_Z)Z_{k-1}\|_{1,\mu}\right).$$

Lemma 10 suggests that the error $\|Q_k - \widetilde{\mathcal{T}}_Q Q_{k-1}\|_{1,\mu}$ can be bounded by two parts, the first part $\|(\mathcal{T}_{Z,\mathcal{F}} - \mathcal{T}_Z)Z_{k-1}\|_{1,\mu}$ is the error caused by function approximation, i.e., replacing the Bellman operator with the project Bellman operator. The second part is the error of replacing the projected Bellman operator with the empirical projected Bellman operator.

**Step 4: Bound $\|(\widehat{\mathcal{T}}_{Z,\mathcal{F}} - \mathcal{T}_{Z,\mathcal{F}})Z_{k-1}\|_{1,\mu}$ and $\|(\mathcal{T}_{Z,\mathcal{F}} - \mathcal{T}_Z)Z_{k-1}\|_{1,\mu}$.**  The last step closes the proof by bounding $\|(\widehat{\mathcal{T}}_{Z,\mathcal{F}} - \mathcal{T}_{Z,\mathcal{F}})Z_{k-1}\|_{1,\mu}$ and $\|(\mathcal{T}_{Z,\mathcal{F}} - \mathcal{T}_Z)Z_{k-1}\|_{1,\mu}$ which shows up on the right hand side of Lemma 10.

**Lemma 11.**

$$\|(\mathcal{T}_{Z,\mathcal{F}} - \mathcal{T}_Z)Z_{k-1}\|_{2,\mu} \leq \epsilon_c.$$

*With probability at least $1-\delta$,*

$$\|(\widehat{\mathcal{T}}_{Z,\mathcal{F}} - \mathcal{T}_{Z,\mathcal{F}})Z_{k-1}\|_{2,\mu} \leq 4\sqrt{\frac{2\log(|\mathcal{F}|)}{N}} + 5\sqrt{\frac{2\log(8/\delta)}{N}}$$

Combining the four steps finishes the proof.

## H.2 PROOF OF LEMMAS IN PROOF SKETCHES

In this section we define the following operator $\mathcal{T}_{V \to Q} : \mathbb{R}^{|\mathcal{S}|} \to \mathbb{R}^{|\mathcal{S}| \times |\mathcal{A}|}$ for notational simplicity:

$$[\mathcal{T}_{V \to Q} V](s, a) = r(s, a) - \gamma \beta^{-1} \log \mathbb{E}_{s' \sim P(\cdot | s, a)} e^{-\beta V(s')} \tag{22}$$

*Proof of Lemma 8.* Define $V_k(s) := \max_a Q_k(s, a)$, then

$$(V^\star - V^{\pi_k})(s_0) = (V^\star - V_k)(s_0) + (V_k - V^{\pi_k})(s_0)$$
$$= (Q^\star(s_0, \pi^\star(s_0)) - Q_k(s_0, \pi_k(s_0))) + (Q_k(s_0, \pi_k(s_0)) - Q^{\pi_k}(s_0, \pi_k(s_0)))$$
$$\leq (Q^\star(s_0, \pi^\star(s_0)) - Q_k(s_0, \pi^\star(s_0))) + (Q_k(s_0, \pi_k(s_0)) - Q^\star(s_0, \pi_k(s_0))) + (Q^\star(s_0, \pi_k(s_0)) - Q^{\pi_k}(s_0, \pi_k(s_0)))$$
$$\leq ([Q^\star - Q_k](s_0, \pi^\star(s_0))) + ([Q_k - Q^\star](s_0, \pi_k(s_0))) + [\mathcal{T}_{V \to Q}(V^\star - V^{\pi_k})](s_0, \pi_k(s_0))$$
$$\overset{\text{Lemma 14}}{\leq} \gamma \mathbb{E}_{s_1 \sim \widehat{P}^{\pi_k}_{s_0, a_0}, a_0 \sim \pi_k(s_0)} [V^\star - V^{\pi_k}](s_1) + ([Q^\star - Q_k](s_0, \pi^\star(s_0))) + ([Q_k - Q^\star](s_0, \pi_k(s_0)))$$

Apply the above inequality recursively, and we get:

$$\mathbb{E}_{s_0 \sim \rho}(V^\star - V^{\pi_k})(s_0) \leq \mathbb{E}_{s_0 \sim \rho, s_{\tau+1} \sim \widehat{P}^{\pi_k}_{s_\tau, a_\tau}, a_\tau \sim \pi_k(s_\tau)} \sum_{t=0}^{+\infty} \gamma^t ([Q^\star - Q_k](s_t, \pi^\star(s_t))) + ([Q_k - Q^\star](s_t, \pi_k(s_t))).$$

$\square$

*Proof of Lemma 9.* We use proof by induction. The inequality naturally holds for $k = 0$. Assume that it holds for $k - 1$, then

$$\mathbb{E}_{s, a \sim \nu}(Q_k - Q^\star)(s, a) = \mathbb{E}_{s, a \sim \nu}[Q_k - \widetilde{\mathcal{T}}_Q Q_{k-1}](s, a) + \mathbb{E}_{s, a \sim \nu}[\widetilde{\mathcal{T}}_Q (Q_{k-1} - Q^\star)](s, a)$$
$$\overset{\text{Lemma 14}}{\leq} \mathbb{E}_{s, a \sim \nu}[Q_k - \widetilde{\mathcal{T}}_Q Q_{k-1}](s, a) + \gamma \mathbb{E}_{s, a \sim \nu} \mathbb{E}_{s' \sim \widehat{P}^\star_{s, a}} \max_{a'} (Q_{k-1} - Q^\star)(s', a')$$
$$= \mathbb{E}_{s, a \sim \nu}[Q_k - \widetilde{\mathcal{T}}_Q Q_{k-1}](s, a) + \gamma \mathbb{E}_{s', a' \sim \nu'}(Q_{k-1} - Q^\star)(s', a'),$$

where $\nu'$ is the marginal distribution on $(s', a')$ given the joint distribution on $(s, a, s', a')$ by $s, a \sim \nu, s' \sim \widehat{P}^\star_{s, a}, a' = \text{argmax}_{a'}(Q_{k-1} - Q^\star)(s', a')$. Since $\nu$ is admissible, $\nu'$ is also admissible. Thus from the induction assumption, we have

$$\mathbb{E}_{s, a \sim \nu}(Q_k - Q^\star)(s, a) \leq \mathbb{E}_{s, a \sim \nu}[Q_k - \widetilde{\mathcal{T}}_Q Q_{k-1}](s, a) + \gamma \left( \frac{\gamma^{k-1}}{1 - \gamma} + C \sum_{m=1}^{k-1} \gamma^{k-1-m} \|Q_m - \widetilde{\mathcal{T}}_Q Q_{m-1}\|_{1, \mu} \right)$$
$$\leq \frac{\gamma^k}{1 - \gamma} + C \sum_{m=1}^{k} \gamma^{k-m} \|Q_m - \widetilde{\mathcal{T}}_Q Q_{m-1}\|_{1, \mu}$$

$\square$

*Proof of Lemma 10.* From the definition of $Q_k$, we have

$$Q_k(s, a) = r(s, a) - \gamma \beta^{-1} \log Z_k(s, a)$$
$$\widetilde{\mathcal{T}}_Q Q_{k-1}(s, a) = r(s, a) - \gamma \beta^{-1} \log \mathbb{E}_{s' \sim P(\cdot | s, a)} e^{-\beta \max_{a'} Q_{k-1}(s', a')}$$
$$= r(s, a) - \gamma \beta^{-1} \log \mathbb{E}_{s' \sim P(\cdot | s, a)} e^{-\beta \max_{a'} (r(s', a') - \gamma \beta^{-1} \log Z_{k-1}(s', a'))}$$
$$= r(s, a) - \gamma \beta^{-1} \log([\mathcal{T}_Z Z_{k-1}](s, a)).$$
$$\implies \|Q_k - \widetilde{\mathcal{T}}_Q Q_{k-1}\|_{1, \mu} = \gamma \beta^{-1} \|\log Z_k - \log \mathcal{T}_Z Z_{k-1}\|_{1, \mu}$$
$$\overset{\text{Lemma 15}}{\leq} \frac{\gamma \beta^{-1}}{\min_{s, a} \min\{Z_k(s, a), \mathcal{T}_Z Z_{k-1}(s, a)\}} \|Z_k - \mathcal{T}_Z Z_{k-1}\|_{1, \mu}$$
$$\leq \gamma \beta^{-1} e^{\frac{\beta}{1 - \gamma}} \|\widehat{\mathcal{T}}_{Z, \mathcal{F}} Z_{k-1} - \mathcal{T}_Z Z_{k-1}\|_{1, \mu}$$
$$\leq \gamma \beta^{-1} e^{\frac{\beta}{1 - \gamma}} \left( \|(\widehat{\mathcal{T}}_{Z, \mathcal{F}} - \mathcal{T}_{Z, \mathcal{F}}) Z_{k-1}\|_{1, \mu} + \|(\mathcal{T}_{Z, \mathcal{F}} - \mathcal{T}_Z) Z_{k-1}\|_{1, \mu} \right).$$

$\square$

*Proof of Lemma 11.* The first inequality can be obtained by the approximate completeness assumption (Assumption 3).

$$\sup_{Z \in \mathcal{F}} \inf_{Z' \in \mathcal{F}} \|Z' - \mathcal{T}_Z Z\|_{2,\mu} \le \epsilon_c$$

$$\implies \inf_{Z' \in \mathcal{F}} \|Z' - \mathcal{T}_Z Z_{k-1}\|_{2,\mu} \le \epsilon_c$$

$$\implies \text{LHS} = \|\widehat{\mathcal{T}}_{Z,\mathcal{F}} Z_{k-1} - \mathcal{T}_Z Z_{k-1}\|_{2,\mu} \le \epsilon_c$$

The second inequality can be obtained by applying Lemma 3, eq(12) in [64]. Here we can set $l(Z, (s, a, s')) = \left( Z(s,a) - \exp\left(-\beta\gamma \max_{a'}(r(s', a') - \beta^{-1} \log Z_{k-1}(s', a'))\right)\right)^2$. Then it is not hard to verify that $l(Z, (s, a, s')) \le 1$, $l(Z, (s, a, s'))$ is 2-Lipschitz in $Z$ and that $|Z(s,a)| \le 1$, thus we can set $c_1 = 1, c_2 = 1, c_3 = 2$ in eq(12) in [64] and obtain that with probability at least $1 - \delta$,

$$\mathbb{E}_{s,a\sim\mu}\mathbb{E}_{s'\sim P_{s,a}} \left[ l(\widehat{\mathcal{T}}_{Z,\mathcal{F}} Z_{k-1}, (s, a, s')) - l(\mathcal{T}_{Z,\mathcal{F}} Z_{k-1}, (s, a, s')) \right] \le 4\sqrt{\frac{2\log(|\mathcal{F}|)}{N}} + 5\sqrt{\frac{2\log(8/\delta)}{N}}$$

$$\implies \mathcal{L}(\widehat{\mathcal{T}}_{Z,\mathcal{F}} Z_{k-1}, Z_{k-1}) - \mathcal{L}(\mathcal{T}_{Z,\mathcal{F}} Z_{k-1}, Z_{k-1})$$

$$= \mathbb{E}_{s,a\sim\mu} \left( \widehat{\mathcal{T}}_{Z,\mathcal{F}} Z_{k-1}(s, a) - \mathbb{E}_{s'\sim P_{s,a}} e^{-\beta\gamma \max_{a'}(r(s',a') - \beta^{-1}\log Z(s',a'))} \right)^2$$

$$= \|(\widehat{\mathcal{T}}_{Z,\mathcal{F}} - \mathcal{T}_{Z,\mathcal{F}}) Z_{k-1}\|_{2,\mu} \le 4\sqrt{\frac{2\log(|\mathcal{F}|)}{N}} + 5\sqrt{\frac{2\log(8/\delta)}{N}}$$

$\square$

## H.3 PROOF OF THEOREM 5

*Proof of Theorem 5.*

$$\mathbb{E}_{s_0\sim\rho}(V^\star - V^{\pi_K})(s_0)$$

$$\overset{\text{Lemma 8}}{=} \mathbb{E}_{s_0\sim\rho, s_{\tau+1}\sim\widehat{P}^{\pi_K}_{s_\tau,a_\tau}, a_\tau\sim\pi_K(s_\tau)} \sum_{t=0}^{+\infty} \gamma^t([Q^\star - Q_K](s_t, \pi^\star(s_t))) + ([Q_K - Q^\star](s_t, \pi_K(s_t)))$$

$$\overset{\text{Lemma 9}}{\le} 2\sum_{t=0}^{+\infty} \gamma^t \left( \frac{\gamma^K}{1-\gamma} + C\sum_{k=1}^{K} \gamma^{K-k} \|Q_k - \widetilde{\mathcal{T}}_Q Q_{k-1}\|_{1,\mu} \right)$$

$$= \frac{2\gamma^K}{(1-\gamma)^2} + \frac{2C}{1-\gamma} \sum_{k=1}^{K} \gamma^{K-k} \|Q_k - \widetilde{\mathcal{T}}_Q Q_{k-1}\|_{1,\mu}$$

$$\overset{\text{Lemma 10}}{\le} \frac{2\gamma^K}{(1-\gamma)^2} + \frac{2C}{1-\gamma} \gamma\beta^{-1} e^{\frac{\beta}{1-\gamma}} \sum_{k=1}^{K} \gamma^{K-k} \left( \|(\widehat{\mathcal{T}}_{Z,\mathcal{F}} - \mathcal{T}_{Z,\mathcal{F}})Z_{k-1}\|_{1,\mu} + \|(\mathcal{T}_{Z,\mathcal{F}} - \mathcal{T}_Z)Z_{k-1}\|_{1,\mu} \right)$$

$$\le \frac{2\gamma^K}{(1-\gamma)^2} + \frac{2C}{1-\gamma} \gamma\beta^{-1} e^{\frac{\beta}{1-\gamma}} \sum_{k=1}^{K} \gamma^{K-k} \left( \|(\widehat{\mathcal{T}}_{Z,\mathcal{F}} - \mathcal{T}_{Z,\mathcal{F}})Z_{k-1}\|_{2,\mu} + \|(\mathcal{T}_{Z,\mathcal{F}} - \mathcal{T}_Z)Z_{k-1}\|_{2,\mu} \right)$$

$$\overset{\text{Lemma 11}}{\le} \frac{2\gamma^K}{(1-\gamma)^2} + \frac{2C}{1-\gamma} \gamma\beta^{-1} e^{\frac{\beta}{1-\gamma}} \sum_{k=1}^{K} \gamma^{K-k} \left( 4\sqrt{\frac{2\log(|\mathcal{F}|)}{N}} + 5\sqrt{\frac{2\log(8/\delta)}{N}} + \epsilon_c \right)$$

$$\le \frac{2\gamma^K}{(1-\gamma)^2} + \gamma\beta^{-1} e^{\frac{\beta}{1-\gamma}} \frac{2C}{(1-\gamma)^2} \left( 4\sqrt{\frac{2\log(|\mathcal{F}|)}{N}} + 5\sqrt{\frac{2\log(8/\delta)}{N}} + \epsilon_c \right)$$

$\square$

## I AUXILIARIES

**Lemma 12.**

$$|Q^{\pi'}(s_0, a_0) - Q^{\pi}(s_0, a_0)| \le \frac{1}{(1-\gamma)^2} \max_s \|\pi'_s - \pi_s\|_1,$$

*where $\pi_s$ denotes $\pi(\cdot|s) \in \Delta^{|\mathcal{S}|}$ and $\|\pi'_s - \pi_s\|_1 = \sum_a |\pi'(a|s) - \pi(a|s)|$.*

*Proof.*

$$Q^{\pi'}(s_0, a_0) = \min_{\{\widehat{P}_t\}_{t\geq 1}} \mathbb{E}_{s_t, a_t \sim \pi', \widehat{P}} \sum_{t=0}^{+\infty} \gamma^t \left( r(s_t, a_t) + \gamma d(\widehat{P}_{t; s_t, a_t}, P_{s_t, a_t}) \right)$$

$$\leq \mathbb{E}_{s_t, a_t \sim \pi', \widehat{P}^{\pi}} \sum_{t=0}^{+\infty} \gamma^t \left( r(s_t, a_t) + \gamma d(\widehat{P}_{t; s_t, a_t}, P_{s_t, a_t}) \right).$$

$$Q^{\pi}(s_0, a_0) = \mathbb{E}_{s_t, a_t \sim \pi, \widehat{P}^{\pi}} \sum_{t=0}^{+\infty} \gamma^t \left( r(s_t, a_t) + \gamma d(\widehat{P}_{t; s_t, a_t}, P_{s_t, a_t}) \right)$$

Thus

$$Q^{\pi'}(s_0, a_0) - Q^{\pi}(s_0, a_0) \leq \mathbb{E}_{s_t, a_t \sim \pi', \widehat{P}^{\pi}} \sum_{t=0}^{+\infty} \gamma^t \left( r(s_t, a_t) + \gamma d(\widehat{P}_{t; s_t, a_t}, P_{s_t, a_t}) \right)$$

$$- \mathbb{E}_{s_t, a_t \sim \pi, \widehat{P}^{\pi}} \sum_{t=0}^{+\infty} \gamma^t \left( r(s_t, a_t) + \gamma d(\widehat{P}_{t; s_t, a_t}, P_{s_t, a_t}) \right)$$

$$= \mathbb{E}_{s_t \sim \pi', \widehat{P}} \sum_{t=0}^{+\infty} \gamma^t \sum_a (\pi'(a|s) - \pi(a|s)) Q^{\pi}(s, a) \quad \text{(by Lemma 7)}$$

$$\leq \frac{1}{1 - \gamma} \mathbb{E}_{s_t \sim \pi', \widehat{P}} \sum_{t=0}^{+\infty} \gamma^t \sum_a |\pi'(a|s) - \pi(a|s)|$$

$$\leq \frac{1}{(1 - \gamma)^2} \max_s \|\pi'_s - \pi_s\|_1$$

$\square$

**Lemma 13.** *For any convex set $\mathcal{X} \subset \mathbb{R}^n$ and $x \in \mathcal{X}, f \in \mathbb{R}^n, \eta > 0$,*

$$\langle \mathrm{Proj}_{\mathcal{X}}(x + \eta f), f \rangle \geq \frac{1}{\eta} \|\mathrm{Proj}_{\mathcal{X}}(x + \eta f) - x\|_2^2.$$

*Proof.* From the definition of projection, for any $y \in \mathcal{X}$,

$$\langle y - \mathrm{Proj}_{\mathcal{X}}(x + \eta f), x + \eta f - \mathrm{Proj}_{\mathcal{X}}(x + \eta f) \rangle \leq 0,$$

Set $y = x$ we get:

$$\|\mathrm{Proj}_{\mathcal{X}}(x + \eta f) - x\|_2^2 \leq \langle \mathrm{Proj}_{\mathcal{X}}(x + \eta f), \eta f \rangle,$$

which completes the proof. $\square$

**Lemma 14.** *The operators $\mathcal{T}_Q, \mathcal{T}_{V \to Q}$ defined in (15), (22) satisfies*

$$[\mathcal{T}_{V \to Q}(\overline{V} - V)](s, a) \leq \gamma \mathbb{E}_{s' \sim \widehat{P}_{s,a}} (\overline{V} - V)(s')$$

$$[\mathcal{T}_Q(\overline{Q} - Q)](s, a) \leq \gamma \mathbb{E}_{s' \sim \widehat{P}_{s,a}} \max_{a'} (\overline{Q} - Q)(s', a'),$$

*where $\widehat{P}_{s,a}$ is defined as:*

$$\widehat{P}(s'|s, a) \propto P(s'|s, a) \exp(-\beta V(s')) \quad \text{or} \quad \widehat{P}(s'|s, a) \propto P(s'|s, a) \exp(-\beta \max_{a'} Q(s', a')).$$

*Proof.* Let $\sigma(P_{s,a}, V) := \beta^{-1} \log \mathbb{E}_{s' \sim P_{s,a}} e^{-\beta V(s')}$, then from Example 1 and the dual representation theorem 1

$$\sigma(P_{s,a}, V) := \sup_{\widetilde{P}_{s,a}} -\mathbb{E}_{s' \sim \widetilde{P}_{s,a}} V(s') - \beta^{-1} \mathrm{KL}(\widetilde{P}_{s,a} \| P_{s,a}).$$

$$[\mathcal{T}_{V \to Q}(\overline{V} - V)] = -\gamma \left( \sigma(P_{s,a}, \overline{V}) - \sigma(P_{s,a}, V) \right) \tag{23}$$

$$= \gamma \left( \inf_{\widetilde{P}_{s,a}} \left( \mathbb{E}_{s' \sim \widetilde{P}_{s,a}} \overline{V}(s') + \beta^{-1} \mathrm{KL}(\widetilde{P}_{s,a} || P_{s,a}) \right) - \inf_{\widetilde{P}_{s,a}} \left( \mathbb{E}_{s' \sim \widetilde{P}_{s,a}} V(s') + \beta^{-1} \mathrm{KL}(\widetilde{P}_{s,a} || P_{s,a}) \right) \right)$$

$$= \gamma \left( \inf_{\widetilde{P}_{s,a}} \left( \mathbb{E}_{s' \sim \widetilde{P}_{s,a}} \overline{V}(s') + \beta^{-1} \mathrm{KL}(\widetilde{P}_{s,a} || P_{s,a}) \right) - \left( \mathbb{E}_{s' \sim \widehat{P}_{s,a}} V(s') + \beta^{-1} \mathrm{KL}(\widehat{P}_{s,a} || P_{s,a}) \right) \right)$$

$$\leq \gamma \left( \left( \mathbb{E}_{s' \sim \widehat{P}_{s,a}} \overline{V}(s') + \beta^{-1} \mathrm{KL}(\widehat{P}_{s,a} || P_{s,a}) \right) - \left( \mathbb{E}_{s' \sim \widehat{P}_{s,a}} V(s') + \beta^{-1} \mathrm{KL}(\widehat{P}_{s,a} || P_{s,a}) \right) \right)$$

$$= \gamma \mathbb{E}_{s' \sim \widehat{P}_{s,a}} (\overline{V} - V)(s') \tag{24}$$

Similarly,

$$[\mathcal{T}_Q(\overline{Q} - Q)](s,a) = -\gamma \beta^{-1} \log \mathbb{E}_{s' \sim P(\cdot|s,a)} e^{-\beta \max_{a'} \overline{Q}(s',a')} + \gamma \beta^{-1} \log \mathbb{E}_{s' \sim P(\cdot|s,a)} e^{-\beta \max_{a'} Q(s',a')}$$

$$= -\gamma \left( \sigma(P_{s,a}, \max_a Q(\cdot, a)) - \sigma(P_{s,a}, \max_a \overline{Q}(\cdot, a)) \right)$$

Using the same inequality from (23) to (24) we get

$$-\gamma \left( \sigma(P_{s,a}, \max_a Q(\cdot, a)) - \sigma(P_{s,a}, \max_a \overline{Q}(\cdot, a)) \right) \leq \mathbb{E}_{s' \sim \widehat{P}_{s,a}} \max_a \left( \overline{Q}(s', a) - \max_a \overline{Q}(s', a) \right)$$

$$\leq \mathbb{E}_{s' \sim \widehat{P}_{s,a}} \max_a \left( \overline{Q}(s', a) - \overline{Q}(s', a) \right),$$

which completes the proof. $\qquad \square$

**Lemma 15.**

$$|\log x - \log y| \leq \frac{1}{\min\{x, y\}} |x - y|$$

*Proof.* Without loss of generality, we assume $x \geq y$, then

$$|\log x - \log y| = \log x - \log y = \log(1 + \frac{x - y}{y})$$

$$\leq \frac{x - y}{y} = \frac{1}{y}(x - y) = \frac{1}{\min\{x, y\}} |x - y|,$$

which completes the proof. $\qquad \square$