# OpenReview forum: "Soft Robust MDPs and Risk-Sensitive MDPs: Equivalence, Policy Gradient, and Sample Complexity"
_ICLR.cc/2024/Conference — ICLR 2024 poster_

### Official Review · Reviewer_KqgB · 2023-10-17

**Soundness:** 3 good
**Presentation:** 3 good
**Contribution:** 2 fair
**Rating:** 6
**Confidence:** 4

**Summary:**

The paper establishes an equivalence between a specific class of regularised robust Markov Decision Processes (MDP) and the risk-sensitive MDP. The authors derive the exact policy gradient method and demonstrate its global convergence in tabular-setting MDP. Specifically, the authors study the regularised MDP with KL divergence regularisation term, and design the sample-based offline learning algorithm, robust fitted-Z iteration.

**Strengths:**

The paper bridges two distinct types of MDP, and introduces a policy gradient method to compute their solutions. The theoretical equivalences between these MDP types are intriguing. Moreover, the proposed algorithm demonstrates global convergence. The writing of the paper is good.

**Weaknesses:**

The framework of the robust MDP and risk-sensitive MDP is not clear enough: The authors fail to explicitly differentiate between stationary and non-stationary policies, as well as between stochastic and deterministic policies. While Equations (1) and (2) suggest that the authors are considering a robust MDP with a non-stationary and deterministic policy, Equations (7) and (8) indicate a stationary and stochastic policy for the risk-sensitive MDP. A similar inconsistency is observed in the transition kernel: while the robust MDP seems to consider the worst-case transition kernel for each time horizon (as evidenced by the decision variables in the infimum problem of Equations (1) and (2)), the risk-sensitive MDP presumes the existence of a transition kernel that is independent of time $t$.

**Questions:**

1. Within the framework of risk-sensitive MDP, the value function for a given policy $\pi$, denoted as $\tilde{V}^{\pi}(s)$, is defined by equation (7). The optimal value function, $\tilde{V}^\star$, is defined as $\max_{\pi} \tilde{V}^{\pi}$. Could the authors elucidate how $\tilde{V}^\star$ satisfies the equation (9)?

2. As noted in remark 7, the sample complexity of the proposed algorithm appears less favorable than that detailed in reference [11], assuming that there is no function approximation error in authors' approach, $\epsilon_c = 0$. What then are the advantages of the corresponding RFZI algorithm?
Furthermore, given the resemblance between Equations (7) and (9) and the traditional Bellman equation, and considering that the operators are proven contractions, is it feasible for the authors to employ value iteration or policy iteration to resolve the MDP? Such methods should theoretically exhibit rapid (geometric) convergence and remain unaffected by unknown exact gradients present in policy gradient methods.

3. At the end of remark 7, the authors claim that the choice of $\beta$ should not be too large or too small, and would be better if it is on the same scale with $1-\gamma$. $\beta$ is the radius of the ambiguity set related to transition kernel and $\gamma$ is the discounted factor. It seems unconventional to determine the ambiguity set's radius based solely on the discount factor. Could this choice be elaborated upon?

4. In 'E.2 Proof of theorem 2 (infinite horizon case)' on page 23, the authors claim that it is not hard to show $\lim_{h\to +\infty} \bar{V}^\star_{0:h} = \bar{V}^\star, \lim_{h\to +\infty} \bar{V}^{\pi}_{0:h} = \bar{V}^{\pi}$. This assertion doesn't immediately resonate with me. Would it be possible to furnish a detailed proof for this claim? Subsequent statements, $\bar{V}^\star = \mathcal{T}^\star \bar{V}^\star $ and $\bar{V}^\pi = \mathcal{T}^\pi \bar{V}^\star $ also require clarification.

---

> ### Author Response · Authors · 2023-11-16
> **Response to Reviewer KqgB (Part 1)**
>
> We sincerely thank the reviewer for a thorough examination of our paper and for giving critical and valuable feedback! Prior to addressing the reviewer's comments, we would like to note that, due to the revision, reference/equation/theorem numbers may differ. To maintain consistency with the reviewers' comments, *our responses will refer to the original version's reference/equation/theorem numbers unless explicitly specified otherwise.* Additionally, based on the insightful suggestion of Reviewer RSEz, we changed the term 'regularized robust MDP' to 'soft robust MDP'.
>
> We now address the reviewer's questions and concerns as follows:
>
> Weakness:
>
> - *'The authors fail to explicitly differentiate between stationary and non-stationary policies, as well as between stochastic and deterministic policies. While Equations (1) and (2) suggest that the authors are considering a robust MDP with a non-stationary and deterministic policy, Equations (7) and (8) indicate a stationary and stochastic policy for the risk-sensitive MDP. A similar inconsistency is observed in the transition kernel: while the robust MDP seems to consider the worst-case transition kernel for each time horizon (as evidenced by the decision variables in the infimum problem of Equations (1) and (2)), the risk-sensitive MDP presumes the existence of a transition kernel that is independent of time.'*
>
> We sincerely apologize for the confusion. However, we would like to clarify that our definitions and derivations are technically correct. For the sake of generality, in Eq (1),(2) and (3) of the definition of robust MDP, we allow the policy as well as the transition kernel to be non-stationary. However, Theorem 2 points out that the sup-inf solution of equation (1)(2)(3) can be obtained by stationary policies and stationary transition kernels (Eq (11),(12)). Thus it is sufficient to restrict the later discussions to stationary policies and transition kernels, which is why our definition of value functions (e.g. Eq (4) and (7)) are only defined for stationary policies. Regarding the reviewers' concern about the differentiation between deterministic policies and stochastic policies, we apologize for our inaccurate notations which caused this confusion. The optimal value function is actually defined as the supremum over all possible policies which can be potentially stochastic, non-stationary and non-Markovian. However, the optimal solution can be obtained by a deterministic stationary Markov policy. We have updated our definition of equation (1)(2)(3) and added the above discussion as a footnote in the revised paper (highlighted in blue).
>
> We would also like to note that our definition and analysis resemble the standard argument for risk-neutral MDP counterparts, where the original definition of the optimal value function considers taking the supremum over all possible non-stationary, even non-Markovian policies, yet it is shown that the optimal policy can be obtained by a stationary deterministic Markov policy. We hope that the above discussion clarifies the reviewer's confusion and we are open to further discussions if there are any follow-up questions from the reviewer.
>
> Questions:
>
> - *'Within the framework of risk-sensitive MDP, the value function for a given policy $\pi$, denoted as $\widetilde{V}^\pi(s)$, is defined by equation (7). The optimal value function, $\widetilde{V}^\star$
> , is defined as $\max_\pi \widetilde{V}^\pi(s)$. Could the authors elucidate how $\widetilde{V}^\star$ satisfies the equation (9)?'*
>
> Thank you for reading carefully into the technical details in the paper! We would like to clarify that the definition of the optimal value function $\widetilde{V}^\star$ is actually initially defined as Eq (9) and then in Theorem 2, we show that $\widetilde{V}^\star(s) = \max_{\pi} \widetilde{V}^\pi(s)$, i.e. the optimal value function can be achieved by a Markov policy, which is proved in Appendix E at the beginning of page 24, where it shows that there exists a $\pi^\star$, such that $\widetilde{V}^\star = \widetilde{V}^{\pi^\star}$.

---

> > ### Author Response · Authors · 2023-11-16
> > **Response to Reviewer KqgB (Part 2)**
> >
> > - *'As noted in remark 7, the sample complexity of the proposed algorithm appears less favorable than that detailed in reference [11], assuming that there is no function approximation error in the authors' approach, $\epsilon_c = 0$. What then are the advantages of the corresponding RFZI algorithm? Furthermore, given the resemblance between Equations (7) and (9) and the traditional Bellman equation, and considering that the operators are proven contractions, is it feasible for the authors to employ value iteration or policy iteration to resolve the MDP? Such methods should theoretically exhibit rapid (geometric) convergence and remain unaffected by unknown exact gradients present in policy gradient methods.'*
> >
> > Regarding the reviewer's question about the advantage of RFZI over reference [11], we would like to clarify that reference [11] considers the standard non-robust/risk-neutral RL, which is a different and possibly simpler setting compared with the robust/risk-sensitive setting considered in this paper. Based on the suggestion by other reviewers, we also added a discussion that compares our paper with an offline learning algorithm for RMDP [61] which is a more fair comparison (see Remark 6 in the revised paper (previously Remark 7 in the original version) and the response to Reviewer SuJM [Performance comparison with RFQI algorithm in [61]] for more details).
> >
> > Regarding the reviewer's question of employing value iteration or policy iteration to resolve the MDP, our RFZI algorithm actually resembles policy iteration. In specific, the motivation of the algorithm is to solve the optimal Q function by solving the Bellman equation $Q^\star =T_{Q} Q^\star$. However, $T_{Q}$ involves a term $\log E_{s'\sim P_{s,a}}$ which is hard to approximate with empirical estimation. Thus, instead of directly solving $Q^*$ using $Q^\star=T_Q Q^\star$, we introduce the Z-functions as auxiliary variables and solve the approximated fix point equation for $Z$. And the contraction of the Bellman operator is also reflected in our theoretical bound, where the first part $\frac{2\gamma^K}{(1-\gamma)^2}$ captures the effect of $\gamma$-contraction of the Bellman operators.
> >
> > - *'At the end of remark 7, the authors claim that the choice of $\beta$ should not be too large or too small, and would be better if it is on the same scale with $1-\gamma$. $\beta$ is the radius of the ambiguity set related to the transition kernel and $\gamma$ is the discounted factor. It seems unconventional to determine the ambiguity set's radius based solely on the discount factor. Could this choice be elaborated upon?'*
> >
> >  Thank you for the valuable question. We would first like to clarify that $\beta$ serves as an indicator of the penalty strength of the deviation of $\widehat{P}$ from $P$ instead of the radius of the ambiguity set. The smaller $\beta$
> > is, the larger the penalty strength. Thus it can be expected that the sample complexity bound depends on both $\beta$ and $\gamma$, as they together determine the landscape of the problem. We concur with the reviewer that it looks counter-intuitive at first glance that the choice of $\beta$ should be 'on the same scale with $1-\gamma$'. The main reason lies in the exponential dependency term $O\left(e^{\frac{\beta}{1-\gamma}}\right)$ that shows up in the upper bound. It is still unclear to us whether it is a proof artifact or intrinsic in our setting, however, there are results under similar settings that suggest this exponential dependency is fundamental (e.g. Theorem 3 in [23]). Numerical evidence also suggests that large $\beta$ would affect the learning stability and typically people choose the hyper-parameter $\beta$ to be roughly on the same scale as $1-\gamma$ (e.g. [55]). Additionally, we emphasize that the choice of $\beta$ doesn't depend solely on $\gamma$ and might be related to the specific structure of the MDP, hence validation and hyperparameter tuning is still necessary for choosing $\beta$.  We have added part of the above discussion in Remark 6 in the revised paper (previously Remark 7 in the original version).

---

> > > ### Author Response · Authors · 2023-11-16
> > > **Response to Reviewer KqgB (Part 3)**
> > >
> > > - *'In 'E.2 Proof of theorem 2 (infinite horizon case)' on page 23, the authors claim that it is not hard to show $\lim_{h\to+\infty}\overline{V}^\star_{0:h} =\overline{V}^\star, \lim_{h\to+\infty}\overline{V}_{0:h}^\pi = \overline{V}^\pi$. This assertion doesn't immediately resonate with me. Would it be possible to furnish a detailed proof for this claim? Subsequent statements, $\overline{V}^\star = \mathcal{T}^\star\overline{V}^\star $ and $\overline{V}^\pi = \mathcal{T}^\pi \overline{V}^\pi $ also require clarification.'*
> > >
> > > Thank you again for reading carefully into the proofs! We have added a more detailed proof in the corresponding locations (highlighted in blue). The key step is to show that $|\overline{V}^\pi(s) - \overline{V}^\pi_{0:h}(s)| |\overline{V}^\star(s) - \overline{V}_{0:h}^\star(s)| \le \frac{\gamma^h}{1-\gamma}$. And then by leveraging the continuous property of the operators $\mathcal{T}^\pi, \mathcal{T}^\star$, we get $\overline{V}^\star = \mathcal{T}^\star\overline{V}^\star, \overline{V}^\pi = \mathcal{T}^\pi \overline{V}^\pi $.
> > >
> > > Once again, we extend our sincere gratitude to the reviewer for the insightful comments and questions. We hope that our response clarifies the reviewer's concerns and we are more than willing to engage in further discussion to address any additional questions or concerns the reviewer may have.

---

> > > > ### Comment · Reviewer_KqgB · 2023-11-22
> > > >
> > > > Thank the authors for detailed explanations and answers to my concerns, especially for the illustrations of remark 6, due to which I raise the score of presentation and final rating.

---

> > > > > ### Author Response · Authors · 2023-11-23
> > > > > **Thank you for your feedback**
> > > > >
> > > > > Thank you very much for your response. We are glad to know that we have addressed your major concerns. Thank you again for your time in reviewing the paper and providing helpful suggestions for improving the clarity of the paper!

---

### Official Review · Reviewer_BzBp · 2023-10-24

**Soundness:** 3 good
**Presentation:** 3 good
**Contribution:** 3 good
**Rating:** 8
**Confidence:** 4

**Summary:**

The authors provide a sufficient condition for establishing the equivalence between the regularized robust MDPs and risk-sensitive MDPs, and provide the optimal policy for them when the condition is satisfied. Under some assumptions, the authors provide a quasi-policy gradient algorithm with a global convergence rate. They also provide KL-regularized RMDPs as an instantiation of their framework.

**Strengths:**

The theoretical contribution is inspiring. I like theorem 2 where the authors provide an elegant sufficient condition for the equivalence of regularized RMDPs and risk-sensitve MDPs.

**Weaknesses:**

Could the authors provide some examples where theorem 2 is applied to establish the equivalence between some popular regularized RMDPs and risk-sensitve MDPs?

**Questions:**

- In section 2, the authors introduce convex risk measures, where there is only one input in the measure $\sigma$ in (5) and (6), while there are two in examples 1 and 2. Could the authors explain what is the difference between these two $\sigma$'s?

- What is "direct parameterization"?

- Is the right-hand side of (15) independent of the initial distribution $\rho$?

- About the feasible region $\mathcal{X}$ of $\theta$: why it is of dimension $SA$? And why each subvector $\theta_s$ should be in the probability simplex?

- Assumption 1, typo in the first sentence.

- Assumption 3, what is the norm with subscript "$2,\mu$"?

---

> ### Author Response · Authors · 2023-11-16
> **Response to Reviewer BzBp**
>
> We sincerely thank the reviewer for appreciating the contribution of our work! Prior to addressing the reviewer's comments, we would like to note that, due to the revision, reference/equation/theorem numbers may differ. To maintain consistency with the reviewers' comments, *our responses will refer to the original version's reference/equation/theorem numbers, unless explicitly specified otherwise.* Additionally, based on the insightful suggestion of Reviewer RSEz, we changed the term 'regularized robust MDP' to 'soft robust MDP'.
>
> We now address the reviewer's questions as follows:
>
> - *'Could the authors provide some examples where theorem 2 is applied to establish the equivalence between some popular regularized RMDPs and risk-sensitve MDPs?'*
>
> One example of the soft RMDPs and risk-sensitive MDPs is the soft-KL RMDP, which is equivalent to the risk-sensitive MDP with entropy regularizer. This example is discussed in more detail in Section 5 of the paper.
>
> Another example can be the risk-sensitive MDP with Conditional value at risk (CVaR, see Example 2 in the original version). Leveraging Theorem 2, it is equivalent to the RMDP with uncertainty set $\frac{\widehat{P}(s'|s,a)}{P(s'|s,a)}\le {\alpha^{-1}}, \forall ~s',s\in\mathcal{S}, a\in\mathcal{A} $
>
> - *'In section 2, the authors introduce convex risk measures, where there is only one input in the measure $\sigma$ in (5) and (6), while there are two in examples 1 and 2. Could the authors explain what is the difference between these two $\sigma$'s?'*
>
> Thank you for the question! In the paper, we provide an explanation after Theorem 1 that clarifies the two inputs in the risk measure $\sigma$ as well as the penalty function $D$, which we quote as follows: 'In most cases, the convex risk measure $\sigma(V)$ can be interpreted as the risk associated with a random variable that takes on values $V(s)$ where $s$ is drawn from some distribution $s\sim\mu$. Consequently, most commonly used risk measures are typically associated with an underlying probability distribution $\mu\in\Delta^{|\mathcal S|}$ (e.g., Examples 1 and 2). This paper focuses on this type of risk measures and thus we use $\sigma(\mu,\cdot)$ to denote the risk measure, where the additional variable $\mu$ indicates the associated probability distribution.
> Correspondingly, we denote the penalty term $D(\hat{\mu})$ of $\sigma(\mu,\cdot)$ in the dual representation theorem as $D(\hat{\mu},\mu)$.' We hope that this answers the reviewer's question and we are happy to further discuss if there are any parts that remain ambiguous.
>
> - *'What is "direct parameterization"?'* and *'About the feasible region $\mathcal X$ of $\theta$ : why it is of dimension $SA$
> ? And why each subvector $\theta_s$ should be in the probability simplex?'*
>
> Direct parameterization is one of the most commonly considered approaches to parameterize a policy in theoretical RL (c.f. [2]). As we have stated before Corollary 1, 'for direct parameterization, the parameter $\theta_{s,a}$ directly represents the probability of choosing action $a$ at state $s$, i.e., $\theta_{s,a} = \pi_\theta(a|s)$.' We have also added some standard references for direct parameterization in the same paragraph.
>
> Given this definition of direct parameterization, $\theta$ is of dimension $SA$ because for each $s$ and $a$, there's a corresponding entry $\theta_{s,a}$ that represents $\theta_{s,a} = \pi_\theta(a|s)$. Further, the subvect $\theta_{s}$ represents $\theta_s = \pi_\theta(\cdot|s)$ which lies in the probability simplex. We hope that the above response answers the reviewer's question!
>
> *'Is the right-hand side of (15) independent of the initial distribution $\rho$?'*
>
> Thank you for reading into the technical details. The right-hand side of (15) is actually dependent on the initial distribution $\rho$ because the discounted state visitation distribution $d^{\pi_\theta, \widehat P^\theta}$ (defined in Section 2) is dependent on $\rho$.
>
> - *'Assumption 1, typo in the first sentence.'*
>
> Thank you for pointing out the typo! We have corrected it in the revised paper.
>
> - *'Assumption 3, what is the norm with subscript "$2,\mu$"'?*
>
> Thank you very much for the question! We recognized that we didn't give a formal definition of $||\cdot||_{2,\mu}$. We have added the definition to Section 2 in the revised paper, which is stated as below: for any function $f:\mathcal{S}\times\mathcal{A}\to \mathbb{R}$, state-action distribution $\mu\in\Delta(\mathcal{S}\times\mathcal{A})$, the $\mu$-weighted 2-norm of $f$ is defined as
>
> $||f||{2,\mu}  =(E_{s,a\sim\mu}f(s, a)^2)^{1/2}$.
>
> We have added the above definition to the preliminary section of the paper (highlighted in blue). We sincerely appreciate your contribution in highlighting this particular omission.
>
> In conclusion, we express our sincere gratitude to the reviewer for the constructive comments and feedback. We are happy to have further discussions if there are any other questions!

---

> > ### Comment · Reviewer_BzBp · 2023-11-22
> > **Thank you for your detailed reponse!**
> >
> > Thank you for addressing my comments in detail! I will raise my score.

---

> > > ### Author Response · Authors · 2023-11-23
> > > **Thank you for your feedback**
> > >
> > > Thank you very much for your response! We appreciate your recognition of our efforts and contributions. Thank you once again for your thoughtful feedback and valuable suggestions for enhancing the overall quality of the paper.

---

### Official Review · Reviewer_RSEz · 2023-10-27

**Soundness:** 3 good
**Presentation:** 4 excellent
**Contribution:** 2 fair
**Rating:** 6
**Confidence:** 4

**Summary:**

The paper presents an extension of robust MDP that replaces the uncertainty set for the uncertain transition matrix with a penalty for the adversary. Connections are made between this approach and risk sensitive MDPs when the policy is deterministic. For the case of a penalty of the form of KL divergence, the regularized robust MDP reduces to a risk averse MDP with a nested entropic risk measure. A value iteration approach à la Fei et al. (2021) is proposed to solve the problem. When a stochastic policy is used, the property of gradient domination under direct parametrization is established.

**Strengths:**

The paper is very well written. The notation is clear and proofs appear all accurate based on the amount of time I could invest in checking them. The paper makes connection with many different streams of the literature: robust MPD, risk measures, gradient dominance, etc.

**Weaknesses:**

My main concerns are three-fold:

1) Theorem 2 holds only for risk measures \sigma that satisfy quasi-concave and does not hold if the policy is regularized. ﻿ ﻿ ﻿Indeed,  first I enocurage the authors to have a look at Delage et al. (2019) for an investigation of what risk measures have the mixture quasi-concave property and the implications in terms of optimality of randomized policies. Even in the case of mixture quasi-concave risk measures (such as CVaR and entropic risk measure), I find this result to be problematic. First, it implies that the regularized robust MDP will underestimate the risk of stochastic policies (as it is taking the average). Second, the equivalence fails if one include policy regularization which could force the optimal policy to become stochastic, thus misestimated by regularized RMDP. I would therefore limit the connection to a dynamic risk preference relation (see Pichlet et al. 2021) setting of the form:  RPR( [r_1,…, r_infty] ) = rho_0( r_1 + gamma E[ rho_1( r_2 + gamma E[ rho_2(… ) |a_2 ]|s_2)|a_1]).    where I would emphasize that the policies effect is treated as risk neutral and the transitions as risk averse.

2) Many of the results appear to follow closely from existing work. ﻿
Theorem 1 comes from [27]. The equivalence between soft robust optimization and convex risk measure is already established in Ben-Tal et al. (2010). It is therefore no surprise that the result in theorem 2 would hold. Theorem 3, Lemma 2, and Theorem 4 extend the result of [85] from robust MPD to the slighlty more general regularized/soft robust MDP. Section 5 proposes an algorithm that resembles the approach presented in [24] although I am not aware of a sample complexity result for that class of algorithm.

3) The concept of regularized robust is misleading. In problem (2) it is the adversarial transition probability matrix that is being regularized. This is completely different from regularizing the policy as is usually understood as "regularized MDP”. I would suggesting calling it differently to avoid this confusion. Namely, in the robust optimization literature, the concept of regularizing the adversaries actions is referred as soft-robust, comprehensive robust, or globalized robust optimization (see references below).


Ben-Tal, S. Boyd, and A. Nemirovski. Extending scope of robust optimization: Comprehensive robust counterparts of uncertain problems. Mathematical Programming, 107(1-2):63–89, 2006.

A. Ben-Tal, D. Bertsimas, and D. B. Brown. A soft robust model for optimization under ambiguity. Operations Research, 58(4-part-2):1220–1234, 2010

A. Ben-Tal, R. Brekelmans, D. den Hertog, and J.-P. Vial. Globalized robust optimization for nonlinear uncertain inequalities. INFORMS Journal on Computing, 29(2):350–366, 2017.

E. Delage, D. Kuhn, and W. Wiesemann, “Dice”-sion–Making Under Uncertainty: When Can a Random Decision Reduce Risk? Management Science, 65(7):3282-3301, 2019.

A. Pichler, R. P. Liu, and A. Shapiro. Risk-averse stochastic programming: Time consistency and optimal stopping. Operations Research, 70:4, 2439-2455, 2021


Proof to verify:
Page 34, in equation (25), it should be illustrated how the second inequality holds, which seems not obvious.


Typos/clarifications:

Page 5, in the middle "Risk-Sensitive MDPs" misspell "Senstive"
Page 5, the line above equation (7), misspell "valule"
Page 8, Assumption 1, "For any policy pi and " missing P
Page 25, paragraph under section C, last part, repeat expression "is defined as is defined as”
Page 26, the notation would be clearer if the two “T^\pi” and “T^*” operators were denoted as “bar T^\pi” and “bar T^*” since they are defined in terms of the robust MDP
Page 26, last part of Proof of Lemma1, "Lemma 1 is an immediate corollary of Lemma 8", it should be Lemma 3. "\tilde V and \tilde V^\star in (4) and (3)", it should be (7) and (9).
Page 27, in Lemma 4, the definition of \mathcal{T}^* should be clarified.
Page 28, In the proof of Lemma 4, \mathbb{E}{s_1\sim \hat{P}{0}  should have a uniform expression as the passages above and below, i.e., \mathbb{E}{s_1\sim \hat{P}{0;s_0,a_0}.
In line 4, there is a missing bracket in the last equation.
Page 32, in Part G, the performance Difference Lemma [40] should be stated separately, as it has been applied a lot of times in the proofs. It is better to illustrate how this lemma is applied to this specific case.

**Questions:**

The authors are invited to comment on the three main concern described above.

---

> ### Author Response · Authors · 2023-11-16
> **Response to Reviewer RSEz (Part 1)**
>
> We sincerely thank the reviewer for carefully reading our paper, acknowledging our contributions, and giving insightful and critical feedback! Prior to addressing the reviewer's comments, we would like to note that, due to the revision, reference/equation/remark numbers may differ. To maintain consistency with the reviewers' comments, *our responses will refer to the original version's reference/equation/theorem numbers unless explicitly specified otherwise.* We now address the concerns raised by the reviewer in the 'Weakness' section as follows:
>
> **[Term of 'regularized robust MDP']** The reviewer pointed out that the term 'regularized robust MDP' might be misleading. We sincerely thank the reviewer for presenting more accurate alternatives and providing related references. After reading the references, we fully agree with the reviewer that 'soft robust MDP' indeed encapsulates the intended concept more accurately. We have changed the name of the term to `soft RMDP' and added related references in our revised paper. We will use soft RMDP in the response as well.
>
> **[Limitation of Theorem 2]** The reviewer has pointed out several limitations of Theorem 2, which we summarize below: 1. The equivalence only holds for convex risk measures (referred to as 'quasi-concave/mixture quasi-concave' by the reviewer) and does not generalize to other risk metrics such as mean (semi-)deviation
> and mean (semi-)moment measures in Delage et al. (2019) ; 2. The equivalence would fail if policy regularization is added to the objective function, in which case the optimal policy might be stochastic. 3. The risk-sensitive MDP underestimates the risk caused by the stochasticity generated by the policy.
>
> We now address the reviewer's comments point by point:
>
> For point 1, we agree with the reviewer that the equivalence no longer holds if we consider other risk metrics other than convex risk measures. However, we would like to note that the convex risk measure is an important risk metric and is commonly used in finance and operations research. Consequently, our findings establish a meaningful foundation for exploring risk-sensitive sequential decision-making, albeit within the confines of convex risk measures. An intriguing avenue for future research would involve investigating reinforcement learning or sequential decision-making in the context of diverse risk metrics.
> However, given the distinct properties of risk metrics like semi-deviation and semi-moments in comparison to convex risk measures, different theoretical tools may be necessary, which poses challenges to the analysis, and unfortunately falls outside the current scope of this paper.
>
> For point 2, we would like to clarify that this is actually not the case: the equivalence between the value functions of risk-sensitive MDPs $\widetilde V^\pi$ and soft RMDPs $\overline V^\pi$ will still hold under most classical policy regularization methods (e.g. entropy regularization). Note that Theorem 2 holds not just for deterministic policies, but also for stochastic policies. For a policy $\pi$, adding regularization only requires changing the reward function $r(s,a)$ to be $r^\pi(s,a) = r(s,a) + \mathcal{R}(\pi(\cdot|s))$, where $\mathcal{R}$ is the policy regularizer. In this setting, the proof of Theorem 2 can still carry through naturally. We have added the above discussion as a footnote in the revised paper (highlighted in blue).
>
> For point 3, we would like to first thank the reviewer for the insightful comment! The reviewer is correct that the proposed risk-sensitive MDP focuses on the risk induced by the uncertainty set but does not consider the risk caused by the stochastic policy. However, we would like to emphasize that this different definition of risk has its advantages. Although the formulation of risk-sensitive MDP differs from the standard definition of dynamic risk (Pichlet et al. 2021), which we believe is the same definition as the Markov risk measure [67], they both yield identical optimal values and optimal policies (see Remark 2 and Appendix C for more detail). Further, this different formulation results in a cleaner equivalence with (soft) RMDPs and better convergence property compared with directly optimizing the Markov risk measure, which is known to be possibly non-gradient dominant (c.f. [36])  Consequently, our results provide an alternative approach for the efficient optimization of the Markov risk measure, despite modeling risk in a different manner. However, we also agree with the reviewer that optimizing the Markov risk measure with policy regularization would potentially result in a different optimal solution compared with the solution of our risk-sensitive MDP with policy regularization.

---

> > ### Comment · Reviewer_RSEz · 2023-11-18
> > **Authors falsely claim that all convex risk measure are mixture quasiconcave**
> >
> > Point 1) The authors still seem to confuse the notion of convex risk measure and notion of mixture quasiconcavity. The entropic risk measure is an example of convex and mixture quasi-concave measure. The mean semi deviation of order p, with p>=1 and alpha<=1, is an example of convex but not mixture quasi concave risk measure (see 4.4 of Delage et al. [2019]). The latter therefore is a counter-example to remark 2 that claims falsely that for all convex risk measure, the stochastic policy problem returns the same optimal value as the deterministic policy problem. Bottom line, the remark needs to indicate that the remark only applies for mixture-quasiconcave risk measure.

---

> > > ### Author Response · Authors · 2023-11-20
> > > **Response to Reviewer RSEz 'Authors falsely claim that all convex risk measure are mixture quasiconcave'**
> > >
> > > We would like to sincerely thank the reviewer for the timely response and for giving further suggestions! We now address the reviewer's comments as follows.
> > >
> > > We would like to apologize for our misinterpretation of the reviewer's comments in the first round, especially on the term mixture quasi-concave, and for misstating that ``... convex risk measures (referred to as 'quasi-concave/mixture quasi-concave' by the reviewer)''. From the reviewer's new comment and after a more careful checking of the reference Delage et al. [2019], we realize that most of the confusion and miscommunications are due to the different definitions of (convex) risk measures. Before we elaborate on the details, we would like to note that the results in this paper do not conflict with Delage et al. [2019] and Remark 2 (as well as Theorem 2) holds for the convex risk measure defined in the paper.
> > >
> > > First of all, we would like to note that the definition of convex risk measure in Delage et al. is different from the definition of our paper as well as the classical definitions of convex risk measure in literature [29,30,72]. Here we quote the paragraph at the beginning of page 11 in Delage et al. [2019]: 'We alert the reader that we do not implicitly assume convex risk measures to be monetary as is frequently done in the literature. The explicit distinction between convex and convex monetary risk measures will facilitate a more concise description of the main results in this paper.' From the quote, Delage et al. define the risk measure without the monetary assumption, which is the same as the monotonicity property in the definition of convex risk measures in our paper, whereas monotonicity is a necessary property in our definition as well as in the definition of the Markov risk measure [70]. Specifically, the semi-moment risk measures as mentioned by the reviewer 'typically fails to be monotonic (Shapiroetal.2009, Example 6.19)' (quoted from the end of page 17 in Delage et al.). Thus, our statement that the risk-sensitive MDP obtains optimal deterministic policy is not in conflict with the fact that the optimal policy for optimizing semi-moment risk measures might be stochastic, as semi-moment risk measures are not within the discussion of our paper. We have added the above discussion to the footnote after Remark 2.
> > >
> > >
> > > Secondly, in Delage et al., they stated that even for convex risk measures without monotonicity, the optimal policy is deterministic as long as the feasible region is convex. (Quote from the introduction: 'deterministic decisions are optimal if both the risk measure and the feasible region are convex, or alternatively if the risk measure is mixture-quasiconcave.') This further corroborates that our results are not in conflict with Delage et al.
> > >
> > > Lastly, we would also like to discuss the relation and difference between the mixture quasiconcavity and convexity. Based on our own understanding of the mixture quasiconcavity (Definition 6 in Delage et al.), a risk measure is mixture quasiconcave if $\rho(\theta X + (1-\theta)Y) \ge \min\(\rho(X), \rho(Y)\)$, which is a different, neither stronger nor weaker, definition compared with convexity.
> > > Thus, neither of the two notions includes the other. A risk measure can be convex but not quasiconcave, and vice versa. For this reason,  our results on convex risk measure with monotonicity do not necessarily apply to mixture-quasiconcave risk measure.
> > >
> > > To summarize, we concur with the reviewer that our work has its limitations and could not include convex risk measures without the monotonicity assumption. We also note that our result only holds for convex risk measures with monotonicity instead of mixture quasiconcave risk measures. Accordingly, we have added a footnote in Remark 2 to clearly state that our results hold for convex risk measures with monotonicity; the equivalence might fail if we consider other risk metrics such as semi-moment, where the monotonicity property does not hold. We hope that the above clarifications address the reviewer's concerns. We are happy to further address this if there remains unclearness and misscommunication.

---

> > > > ### Comment · Reviewer_RSEz · 2023-11-20
> > > > **Authors falsely claim that all convex risk measure are mixture quasiconcave**
> > > >
> > > > The author appear to misunderstand the definition of mixture quasi-concavity, and to confuse semi-moment messures with mean semi-deviation mesures. The bottom line of my argument is that the claim made in Remark 2 and Appendix C that "the optimal policy $\pi$ can be chosen as a deterministic policy (for the Markov risk measure optimal Bellman operator)” is unsupported and misleading. Namely, to support this claim the authors would need to demonstrate that: for all $r$, $V$, and $P$, one has that $\max_{\pi} r(s) - \gamma\sigma(\sum_a \pi(a|s)P_{s,a},V)  \leq \max_a r(s) - \gamma\sigma(\pi(a|s)P_{s,a},V)$ or in other words that for all $\pi$,  $\sigma(\sum_a \pi(a|s)P_{s,a},V) \geq \min_a \sigma(P_{s,a},V)$.
> > > >
> > > > In fact, my previous point was that the mean semi-deviation of order $p$, with $p\geq 1$ and $\alpha\leq 1$ is a convex (actually coherent) risk measure (according to the definition of this paper) for which there exists some $V$, transition dynamics $P$, and $\pi$ for which $\sigma(\sum_a \pi(a|s)P_{s,a},V) < \min_a \sigma(\pi(a|s)P_{s,a},V)$ given that this risk measure fails to be mixture quasi-concave (as demonstrated in Proposition 7 of Delage et al. (2019) and illustrated in Example 1). The authors might not understand this argument but should at least realize that their paper lacks supporting arguments for the claim that Remark 2 makes. As it is unsupported and(based on my understanding) wrong, Remark 2 offers a false sense of security in using the “risk sensitive” measures proposed of this paper.
> > > >
> > > > False statements in previous response:
> > > >
> > > > "the semi-moment risk measures as mentioned by the reviewer” : I did not refer to the semi-moment risk measure but to the mean semi-deviation measure. Please refer to section 4.4 (as indicated in my comment) instead of 4.3 of Delage et al. (2019).
> > > >
> > > > “our results are not in conflict with Delage et al.” while Delage et al. (2019)’s result imply that*deterministic decisions are optimal if the risk measure is convex, monotone, and translation invariant irrespective even AND the feasible region is convex*, your results assume that *deterministic decisions are optimal if the risk measure is convex, monotone, and translation invariant for all feasible regions*. While this is not a conflict, their is a conflict in the fact that Example 1 presents an example on non-convex (actually finite) feasible region and convex risk measure (actually coherent) where your assumption is violated.
> > > >
> > > > "a risk measure is mixture quasiconcave if $\rho(\theta X+(1-\theta)Y)\geq \min(\rho(X),\rho(Y))$”: the authors confuse the concept of convex combination of random variables vs. mixtures. To use the notation of their paper, convexity states that $\sigma(p,\lambda V’ + (1-\lambda) V)\leq \lambda\sigma(p,V’) + (1-\lambda)\sigma(p,V)$, where $p$ is a probability vector, while mixture quasiconcavity states that  $\sigma(\lambda p’+(1-\lambda)p,V)\geq \min \{\sigma(p’,V), \sigma(p,V)\}$ for two probability vectors $p$ and $p’$.

---

> > ### Comment · Reviewer_RSEz · 2023-11-18
> > **Further comments on limitation #2 and #3 of Theorem 2**
> >
> > Point 3) I am glad to see that the authors realize that the "robust/risk averse" MDP that they propose is being "optimistic" with respect to the risk associated to the stochasticity of the stochastic policy. To me, this is problematic for an approach said to provide"robust/risk averse" policies. While I find interesting that this formulation has attractive properties from the point of view of optimization, I am worried of what  is the price that one pays in terms of unnecessary risk exposure when implementing slightly modified versions of this model. E.g. the authors agree that a different optimal policy will be obtained if policy regularization is used.
> >
> > Point 2) I agree that the theorem is accurate. Yet, remark 2 seems to indicate that the claim also holds for a risk sensitive formulation of the form $RPR( [r_1,…, r_\infty] ) = \rho_0( r_1 + \gamma \rho_1( r_2 + \gamma \rho_2(… ) a_2,s_3) |a_1,s_2)$
> > Instead of $RPR( [r_1,…, r_\infty] ) = \rho_0( r_1 + \gamma E[ \rho_1( r_2 + \gamma E[ \rho_2(… ) |a_2 ]|s_2)|a_1])$
> > This equivalence fails if policy regularization is used even with mixture quasi concave measures.
> >
> > A further concern that have come to my attention since my initial report is that the risk sensitive MDP is not presented in the standard format of minimizing the risk of long term discounted sum of rewards (see [16] who employs dynamic convex risk measures):
> > $\min_\pi \rho(\sum_t \gamma^t r(s_t,a_t))$.
> > In fact, there is no equivalence between the model in [16] and equation (8) even in the case of mixture quasi concave measure because of the fact that one cannot exploit the scale invariance property of the risk measure to pull the gamma scalings out of the nested operators and give rise to a stationary problem. The work that is based on [70] employs coherent risk measures, which do satisfy scale invariance. While I dont think that this is a major issue, I would expect the paper to comment on this distinct characteristic of the approach.

---

> > > ### Author Response · Authors · 2023-11-20
> > > **Response to Reviewer RSEz 'Further comments on limitation #2 and #3 of Theorem 2'**
> > >
> > > Thank you again for reading our response and giving constructive feedback!
> > >
> > > For point 3), we first thank the reviewer for finding the formulation to be interesting and have attractive properties from the point of view of optimization. We also agree with the reviewer that our work indeed has its limitations, for example, the equivalence of the optimal solution of the risk-sensitive MDP and the Markov risk measure will no longer hold under certain model variations. We have added this limitation to Appendix C in the further revision, here we quote 'However, we would also like to emphasize that when adding policy regularization, the optimal policy might no longer be a deterministic policy. In this setting, the optimal policy and optimal value of the Markov risk measure and the risk-sensitive MDP may not be the same.'
> > >
> > > For point 2), we apologize that Remark 2 leaves an inaccurate impression. Due to space limits, we apologize that we could not include a detailed comparison in the main text, thus we defer a more detailed and accurate discussion into Appendix C (highlighted in blue) and further emphasize there that the definition of our risk-sensitive risk measure is different from the Markov risk measure (i.e. denoted as $RPR = \rho(r_1+ \gamma\rho( r_2+...|a_2,s_3)| a_1,s_2)$ by the reviewer). We have also further emphasized in Remark 2 and its footnote that the equivalence of the optimal solution of Markov risk measure and the risk-sensitive MDP holds because the optimal policy is deterministic, and it would fail if the optimal policy is no longer deterministic due to policy regularization.
> > >
> > > Regarding the further concern of the reviewer that the definition of the value function of the risk-sensitive MDP is different from the risk metric $\rho(\sum_t\gamma^t r(s_t,a_t))$, we agree with the reviewer that these two notions modeled risk in a different manner and we have pointed it out in Appendix C. However, we would like to also clarify that even for Markov risk measures ($RPR = \rho(r_1+ \gamma\rho( r_2+...|a_2,s_3)| a_1,s_2)$) which considers $\rho$ to be a coherent risk measure with the scale invariance property (as defined in [70]), it is not equivalent to the term $\rho(\sum_t\gamma^t r(s_t,a_t))$. Even in [16], although Eq (2) of the paper writes the risk measure as $\rho(\sum_t\gamma^t r(s_t,a_t))$, the true algorithm considers a dynamic risk measure that resembles the Markov risk measure (see eq(6) and (P1) in [16]). Yet we agree with the reviewer that since convex risk measures might not be scale-invariant, and thus the discounting factor $\gamma$ cannot be pulled inside/outside the risk measure, i.e. $r_1 + \gamma\rho(r_2) \neq r_1 + \rho(\gamma r_2)$.}
> > >
> > > Once again, we express our gratitude to the reviewer for the time and effort put into reading the paper and the discussion. We are grateful for the reviewer's help in improving the clearness and rigorousness of the paper!

---

> > > > ### Comment · Reviewer_RSEz · 2023-11-20
> > > > **Further comments on limitation #2 and #3 of Theorem 2**
> > > >
> > > > "we would like to also clarify that even for Markov risk measures, which considers to be a coherent risk measure with the scale invariance property, it is not equivalent to the term $\rho(\sum_t \gamma^t r_t(s_t,a_t)$” I don’t see how the authors can make this claim. [71] demonstrates that for coherent conditional risk measures, solving the markov risk measure fixed point expression (33) is equivalent to minimizing  $\rho(\sum_t \gamma^t r_t(s_t,a_t)$ (see equation at top of p.242).
> > > >
> > > > "Even in [16], although Eq (2) of the paper writes the risk measure as $\rho(\sum_t \gamma^t r_t(s_t,a_t)$, the true algorithm considers a dynamic risk measure that resembles the Markov risk measure.” I disagree with the the authors' argument here that [16] effectively does not solve $\rho(\sum_t \gamma^t r_t(s_t,a_t)$. In fact, [16] adresses a finite horizon problem and employ a time dependent value function, which allows them to rewrite
> > > > $\rho(\sum_t \gamma^t r_t(s_t,a_t)$ as $\rho(r_0(s_0,a_0)+\rho(\gamma r_1(s_1,a_1) + … \rho(\gamma^T r_T(s_T,a_T))))$ using translation invariance. The submitted paper should be clear and upfront about how it departs from the usual interpretation of the dynamic risk measures formulation of the risk averse MDP problems.

---

> ### Author Response · Authors · 2023-11-16
> **Response to Reviewer RSEz (Part 2)**
>
> **[Novelty of Results]** The reviewer pointed out that some of the results in this paper follow closely from existing work. Here we respectfully reference the comments from the reviewer and aim to address these observations by clarifying our contributions and novelty in comparison to prior studies.
>
> - *'Theorem 1 comes from [27]. The equivalence between soft robust optimization and convex risk measure is already established in Ben-Tal et al. (2010). It is therefore no surprise that the result in Theorem 2 would hold.'*
>
> We would like to first clarify that the dual representation theorem (Theorem 1) is not claimed to be a contribution of our work; instead, we reference this theorem from classical literature, as it forms the foundation for the subsequent content of our paper. Regarding the significance of Theorem 2, we agree with the reviewer that the proof intuition follows from Theorem 1 and it might not be surprising for readers who are familiar with both dynamic programming and risk measures. However, we would like to note that it is not a direct or trivial generalization of Theorem 1, as it considers the MDP setting, which has a sequential structure that is more complicated than the static setting considered in Theorem 1. The proof (detailed in Appendix E) also requires a careful combination of dynamic programming with the dual representation theorem. As far as we know, although there are many works that discuss the relationship between robustness and risk-sensitivity (see Appendix A in our paper for more detail), there's no result that explicitly states the exact equivalence between (soft) RMPD and risk-sensitive MDP. We believe that Theorem 2, therefore, remains a distinctive and interesting contribution with its own merits.
>
> - *'Theorem 3, Lemma 2, and Theorem 4 extend the result of [85] from robust MDP to the slightly more general regularized/soft robust MDP.'*
>
> Thank you for the comment and for directing us to the most related literature [85]. Here we briefly list the major differences between our paper and [85]. Firstly, we obtain an explicit analytical form of the policy gradient in Theorem 3, whereas in [85], they consider the Moreau envelope of the value function and it is implicitly computed through a double-loop manner. This immediately leads to the next two differences--different algorithm designs and different theoretical bounds. [85] needs to consider a modified policy gradient with a double-loop, whereas our algorithm removes the double-loop and studies policy gradient without modification. Given the simpler algorithm design, our iteration complexity also has a better dependency on the size of the state and action spaces $|\mathcal{S}|,|\mathcal{A}|$ as well as the distributional shift factor $M$ (denoted as $D$ in [85]).
> However, we also acknowledge that [85] has considered settings more general than our paper, e.g. it also takes the $s$-rectangularity setting into consideration. It would be an interesting next step to see if we could combine insights from both works to have more general and improved results.
>
> - *'Section 5 proposes an algorithm that resembles the approach presented in [24] although I am not aware of a sample complexity result for that class of algorithm.'*
>
> We express our gratitude to the reviewer for recognizing the novelty of our sample complexity result. We would like to first clarify that the objective function considered in [24], namely the exponential utility, differs from our objective function (see more elaboration in the third paragraph in 'Other related works' in Appendix A). Additionally, the learning settings as well as the algorithms are different. [24] considers online learning in the tabular setting and the algorithm is based on Q-learning, whereas we consider offline learning with function approximation and the algorithm resembles policy iteration type of algorithms.
>
> **[Proof to verify]** The reviewer suggests a more detailed verification of equation (25) on page 34. We have already added more details for (25) to the revised paper (highlighted in blue on page 31 in the revised version). However, we find that equation (25) is on page 27 instead of page 34, so we are uncertain if we've tackled the right equation that the reviewer questioned. We would be happy to discuss the technical proofs if there are still equations that are unclear.

---

> > ### Author Response · Authors · 2023-11-16
> > **Response to Reviewer RSEz (Part 3)**
> >
> > **[Typos/clarifications]** We sincerely thank the reviewer for carefully going through the paper and pointing out typos and other possible corrections and improvements. We have incorporated the reviewer's suggestions into the revised version of the paper. (For the notation $\mathcal{T}^\star, \mathcal{T}^\pi$, because it is originally defined as the Bellman operator for the risk-sensitive MDP instead of soft RMDP, we decide to keep the original notation $\mathcal{T}$ rather than $\overline{\mathcal{T}}$ as suggested by the reviewer.) It seems that the page numbers referenced by the reviewer might be slightly inaccurate. We remain open to making additional revisions if our current modifications do not align with the reviewer's intended recommendations.
> >
> > Once again, we are grateful for the reviewer's insightful comments and constructive suggestions! We remain at the reviewer's disposal for any further inquiries or points requiring additional discussion. Your valuable input is greatly appreciated.

---

> > > ### Comment · Reviewer_RSEz · 2023-11-18
> > > **Typos**
> > >
> > > Regarding the list of typos, I apologize for the misleading page numbering. I see that the authors were able to integrate all suggested modification and appreciate the efforts. Here are a few I have found in this round:
> > > - Eq (22) missing parenthesis around $V^{(K)}-V^{(0)}$
> > > - P. 4 "Risk Senstive" should be "Risk Sensitive"
> > > - Eq. (17) and (18), I actually intended to suggest that the $T$ operator should be defined as $\tilde{T}$ to clarify that they refer to an update based on risk sensitivity.

---

> > > > ### Author Response · Authors · 2023-11-20
> > > > **Response to Reviewer RSEz 'Typos'**
> > > >
> > > > We would like to once again thank the reviewer for a thorough reading of our work! We have fixed the typos and changed the notation based on the reviewer's suggestions.

---

> ### Author Response · Authors · 2023-11-21
> **Further Response to Reviewer RSEz**
>
> **[In Response to 'Authors falsely claim that all convex risk measure are mixture quasiconcave']** We sincerely thank the reviewer for providing detailed clarifications to help resolve misunderstandings and miscommunications. We now have a better understanding of the source of the misunderstanding. We concur with the reviewer that for the statement 'the optimal solution of Markov risk measure and risk-sensitive MDP is the same' to be accurate, apart from showing that the optimal solution of risk-sensitive MDP is deterministic (which is proved in Theorem 2), we must also ensure that the optimal solution of the Markov risk measure is deterministic, which requires further assumption of mixture quasiconcavity. We have made revisions in Remark 2 as well as Appendix C. Here we quote the sentences from the revised Appendix C: 'Additionally, when the risk-measure $\sigma$ is mixture quasiconcave (c.f. Delage et al.), it can be shown that the optimal policy $\pi$ for the Markov risk measure can also be chosen as a deterministic policy; thus under this case the Markov risk measure and the risk-sensitive MDP obtain the same optimal value, i.e. $V_{MRM}^\star = \widetilde V^\star$. However, we would also like to emphasize that when adding policy regularization or that the risk measure $\sigma$ is not a mixture quasiconcave (e.g. mean semi-deviation), the optimal policy might no longer be a deterministic policy. In this setting, the optimal policy and optimal value of the Markov risk measure and the risk-sensitive MDP may not be the same.'
>
> Moreover, in case there is still unclearness or miscommunication since there are several ``equivalence'' being discussed, we would like to summarize a few points here, 1) The main equivalence theorem, Theorem 2, focuses on the equivalence of soft robust MDP and risk-sensitive MDP proposed in the paper. This theorem is accurate and it does not discuss the relationship of the optimal solution of Markov risk measure and risk-sensitive MDP. 2) Remark 2 and Appendix C discuss the relationship between the risk-sensitive MDP and Markov risk measure. We have revised the remark and appendix based on the discussion.
>
> Again, we would like to express our regards and gratitude for the reviewer's time and effort in keeping the discussion and clarifying the confusion. We remain at the reviewer's disposal for any points requiring further discussion.
>
> **[In Response to 'Further comments on limitation 2 and 3 of Theorem 2']** We apologize for misunderstanding the reviewer's comments. Previously, we interpreted the $\rho$ in $\rho(\sum_t^T\gamma^t r(s_t,a_t))$ in the reviewer's comment to be any type of coherent risk measures. After reading the reviewer's clarification, we agree with the reviewer that when $\rho_{0,T}$ is a time-consistent dynamic risk measure, it can be written as, $\rho_{0,T}(r_1, r_2, \ldots, r_T) = r_0(s_0, a_0) + \rho_0(\gamma r_1(s_1,a_1) + \dots + \rho_{T-1}(\gamma^T r_T(s_T, a_T)))=\rho_0(\rho_1(\ldots \rho_T(\sum_{t=0}^T \gamma^t r(s_t,a_t))\ldots))$, where $\rho_t, t=0,\ldots, T$ are one-step conditional risk measures as defined in [71] and satisfy the properties listed in the bottom of p.241.  Since we consider a different risk measure, it can no longer be written in this formulation. We have added a footnote in Remark 2 (highlighted in blue) to emphasize that our definition is different from the usual interpretation of the dynamic risk measures formulation of the risk-averse MDP problems.
>
> We would like to thank the reviewer again for the timely replies and active discussions. We hope that our paper revision as well as response clarifies the miscommunication and addresses the reviewer's questions and concerns. We are more than happy to engage in further discussion if there are any additional questions or concerns from the reviewer.

---

### Official Review · Reviewer_SuJM · 2023-10-30

**Soundness:** 3 good
**Presentation:** 2 fair
**Contribution:** 3 good
**Rating:** 6
**Confidence:** 4

**Summary:**

This work introduces an alternative formulation of risk-sensitive MDP (RS-MDP in the sequel) that enables to show equivalence with regularized-RMDPs. Leveraging this equivalence, the authors derive a converging risk-sensitive policy-gradient method that asymptotically recovers the optimal value and policy of Markov RS-MDPs. Thus, they tackle the potential non-gradient domination of the Markov RS-MDP setting. Focusing on KL-regularized RMDP, an offline algorithm called RFZI is then introduced, that presumably improves upon the RFQI algorithm of [61] in terms of computational complexity and required assumptions.

**Strengths:**

The paper is well-written and the mathematical setting rigorously specified.

The reformulation via regularized-robustness to provide a converging policy-gradient is crafty.

**Weaknesses:**

- It would be preferable to give more space to motivate this work rather than explaining the method (see suggestions below)

- To me it is a bit frustrating to see so many references in the bibliography with no discussion of them in the text body. Specifically, how does the regularization term of this work compare to [18]? Also, the regularized+robust objective considered in this work reminds that of [32] who also combine the two to derive a convex formulation of RMDPs. This leads me to the following question: does the same convex formulation applies to the authors' regularized RMDPs? Is it the aforementioned convex property that ensures gradient domination in this setting as well?

- As mentioned in Rmk 4, sampling from $\hat{P}^{\theta}$ rather than the nominal represents the main challenge in estimating the policy-gradient. In this respect, we refer to the work [101] that explicitly formulate the policy gradient in RMDPs: by focusing on $\ell_p$-ball constrained uncertainty sets, the authors are able to explicitly derive the worst transition model and as a result, a robust policy-gradient that is more amenable to be estimated from nominal samples. In fact, I assume that specific assumptions on the convex risk measure $\sigma$ can similarly be leveraged to explicit a risk-sensitive gradient. For example, departing from the equivalence between coherent risk measures and RMDPs with risk envelope, one may write a robust policy-gradient as in [101] and broaden the class of measures to be considered while preserving tractability of the gradient.

- The offline learning part is a bit of an outlier in this paper. In particular, I do not understand the motivation to section 5. Simulation experiments on the other hand, would study the risk-sensitive behavior of regularized-robust optimal policies and compare them to robust optimal policies.

- It is not clear to me how the performance gap of Thm 5 compares to that of RFQI [61]. The authors briefly mention the computational cost incurred by introducing dual variables in [61], but the performance gap obtained from each approach should further be discussed. More generally, Sec. 5 raises several open questions without further discussing them, which is a bit frustrating: can Assm 2 be avoided? How to empirically select the penalty factor $\beta$ and shrink the performance gap? Is the exponential dependence a proof artifact or is it intrinsic to the problem?

[101] N. Kumar, E. Derman, M. Geist, K.Y. Levy, S. Mannor. Policy Gradient for Rectangular Robust Markov Decision Processes. *Advances in Neural Information Processing Systems*, 2023. https://arxiv.org/pdf/2301.13589.pdf

**Questions:**

Questions:
- I found the second paragraph of Sec. 1 quite confusing: "when the model is given", "in cases of unknown models... most are model-based for tabular cases", "model-free setting". How are the shortcomings mentioned in this paragraph addressed in the present work? How do the authors distinguish between "unknown model" and "model-free"? Does it refer to the uncertainty set or the nominal model? For example, in my understanding, [48] is model-free in the sense that it does not take the uncertainty set as given but estimates it from data. Accordingly, most robust learning approaches are model-based in the sense that they consider the uncertainty set to be known and solve robust planning from there.
- Rmk 1 was also confusing to me. What do the authors mean by "the boundary of the uncertainty set"? What is a "hard boundary"? Referring here to the fact that $(\hat{P}_t)_t$ are in the whole simplex for regularized RMDPs, as opposed to a specified uncertainty set $\mathcal{P}$ in RMDPs would greatly help understanding the authors' motivation for that setting. Also, is this remark related to the discussion on "model-free" vs "model-based" in the intro?
- Thm 1 seems to simply relate the double Fenchel conjugate with the original function. Is it what it does? If so, it would be useful to mention it in the text and if not, how is this result different?
- Could you please provide an intuitive explanation as to why the value function is different from Markov risk measures of previous works when the policy is stochastic, but the same when it is deterministic?


Suggested corrections:
- Last line of Sec. 1: "appendix" without capital letter
- Typo "valule" right before Eq. (7)
- Eqs. (12), (14), (18) are numbered without being referred to in the text.
- Missing capital letters in references: Markov in [6, 10, 32, 35, 36, 43, 48, 54, 59, 71, 94] and others, Bellman in [4]
- Published/workshop papers should be referred as such: [16, 17, 20, 21, 22, 40, 44, 52, 61, 78, 85, 96]
- I do not see the utility of the paragraph on finite-horizon (referred to as (*) in the following item), to explain the fixed point equation (7). This is standard in MDPs and would save some valuable space that seems to be missing in this work. Instead, it would be useful to briefly mention the properties of the operator (7) -- monotonicity, sub-distributivity, contraction -- and how they follow from those of the convex risk measure $\sigma$.
- Sec. 3 is very short. I would rather incorporate the "risk-sensitive MDPs"-paragraph of Sec. 2 into that one.
- In general, I think that the authors could and should shrink some parts of the text to better motivate their approach, and defer some explanation details of their method to the appendix. For example, Ex. 2 could be removed/put into appendix, as it is not further used in the paper (as opposed to Ex. 1); the paragraph (*) could be removed completely. As such, instead of declaring "Due to space limit, we defer a detailed literature review and numerical simulations to the appendix", the authors could provide more synthetic exposition of their method, but detail more on its positioning w.r.t. previous works. Please see my questions in that respect.
- In the same spirit of saving space, the "other notations" paragraph at the end of Sec. 2 could be written inline and deferred to the "MDPs" paragraph at the beginning of Sec. 2; Remarks may be written as regular text.
- Typo in Assmp 1

---

> ### Author Response · Authors · 2023-11-16
> **Response to Reviewer SuJM (Part 1)**
>
> We first sincerely thank the reviewer for carefully reading our paper and appreciating our contributions! We greatly appreciate the constructive suggestions and valuable questions from the reviewer. Prior to addressing the reviewer's comments, we would like to note that, due to the revision, reference/equation/remark numbers may differ. To maintain consistency with the reviewers' comments, *our responses will refer to the original version's reference/equation/theorem numbers unless explicitly specified otherwise.* Additionally, based on the insightful suggestion of Reviewer RSEz, we changed the term 'regularized robust MDP' to 'soft robust MDP'. We now address the questions and concerns raised by the reviewer as follows:
>
> Response to the reviewer's concerns in the 'Weakness' section:
>
> **[Comparison with [18] and [32]]** The reviewer raised the question of how our regularized MDP compares with the regularization term in [18]. We would like to first apologize for our misleading use of the term 'regularized robust MDP' (which is also pointed out by the third comment in the 'weakness' section of  Reviewer RSEz). As is also pointed out by Reviewer RSEz, the regularization in [18] and ours are different in the sense that the 'regularized MDP' in [18] considers regularization on the policy whereas in our setting the regularization is on the transition probability matrices. To avoid confusion, as suggested by Reviewer RSEz, we have changed 'regularized robust MDP' to 'soft robust MDP' in the revised version of the paper. Thank you for raising this valuable question!
>
> The reviewer also asked whether the same convex formulation in [32] applies to the regularized RMDPs (soft RMDPs). We are also interested in the same question and we actually had a brief discussion with one of the authors in [32]. Based on the discussion and our preliminary attempt, it might not be a trivial combination, and it is an interesting future direction to study how to leverage our formulation to get simplified convex formulations for (soft) RMDPs. In response to the reviewer's question of whether the gradient domination property is ensured by the above convex formulation, our current proof leverages the performance difference lemma and the key technique is inspired by the proof of gradient domination in standard MDP [2]. It would be an interesting research topic to study if the gradient domination property can be proved from the convex formulation perspective, but currently, it is unclear to us whether the convex formulation is directly related to the gradient domination property.
>
> **[Application of techniques in [101]]** The reviewer asked whether we could apply similar techniques in [101] (which we believe is the same reference as [44] in the original paper, we have updated the publisher info of [44] in the revised paper) by explicitly deriving the worst transition model $\widehat{P}^\pi$. We agree with the reviewer that in certain cases the worst-case probability transition model can indeed obtain an explicit/analytical form that can simplify the problem to a certain extent (e.g. entropy risk measure $\widehat{P}^\pi$ takes the form $\widehat{P}^\pi(s'|s,a)\propto P(s'|s,a)\exp(-\beta V^\pi(s'))$). We believe that in this case we could leverage techniques such as importance sampling or rejection sampling to make gradient estimation more amenable. However, based on our own calculation, the sample complexity analysis is more technically involved, which is one of the main reasons why we considered the RFZI algorithm instead of policy gradient algorithm for the sample-based learning setting and we left it as the next step of our research to study the performance of sample-based policy gradient for risk-sensitive MDPs and (soft) RMDPs.

---

> > ### Author Response · Authors · 2023-11-16
> > **Response to Reviewer SuJM (Part 2)**
> >
> > **[Motivation for the offline learning algorithm RFZI]** The reviewer has raised questions regarding the motivation behind the offline learning algorithm presented in Section 5. The primary motivation behind Section 5 is to introduce a sample-based learning algorithm for risk-sensitive MDPs and soft robust MDPs. This is essential as Section 4 only focuses on the learning algorithm under full information and explores the iteration complexity. Ideally, it would be better to design sample-based policy gradient methods.  However, as mentioned in Remark 4, it is difficult to estimate the policy gradient by samples, which is why we first consider the RFZI algorithm which resembles policy iteration rather than policy gradient. The reason for considering the specific setting of offline learning algorithm with nonlinear function approximation is that, as mentioned in the introduction, it is a less-studied setting for RMDPs and we hope that our results could bring insights to make algorithm design and theoretical analysis easier for this less-studied scenario. We apologize for not explaining our motivations clearly and we hope that the above explanation clarifies the confusion of the reviewer. We have included part of the above discussion at the beginning of Section 5 and we are happy to have further discussions. We have also added more experimental results that study the robustness/risk-sensitive behavior of soft robust optimal policies and compare it with the robust optimal baseline based on the reviewer's suggestions (see Figure 4 and its related paragraphs in Appendix B).
> >
> > **[Performance comparison with the RFQI algorithm in [61] and open questions in Section 5]** The reviewer suggests adding a comparison with the sample complexity bound of RFQI algorithm in [61]. We first thank the reviewer for the valuable suggestion. We agree with the reviewer that a comparison between the results of the two algorithms is necessary and we have added related discussions to the revised paper (highlighted in blue). Here we quote the discussion: 'We also compare our performance bound with the RFQI algorithm which considers a similar offline learning setting and obtains sample complexity of $O\left(\frac{\log(|\mathcal{F}||\mathcal{G}|)}{(\beta\epsilon)^2(1-\gamma)^6}\right)$, where $\beta$ in their setting is the radius of the uncertainty set. Note that both results share the same dependency on $\epsilon$ and the concentrability constant $C$. However, the bound in [61] includes an additional term on the size of the dual variable space $\log |\mathcal{G}|$, whereas we have the exponential dependence term $e^{\frac{2\beta}{1-\gamma}}$.' Yet we would also like to note that the two settings are not completely comparable as we consider different types of uncertainty sets and the algorithms are very different.
> >
> > The reviewer also raised questions on the assumption and the theoretical bounds in Section 5. We address the questions as follows:
> >
> > - *'Can Assumption 2 be avoided?'* We would like to refer the reader to Remark 6 (now Remark 5 in the revised paper) where we provide a detailed discussion on Assumption 2. In particular, Assumption 2 is adapted from the
> > corresponding notions defined for the standard MDP setting [11], where they also demonstrate the necessity of this assumption for standard RL with function approximation. However, for the risk-sensitive and soft robust MDP settings, we currently don't have theoretical results to verify the necessity of Assumption 2. It would be an interesting
> > open question whether Assumption 2 is also necessary.
> >
> >
> > - *'How to empirically select the penalty factor and shrink the performance gap?'* Remark 7 (now Remark 6 in the revised paper) provides a brief discussion on the choice of $\beta$ where we quote as follows:  the term $\beta^{-1}e^{\frac{\beta}{1-\gamma}}$ first decreases and then increases with $\beta$ as it goes from $0$ to $+\infty$, suggesting that the hyperparameter $\beta$ also affects the learning difficulty of the problem. The choice of $\beta$ should be neither too large nor too small, ideally on the same scale as $1-\gamma$. We would also love to add further discussions if the above discussion is not detailed enough!
> >
> > - *'Is the exponential dependence a proof artifact or is it intrinsic to the problem?'* Thanks for the question! It is still unclear to us whether it is a proof artifact or intrinsic in our setting, however, there are results under similar settings that suggest this exponential dependency on parameter $\beta$ and the effective length $\frac{1}{1-\gamma}$ is fundamental (e.g. Theorem 3 in [23]). We have added the above discussion to Remark 6 in the revised paper (previously Remark 7 in the original version, highlighted in blue).

---

> > > ### Author Response · Authors · 2023-11-16
> > > **Response to Reviewer SuJM (Part 3)**
> > >
> > > Response to the reviewer's questions in the 'Questions' section:
> > >
> > > **[Discussion between terms 'unknown model', 'model-based', 'model-free']** We apologize for the confusion. By 'unknown model' we refer to the setting where the nominal probability transition $P$ model is unknown. Both 'model-based' and 'model-free' methods belong to the setting with 'unknown model'. Although 'Model-based' methods don't have knowledge about the nominal probability transition, they generally keep an empirical estimate of the nominal model $P^{\textup{est}}$ and the algorithms rely heavily on the estimated model $P^{\textup{est}}$. However, in comparison 'model-free' algorithms don't require this empirical estimation step and thus might be more suitable for large-scale problems. We categorize [48] as model-based because their algorithm requires an empirical estimation (denoted as $\hat{P}_k$ in [48]) as an input. Our algorithm only requires learning the Z-function and does not rely on an estimated model and thus belongs to the 'model-free' category. We have added the above discussion as a footnote in the revised paper. Additionally, we would also like to clarify that for RMDP, people generally refer 'known/unknown model' as the nominal model rather than the uncertainty set. In the RMDP setting, the uncertainty set is generally assumed to be known and pre-given as an algorithm input. Based on our understanding of [48], we believe they still require knowledge of the uncertainty set but don't require knowledge of the nominal model.
> > >
> > > **[Clarification of Remark 1]** We apologize for the imprecise language and thank the reviewer for pointing out potential improvements. We have revised the sentence based on the suggestion of the reviewer. Regarding the relationship between 'model-based' and 'model-free' learning, as mentioned in the previous paragraph, for RMDP, people generally refer 'known/unknown model' as the nominal model rather than the uncertainty set, so it is not directly related.
> > >
> > > **[Relationship between Theorem 1 and Fenchel conjugate]** The reviewer's understanding is correct. In the finite-dimensional case, the penalty function $D$ is the Fenchel conjugate of $\sigma$. We have added a short remark in our revised paper to make it clearer. We would also like to clarify that Theorem 1 is not claimed to be a contribution of our work; instead, we reference this theorem from classical literature [27], as it forms the foundation for the subsequent content of our paper.
> > >
> > > **[Intuitive explanation of the difference of the value function for risk-sensitive MDP and the Markov risk measure]** Thanks for the question. Due to the space limit, the more detailed explanation and comparison of the risk-sensitive MDP and the Markov risk measure was deferred in Appendix C. Intuitively speaking, the Markov risk measure also takes the risk generated by the randomness of the policy into account whereas our definition treats the randomness from the policy in a risk-neutral manner and only considers risk from the uncertainty of the transition probability. This intuitively explains why the two notions are equivalent for deterministic policies but not for stochastic policies. We have added the above intuitive explanation to Appendix C in the revised paper (highlighted in blue). We would also like to emphasize that our definition of risk-sensitive MDPs has its own benefits. It results in a cleaner equivalence with (soft) RMDPs and better convergence property compared with directly optimizing the Markov risk measure, which is known to be possibly non-gradient dominant (c.f. [36]).  Consequently, our results provide an alternative approach for the efficient optimization of the Markov risk measure, despite modeling risk in a different manner.

---

> > > > ### Author Response · Authors · 2023-11-16
> > > > **Response to Reviewer SuJM (Part 4)**
> > > >
> > > > **[Suggested corrections]** We extend our sincere gratitude to the reviewer for carefully going through the paper and pointing out typos, inaccurate reference information, and other corrections! We have fixed the typos and references, deleted unnecessary content and remarks, and revised our paper based on the suggestions from the reviewer. We have also provided additional elaboration in the introduction, specifically addressing the motivation behind our offline learning algorithm and how it overcomes the shortcomings highlighted earlier. Given the richness of the field of robust and risk-sensitive learning, we regret that we could not include all the literature in the main text. We still kindly refer the reviewer to Appendix A for a more comprehensive comparison with prior works.
> > > >
> > > > We would also like to note that the publisher information is not updated for a few papers in the reference because they are not officially published yet based on our literature search. We would be happy to make further updates on the references if there are any further questions or suggestions!
> > > >
> > > > With regard to the reviewer's suggestion of putting the definition of risk-sensitive MDP into Section 3, we apologize that, after careful consideration, we still decide to keep the original structure. We believe that maintaining Section 2 as a self-contained section with all the preliminaries logically aligns with the overall flow of the paper. We are open to further discussion regarding the paper's structure and appreciate the reviewer's input on this matter.
> > > >
> > > >
> > > > Once again, we sincerely thank the reviewer for dedicating time and effort to a thorough review of our paper. We hope that our response could clarify the questions and concerns of the reviewer and we are happy to address any other questions the reviewer has!

---

### Meta-Review · Area_Chair_iNmy · 2023-12-07

**Metareview:**

In this paper, the authors establish an equivalence between risk-sensitive MDPs and a class of soft-robust MDPs. They use this equivalence and derive the policy gradient theorem for both problems, and prove gradient domination and global convergence in the tabular setting with direct parameterization. They also propose a sample-based offline learning algorithm (RFZI), inspired by RFQI, for a specific soft robust MDP problem with a KL-divergence regularizer, which is equivalent to the risk-sensitive MDP with a nested entropic risk measure.

All the reviewers found the paper well-written and clear in formulation and execution. They believe some of the results in the paper are novel and non-trivial. They also appreciate that the paper establishes connection between two important models for decision-making under uncertainty: robust and risk-sensitive MDPs.

There are concerns about the novelty of several results reported in the paper (see 2nd weakness by Reviewer RSEz), the lack of proper motivation for the work (see 1st weakness by Reviewer SuJM), and a few technical issues, mainly raised by Reviewers SuJM and RSEz. I totally agree with Reviewer RSEz that several results follow closely from existing work. I would suggest that the authors make this clear in their next revision. Also, motivate their work better and explain in which ways establishing this connection between risk and robustness could be useful. Finally, take the technical comments by Reviewer RSEz with regards to Theorem 2 into account in your next revision.

**Justification For Why Not Higher Score:**

Concerns on limited novelty, lack of proper motivation, and some technical issues.

**Justification For Why Not Lower Score:**

Well-written and executed paper, some novel and non-trivial results, and establishing connection between two important models for decision-making under uncertainty: robust and risk-sensitive MDPs.

---

### Decision · Program_Chairs · 2024-01-16

Accept (poster)